# RETHINKING THE FLOW-BASED GRADUAL DOMAIN ADAPTION: A SEMI-DUAL TRANSPORT PERSPECTIVE

## ABSTRACT

Gradual domain adaptation (GDA) aims to mitigate domain shift by progressively adapting models from the source domain to the target domain via intermediate domains. However, real intermediate domains are often unavailable or ineffective, necessitating the synthesis of intermediate samples. Flow-based models have recently been used for this purpose by interpolating between source and target distributions; however, their training typically relies on sample-based log-likelihood estimation, which can discard useful information and thus degrade GDA performance. The key to addressing this limitation is constructing the intermediate domains via samples directly. To this end, we propose an Entropy-regularized Semidual Unbalanced Optimal Transport (E-SUOT) framework to construct intermediate domains. Specifically, we reformulate flow-based GDA as a Lagrangian dual problem and derive an equivalent semi-dual objective that circumvents the need for likelihood estimation. However, the dual problem leads to an unstable min–max training procedure. To alleviate this issue, we further introduce entropy regularization to convert it into a more stable alternative optimization procedure. Based on this, we propose a novel GDA training framework and provide theoretical analysis in terms of stability and generalization. Finally, extensive experiments are conducted to demonstrate the efficacy of the E-SUOT framework.

## 1 INTRODUCTION

Unsupervised Domain Adaptation (UDA) (Pan & Yang, 2010; Tzeng et al., 2017; Long et al., 2015; Courty et al., 2014; 2017a), which transfers knowledge from a well-trained source domain to a related yet unlabeled target domain, is of great importance across fundamental application areas. For example, in recommender systems (Liu et al., 2023; Zheng et al., 2024), a cold-start user has no interaction history with new items, so domain adaptation helps transfer user and item knowledge from an existing system to improve recommendations. Similar scenarios occur in machine translation, where a model trained on high-resource language pairs like English-French can be adapted to translate between English and low-resource languages with limited parallel data (Gazdieva et al., 2023). These scenarios highlight the importance of conducting UDA to bridge domain gaps and ensure reliable performance in real-world applications.

Despite these methodological advances, directly performing UDA can be brittle when the source–target shift is substantial or class overlap is weak. In such cases, one-shot alignment often degrades discriminability and amplifies pseudo-label errors during self-training. This challenge motivates a transition from the traditional UDA setting to the Gradual Domain Adaptation (GDA) setting (He et al., 2024), where adaptation proceeds through a sequence of intermediate distributions that progressively bridge the domain gap. A key aspect of generating intermediate domains in GDA is to interpolate between the source and target domains. Various methods have been proposed to construct such intermediate domains, among which flow-based approaches (Kobyzev et al., 2020; Papamakarios et al., 2021) have attracted increasing attention, primarily due to their property of preserving probability density along the transformation path, thereby enabling consistent and stable probability densities without distortion or loss of information. To drive the samples from the source domain towards those of the target domain, it is necessary to design an appropriate driving force, typically derived from a discrepancy metric. Among these metrics, $f$-divergence (Sason & Verdú, 2016) is most widely used due to its computational efficiency, empirical effectiveness, and principled formulation within the framework of geometry for probability distributions (Amari, 2016).

Despite the success of flow-based approaches in GDA (Sagawa & Hino, 2025; Zhuang et al., 2024; Zeng et al., 2025), we argue that directly applying standard flow-based models leads to suboptimal performance. Specifically, existing flow-based frameworks utilizing $f$-divergence often require the explicit estimation of target domain probability density functions (PDFs) from available target samples (In our setting, for simplicity, we treat both log PDF and its gradient, also known as score function, as forms of density estimation, since they characterize the underlying data distribution.) (Vincent, 2011; Santambrogio, 2017; Ambrosio et al., 2005), whereas the subsequent GDA process relies on these estimated (normalized / unnormalized) PDFs to drive the source-to-target transfer. For example, Zhuang et al. (2024) estimate the unnormalized target domain PDF in the score function form and generate intermediate domains via Langevin dynamics. Consequently, the quality of the intermediate domain heavily depends on the accuracy of the estimated target PDF; if this estimation is inaccurate, the performance of the downstream task is likely to suffer significantly.

To address these limitations, we propose a novel flow-based GDA framework E-SUOT, which leverages the semi-dual formulation of gradient flows. Rather than explicitly estimating PDFs, we recast flow evolution as an optimization problem that combines an $f$-divergence term with a Wasserstein distance regularization term, enabling sample transport toward the target domain without reliance on PDF estimation. However, as the semi-dual reformulation inherently leads to an adversarial training paradigm that can compromise stability and performance, we introduce entropy regularization to the objective to guarantee the stability of the training process. Based on this, we summarize the algorithm for E-SUOT-based intermediate domain generation, prove the convergence of our E-SUOT framework, and empirically demonstrate its effectiveness on representative GDA tasks. Extensive experiments validate that E-SUOT achieves superior performance compared with existing methods.

**Contributions.** The main contributions of this paper are summarized as follows:

- We develop a semi-dual formulation for intermediate domain generation in flow-based GDA, which eliminates the need for explicit estimation of the target-domain PDF—whether normalized or unnormalized—or its score-based representation.

- We introduce an entropy regularization term to address the unstable issue inherent in the semi-dual formulation, resulting in the novel and stable E-SUOT framework.

- We conducted various experiments to demonstrate the superiority of the proposed E-SUOT approach compared to prevalent approaches.

## 2 PRELIMINARIES

### 2.1 SETTINGS AND NOTATIONS

In GDA, we consider a labeled source domain, $T - 1$ unlabeled intermediate domains, and an unlabeled target domain. Let the input space be $\mathcal{X}$ and the label space be $\mathcal{Y}$. We denote inputs as $x \in \mathcal{X}$ and labels as $y \in \mathcal{Y}$. We index the domains by $t \in \{0, 1, \ldots, T\}$, where $t = 0$ denotes the source domain and $t = T$ denotes the target domain. Each domain induces a marginal distribution $p_t$ over $\mathcal{X}$. Let $\mathcal{H}$ be a hypothesis class of classifiers $h : \mathcal{X} \to \mathcal{Y}$. We assume that each domain admits a labeling function $q_t \in \mathcal{H}$. Given a loss function $\mathcal{L} : \mathcal{Y} \times \mathcal{Y} \to \mathbb{R}_{\geq 0}$, the generalization error of $h$ on domain $t$ is defined as $\varepsilon_{p_t}(h) = \mathbb{E}_{p_t(x)}\big[\mathcal{L}\big(h(x), q_t(x)\big)\big]$. A source classifier $q_0 \in \mathcal{H}$ can be learned via supervised learning on the source domain with minimal error $\varepsilon_{p_0}(q_0)$. The objective of GDA is to evolve $q_0$ through the intermediate domains to a classifier $h_T$ so as to minimize the target error $\varepsilon_{p_T}(h_T)$.

### 2.2 FLOWS FOR INTERMEDIATE DOMAIN GENERATION

A flow describes the time-dependent evolution of particles induced by a smooth invertible (diffeomorphic) map. Based on this, the intermediate domains can be seen as a discretization of a continuous flow linking source and target distributions. This motivates flow-based models, which evolve a distribution over a fixed time horizon while preserving normalization, and are thus well-suited for GDA. From the flow perspective, intermediate domains are generated by the following ordinary differential equation:

$$\frac{\mathrm{d}x_t}{\mathrm{d}t} = v_t(x_t) = -\nabla \frac{\delta \mathbb{D}[p(x_t), p_T(x)]}{\delta p(x_t)}, \quad x_{t=0} = x_0, \tag{1}$$

where $p(x_t)$ is the (empirical) PDF induced by $\{x_{t,i}\}_{i=1}^{\mathrm{N}}$, and we desire the law $p(x_T)$ to approximate the target $p_T(x)$. Here $v_t : \mathcal{X} \to \mathcal{X}$ is the velocity field. The core design problem is to choose $v_t$ so that $p(x_t) \xrightarrow[t \to T]{} p_T(x)$. A principled approach is to define $v_t$ as the steepest descent direction of some discrepancy functional $\mathbb{D}[p(x_t), p_T(x)]$ between $p(x_t)$ and $p_T(x)$ as demonstrated in the second equal sign in Eq. (1). Notably, $\delta/\delta p$ denotes the first variation, and the second equality sign is called "gradient flow".

Among various choices, $f$-divergences are favored in GDA for their task-aligned objectives, stable probability-preserving dynamics, and efficient computation when compared to alternatives such as Sinkhorn divergence and maximum mean discrepancy (Glaser et al., 2021). For an $f$-divergence,

$$\mathbb{D}_f[p(x_t), p_T(x)] = \int f\left(\frac{p(x_t)}{p_T(x)}\right) p_T(x) \, \mathrm{d}x, \qquad (2)$$

with $f : (0, \infty) \to \mathbb{R}$ convex and $f(x) = 0$ if and only if $x = 1$. A canonical example is the Kullback–Leibler (KL) divergence with $f(u) = u \log u$. In this case,

$$v_t(x_t) = \nabla \log p_T(x) - \nabla \log p(x_t), \qquad (3)$$

and, in the weak partial differential equation sense (Evans, 2022; Liu, 2017), the induced dynamics yield the classical *Langevin dynamic* (Welling & Teh, 2011; Santambrogio, 2017).

Intuitively, applying the forward Euler scheme with step size $\eta$ to the gradient flow in Eq. (1) under an $f$-divergence yields a discrete-time generation for the intermediate domain, which is equivalent to solving a 2-Wasserstein-distance–regularized optimization problem as (see Section C.1):

$$x_{t+\eta} = x_t - \eta \nabla \frac{\delta \mathbb{D}_f[p(x_t), p_T(x)]}{\delta p(x_t)} \Rightarrow p(x_{t+\eta}) = \underset{\rho(x) \in \mathcal{P}_2(\mathbb{R}^{\mathrm{D}})}{\arg\min} \frac{1}{2\eta} \mathcal{W}_2^2(\rho(x), p(x_t)) + \mathbb{D}_f[\rho(x), p_T(x)],$$
$$(4)$$

where $\mathcal{P}_2(\mathbb{R}^{\mathrm{D}})$ denotes the Wasserstein space (Villani et al., 2009), which is the set of the distributions with finite second moment. Here $\mathcal{W}_2$ is the 2-Wasserstein distance, whose definition is given as follows:

$$\mathcal{W}_2^2(\rho, \xi) = \inf_{\pi \in \Pi(\rho, \xi)} \iint \|x - y\|_2^2 \, \pi(x, y) \, \mathrm{d}x \, \mathrm{d}y, \qquad (5)$$

and $\Pi(\rho, \xi)$ is the set of joint distribution on $\mathbb{R}^{\mathrm{D}} \times \mathbb{R}^{\mathrm{D}}$ with marginal distributions $\rho$ and $\xi$.

## 3 METHODOLOGY

### 3.1 MOTIVATION ANALYSIS

Flow-based approaches, exemplified by gradient-flow methods, interpolate between the source and target distributions by gradually minimizing a discrepancy measure, typically an $f$-divergence, between the two domains. The success of these methods in GDA tasks critically depends on accurately estimating the target distribution's probability density function (PDF). Given a reliable estimate, one can construct a velocity field that progressively pushes source samples toward the target distribution.

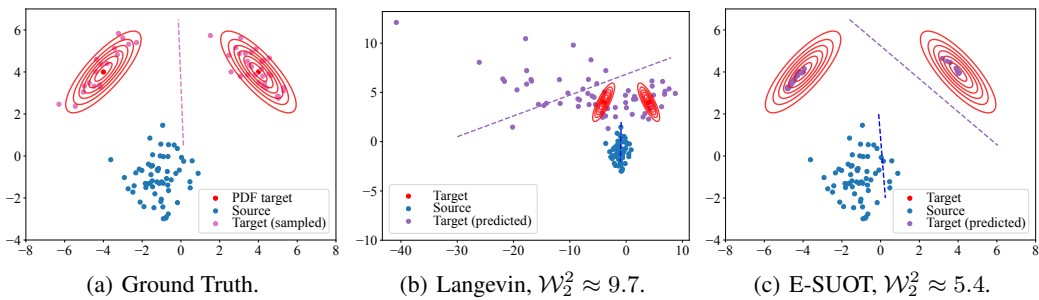

(a) Ground Truth.     (b) Langevin, $\mathcal{W}_2^2 \approx 9.7$.     (c) E-SUOT, $\mathcal{W}_2^2 \approx 5.4$.

Figure 1: Illustrative Example: Comparison between Langevin Dynamics and E-SUOT, where the corresponding decision boundary is shown as a dashed line.

However, directly estimating the PDF from target domain data is generally ill-posed (Vincent et al., 2010; Song et al., 2020). When the estimate is inaccurate, the induced velocity field can push

samples into low-probability regions of the target distribution, causing a substantial shift between the generated and true target domains and degrading downstream task performance. To illustrate this issue, we compare ground-truth target samples with those obtained via Langevin dynamics and E-SUOT in Figs. 1(a) to 1(c). The PDF for the target domain is estimated using denoised score matching (Vincent, 2011). In addition, we also report the Wasserstein distance between the predicted and ground-truth samples (relative to Fig. 1(a)) in the captions of Figs. 1(b) and 1(c), which constituted the lower generalization bound for GDA tasks. From Figs. 1(b) and 1(c), it is evident that when the estimated log-likelihood function is inaccurate, the samples generated for the target distribution deviate substantially from the ground truth and yield a large Wasserstein distance, which may ultimately limits performance on GDA tasks. Although neither method perfectly recovers the ground truth distribution, E-SUOT shows a clearer alignment with the major modes and the corresponding decision boundary of the target domain, resulting in a substantially lower Wasserstein distance compared to Langevin dynamics. In summary, the key questions addressed in this paper can be summarized as follows: How can we generate intermediate domains without compromising the accuracy of the target domain? How can robust intermediate domain generation be achieved within this framework? Does this approach improve the performance for GDA task?

### 3.2 Dual-Form Transportation for Intermediate Domain Generation

As shown in Eq. (4), simulating the gradient flow to generate intermediate domains is precisely equivalent to solving a Wasserstein-distance-regularized optimization problem. This insight opens up a practical alternative: *instead of explicitly estimating the target domain's probability density, one can guide source samples by directly tackling this optimization formulation*. Thus, we have the following proposition regarding the solution property of the problem defined in Eq. (4):

**Proposition 1.** *Consider the following primal problem:*

$$\mathcal{L}^{Primal} = \underset{\rho(x) \in \mathcal{P}_2(\mathbb{R}^D)}{\arg \min} \frac{1}{2\eta} \mathcal{W}_2^2(\rho(x), p(x_t)) + \mathbb{D}_f[\rho(x), p_T(x)]. \tag{6}$$

*This problem is equivalent to the following semi-dual formulation:*

$$\mathcal{L}^{SemiDual} = \sup_w \mathbb{E}_{p(x_t)} \left[ \inf_{\boldsymbol{T}} \left( \frac{1}{2\eta} \|\boldsymbol{T}(x_t) - x_t\|_2^2 - w(\boldsymbol{T}(x_t)) \right) \right] - \mathbb{E}_{p_T(x)}[f^\star(-w(x))], \tag{7}$$

*where $w : \mathbb{R}^D \to \mathbb{R}$ is a measurable continuous function, $\boldsymbol{T} : \mathbb{R}^D \to \mathbb{R}^D$ is the transport map, and $f^\star$ denotes the convex conjugate of $f$, defined as $f^\star(z) := \sup_{y \geq 0} (zy - f(y))$.*

Importantly, the structure of the semi-dual problem ensures that both $p_t(x)$ and $p_T(x)$ are involved only through expectation operators, rather than through explicit density evaluations. This enables the use of Monte Carlo methods to approximate all necessary integrals, thereby eliminating the need for access to the density function—particularly for the target domain—when constructing intermediate distributions. Practically, following prior works (Korotin et al., 2023; Choi et al., 2023; 2024), we can parameterize both the dual potential $w$ and the transport map $\boldsymbol{T}$ by neural networks, denoted as $w_\phi$ and $\boldsymbol{T}_\theta$ respectively. The models are trained in an alternating adversarial scheme to learn the sequence of maps $\{\boldsymbol{T}_{\theta,t}\}_{t=0}^{T-1}$, which can be applied to generate intermediate domains progressively.

### 3.3 Robust Training Procedure for Semi-Dual Form Transportation

While Section 3.2 provides a semi-dual form of the gradient flow problem that avoids explicit PDF estimation in target domain, naively training $\mathcal{L}^{SemiDual}$ in Eq. (7) is intrinsically unstable because of its composite 'sup–inf' structure. This instability is not merely algorithmic: the objective itself may be non-identifiable. We formalize this phenomenon by proving that the dual problem can have non-unique optima, as the following theorem shows:

**Proposition 2.** *The semi-dual formulation in Eq. (7) admits non-unique optimal solutions.*

To address this issue, we incorporate an entropy regularization term into the primal objective Eq. (6), which leads to the following proposition:

**Proposition 3.** *Let $\kappa(x_t, x) := p(x_t)\,p_T(x)$ denote the reference joint PDF. The entropy-regularized primal problem is*

$$\mathcal{L}^{E\text{-}Primal} = \underset{\rho \in \mathcal{P}_2(\mathbb{R}^D)}{\arg\min}\; \frac{1}{2\eta}\,\mathcal{W}_2^2(\rho(x), p(x_t)) + \mathbb{D}_f\big[\rho(x), p_T(x)\big]$$

$$+ \epsilon \iint \pi(x_t, x)\,[\log \frac{\pi(x_t, x)}{\kappa(x_t, x)} - 1]\,\mathrm{d}x_t\,\mathrm{d}x, \tag{8}$$

*and is equivalent to the semi-dual optimization problem*

$$\mathcal{L}^{E\text{-}SemiDual} = \sup_{w}\; -\epsilon\,\mathbb{E}_{p(x_t)}[\log \mathbb{E}_{p_T(x)}(\exp(\frac{w(x) - \frac{1}{2\eta}\|x - x_t\|_2^2}{\epsilon}))] \;-\; \mathbb{E}_{p_T(x)}[f^\star(-w(x))], \tag{9}$$

*where $w : \mathbb{R}^D \to \mathbb{R}$ and $f^\star$ are as defined in Proposition 1.*

On this basis, we provide a theoretical guarantee of uniqueness for the semi-dual objective in Eq. (9):

**Proposition 4.** *The semi-dual formulation in Eq. (9) admits a unique optimal solution.*

Notably, as seen in Eq. (9), the semi-dual objective depends solely on the potential $w$. Consequently, we can optimize a single model, which lowers the computational burden. We therefore parameterize $w$ by a neural network $w_\phi$ and carry out the optimization.

Finally, conditioned on the resulting $w_\phi$, we subsequently optimize the transport map $\boldsymbol{T}_\theta(x)$ via the following objective based on Eq. (7):

$$\underset{\theta}{\arg\min}\; \frac{1}{2\eta}\|x_t - \boldsymbol{T}_\theta(x_t)\|_2^2 - w_\phi(\boldsymbol{T}_\theta(x_t)). \tag{10}$$

Notably, we denote our approach as "E-SUOT", as the derivation of $\boldsymbol{T}_\theta$ is grounded in the Entropy-regularized Semi-dual Unbalanced Optimal Transport framework.

### 3.4 OVERALL WORKFLOW FOR E-SUOT

Although Sections 3.2 and 3.3 have presented the E-SUOT framework for intermediate domain generation, they do not provide a unified view of the overall workflow for generating intermediate domains. To address this, we summarize the complete procedure in Algorithm 1 (Due to page limit, the complete algorithm and other detailed information are summarized in Appendix E) and the corresponding illustration is given in Fig. 2. As shown in the algorithm, the construction of $w_\phi$ and $\boldsymbol{T}_\theta$ are performed as separate steps, corresponding to Fig. 2(a), and are illustrated in *Lines 3–6* and *Lines 7–10*, respectively. By iteratively executing the procedure described in *Lines 3–10*, we obtain a sequence of transport maps, $\mathcal{T} = \{\boldsymbol{T}_{\theta,t}\}_{t=0}^{T-1}$, which progressively transport samples from the source domain to the target domain, as we demonstrate in Fig. 2(b).

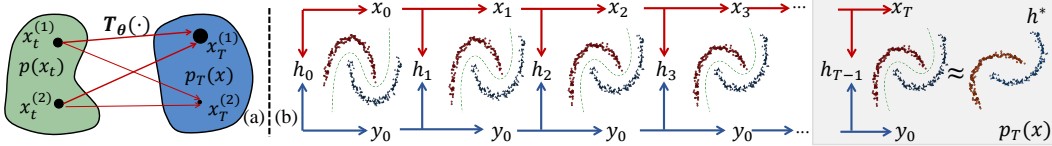

Figure 2: The illustration of the proposed E-SUOT: (a) the unbalanced OT formulation used to solve the transport map $\boldsymbol{T}_\theta(\cdot)$ at time $t$, where thicker arrows and larger points indicate higher mass flows, and (b) the evolution process from the source to the target domain (This figure is conceptually inspired by Zhuang et al. (2024)).

Once the transport map sequence $\mathcal{T} = \{\boldsymbol{T}_{\theta,t}\}_{t=0}^{T-1}$ has been obtained, we proceed to train the classifier $h$ in a stage-wise manner along the transport path. Specifically, at each intermediate step $t$, we first map samples $x_t$ from the current domain to the next intermediate domain $x_{t+1}$ using the corresponding transport map $\boldsymbol{T}_{\theta,t}$. We then update or train the model $h_t$ using the mapped data $x_{t+1}$ as input. By iteratively applying this procedure for $t = 0, \ldots, T-1$, the model is progressively adapted along the sequence of intermediate domains, ultimately bridging the source and target domains.

---

**Algorithm 1** Overall Workflow for Construing E-SUOT-based Intermediate Domain Generation

---

**Input:** Source domain samples: $\{(x_0^{(i)}, y_0^{(i)})\}_{i=1}^{N}$, target domain samples: $\{(x_T^{(i)}, y_T^{(i)})\}_{i=1}^{N}$, entropy regularization strength: $\epsilon$, step size: $\eta$, number of intermediate domain $T-1$, neural network batch size $\mathcal{B}$, and neural network training epochs: $\mathcal{E}$.

**Output:** The set of transportation map: $\mathcal{T} = \{\boldsymbol{T}_{\theta,t}\}_{t=0}^{T-1}$.

1: $\mathcal{T} \leftarrow \varnothing$.
2: **for** $t = 0$ **to** $T - 1$ **do**
3:    **for** $e = 1$ **to** $\mathcal{E}$ **do**
4:      Sample a batch $\{x_t^{(i)}\}_{i=1}^{\mathcal{B}} \sim \{(x_t^{(i)}, y_t^{(i)})\}_{i=1}^{N}$ and $\{x_T^{(i)}\}_{i=1}^{\mathcal{B}} \sim \{(x_T^{(i)}, y_T^{(i)})\}_{i=1}^{N}$.
5:      Update $w_{\phi,t}$ by: $\phi \leftarrow \arg\min_\phi \frac{\epsilon}{\mathcal{B}} \sum_{j=1}^{\mathcal{B}} \log \frac{1}{\mathcal{B}} \sum_{i=1}^{\mathcal{B}} [\exp(\frac{w_{\phi,t}(x_T^{(j)}) - \frac{1}{2\eta}\|x_t^{(j)} - x_T^{(i)}\|_2^2}{\epsilon})] + \frac{1}{\mathcal{B}} \sum_{j=1}^{\mathcal{B}} f^\star(-w_{\phi,t}(x_T^{(j)}))$.
6:    **end for**
7:    **for** $e = 1$ **to** $\mathcal{E}$ **do**
8:      Sample a batch $\{x_t^{(i)}\}_{i=1}^{\mathcal{B}} \sim \{(x_t^{(i)}, y_t^{(i)})\}_{i=1}^{N}$.
9:      Update $\boldsymbol{T}_{\theta,t}$ by: $\theta \leftarrow \arg\min_\theta \frac{1}{\mathcal{B}} \sum_{i=1}^{\mathcal{B}} \frac{1}{2\eta} \|x_t^{(i)} - \boldsymbol{T}_{\theta,t}(x_t^{(i)})\|_2^2 - w_{\phi,t}(\boldsymbol{T}_{\theta,t}(x_t^{(i)}))$.
10:    **end for**
11:    $x_{t+1}^{(i)} \leftarrow \boldsymbol{T}_{\theta,t}(x_t^{(i)}), \forall i \in \{1, \dots, N\}$.
12:    $\mathcal{T} \leftarrow \mathcal{T} \cup \{\boldsymbol{T}_{\theta,t}\}$
13: **end for**

---

### 3.5 THEORETICAL ANALYSIS

Notably, our derivation sidesteps the explicit estimation of the PDF of the target domain by leveraging the semi-dual formulation. This naturally leads to two important questions: (1) Can the proposed E-SUOT framework transport the source domain sufficiently close to the target domain? (2) How does the model perform on the target domain after transport?

To address the first question, we present the following theorem, which quantitatively characterizes the discrepancy between $\rho(x)$ and $p_T(x)$:

**Theorem 5.** *The optimal solution $\rho^*(x)$ to problem defined in Eq. (8) satisfies the following bound:*

$$\mathbb{D}_f[\rho^*(x), p_T(x)] \leq \mathcal{W}_2(p(x_t), p_T(x)). \tag{11}$$

From Theorem 5, we observe that as $t$ increases, the transported PDF $\rho(x)$ progressively becomes similar to $p_T(x)$. Based on this result, we present the following theorem, which provides a theoretical guarantee for the model's performance on the target domain:

**Theorem 6.** *Under mild assumptions, the E-SUOT-based GDA ensures that the target domain generalization error is upper-bounded by the following inequality:*

$$\varepsilon_{p_T}(h_T) \leq \varepsilon_{p_0}(h_0) + \varepsilon_{p_0}(h_T^*) + \iota\zeta\mathcal{C} + \mathcal{S}_{stat}, \tag{12}$$

*where $\iota$ is the Lipschitz constant of the loss function, $\zeta$ is the Lipschitz constant bound for hypotheses in $\mathcal{H}$, $\mathcal{C}$ aggregates the cumulative domain transportation and label continuity costs along the adaptation path, and $\mathcal{S}_{stat}$ is the statistical error term.*

## 4 EXPERIMENTAL RESULTS

### 4.1 EXPERIMENTAL SETUP

**Datasets:** We conduct case studies on four datasets. Specifically, for GDA task, the datasets are "Portraits" (Kumar et al., 2020), "MNIST 45°" and "MNIST 60°" (LeCun, 1998; Deng, 2012). For UDA task, we conduct experiment on the "Office-Home" dataset (Venkateswara et al., 2017). Detailed information about these datasets is given in Appendix E.1.

**Implementation:** Following prior work (Zhuang et al., 2024; Sagawa & Hino, 2025), we employ semi-supervised UMAP to produce low-dimensional embeddings while preserving class discriminability. Unless stated otherwise, we use the KL divergence in the implementation of the E-SUOT. Additional details are available in Appendix E.2.

## 4.2 Baseline Comparison Results

We first compare our proposed approach with several existing GDA-based methods, including Self-training, GST (4 intermediate domains) (Kumar et al., 2020), GOAT (He et al., 2024), CNF (Sagawa & Hino, 2025), and GGF (Zhuang et al., 2024). The detailed information of the experiments are provided in Section E. All baseline models for the GDA task are evaluated with five groups of random seeds, each repeated three times.

Table 1: Baseline comparison on the GDA task with standard deviation and $p$-value.

| Method | Portraits | | | MNIST 45° | | | MNIST 60° | | |
|---|---|---|---|---|---|---|---|---|---|
| | Accuracy (%) | $\Delta$ | $p$-value | Accuracy (%) | $\Delta$ | $p$-value | Accuracy (%) | $\Delta$ | $p$-value |
| Source | 71.2 | - | - | 58.4 | - | - | 36.8 | - | - |
| Self Train | $77.4^*_{\pm 5.02E\text{-}2}$ | ↑8.7% | 1.25E-47 | $58.7^*_{\pm 2.24E\text{-}2}$ | ↑0.5% | 4.99E-50 | $39.9^*_{\pm 2.00E\text{-}2}$ | ↑8.5% | 1.19E-48 |
| GST (4) | $76.1^*_{\pm 6.00E\text{-}2}$ | ↑6.9% | 1.98E-21 | $59.2^*_{\pm 2.45E\text{-}2}$ | ↑1.3% | 1.21E-22 | $39.9^*_{\pm 1.00E\text{-}2}$ | ↑8.5% | 2.14E-23 |
| GOAT | $74.9^*_{\pm 6.21E\text{-}1}$ | ↑5.3% | 2.27E-17 | $65.0^*_{\pm 1.05E\text{-}1}$ | ↑11.3% | 4.07E-20 | $37.2^*_{\pm 8.43E\text{-}2}$ | ↑1.1% | 1.08E-19 |
| CNF | $80.0^*_{\pm 1.85E0}$ | ↑12.4% | 1.79E-15 | $57.6^*_{\pm 1.08E0}$ | ↓1.4% | 4.60E-16 | $41.8^*_{\pm 1.92E0}$ | ↑13.5% | 2.90E-14 |
| GGF | $83.4^*_{\pm 8.79E\text{-}1}$ | ↑17.2% | 9.18E-17 | $57.7^*_{\pm 6.55E\text{-}1}$ | ↓1.2% | 6.15E-17 | $40.8^*_{\pm 8.35E\text{-}1}$ | ↑11.0% | 1.04E-15 |
| E-SUOT | $\mathbf{86.4}^*_{\pm 8.72E\text{-}2}$ | ↑21.5% | 8.88E-21 | $\mathbf{72.1}^*_{\pm 4.62E\text{-}1}$ | ↑23.4% | 1.55E-17 | $\mathbf{51.0}^*_{\pm 5.81E\text{-}1}$ | ↑38.6% | 2.48E-16 |

*Kindly Note*: "*" marks the variants that E-SUOT outperforms significantly at $p$-value $< 0.05$ over paired sample $t$-test. $\Delta$ denotes the performance change relative to the initial classifier. The accuracy is reported as mean (%) $\pm 1.0 \times$ standard deviation error. In addition, **Bolded** results are the best results. Underlined results are the second best results.

As shown in Table 1, our proposed E-SUOT framework consistently outperforms the current state-of-the-art GDA approaches on all evaluated datasets. These results demonstrate the effectiveness and superiority of the E-SUOT framework. In addition, we observe that flow-based methods, such as CNF and GGF, generally achieve top-2 performance on most datasets, highlighting the potential of incorporating flow-based methods in GDA tasks. However, we also note that flow-based methods, occasionally underperform. This observation suggests that flow-based GDA, which requires explicit PDF estimation on target domain, may have inherent limitations, as discussed in Section 3.1.

On this basis, we further evaluate the performance of E-SUOT under the UDA task by comparing it to the DANN (Ganin & Lempitsky, 2015), MSTN (Xie et al., 2018), GVB-GD (Cui et al., 2020), RSDA (Gu et al., 2020; 2022), LAMBDA (Le et al., 2021), SENTRY (Prabhu et al., 2021), FixBi (Na et al., 2021), CST (Liu et al., 2021a), CoVi (Na et al., 2022), and GGF (Zhuang et al., 2024) on the Office-Home dataset (Venkateswara et al., 2017). Detailed information on the experiments is provided in the Section E.2. The results, summarized in Table 2, show that E-SUOT outperforms most existing methods on the majority of UDA tasks and achieves the highest overall average performance. Despite not achieving top performance on every individual task, E-SUOT attains the highest average result across the board, confirming its overall superiority in UDA task.

Table 2: Accuracy (%) comparison on the Office-Home dataset under the UDA setting.

| Method | Ar→Cl | Ar→Pr | Ar→Rw | Cl→Ar | Cl→Pr | Cl→Rw | Pr→Ar | Pr→Cl | Pr→Rw | Rw→Ar | Rw→Cl | Rw→Pr | Avg. |
|---|---|---|---|---|---|---|---|---|---|---|---|---|---|
| DANN | 45.6 | 59.3 | 70.1 | 47.0 | 58.5 | 60.9 | 46.1 | 43.7 | 68.5 | 63.2 | 51.8 | 76.8 | 57.6 |
| MSTN | 49.8 | 70.3 | 76.3 | 60.4 | 68.5 | 69.6 | 61.4 | 48.9 | 75.7 | 70.9 | 55.0 | 81.1 | 65.7 |
| GVB-GD | 57.0 | 74.7 | 79.8 | 64.6 | 74.1 | 74.6 | 65.2 | 55.1 | 81.0 | 74.6 | 59.7 | 84.3 | 70.4 |
| RSDA | 53.2 | 77.7 | 81.3 | 66.4 | 74.0 | 76.5 | 67.9 | 53.0 | 82.0 | 75.8 | 57.8 | 85.4 | 70.9 |
| LAMDA | 57.2 | 78.4 | 82.6 | 66.1 | **80.2** | **81.2** | 65.6 | 55.1 | 82.8 | 71.6 | 59.2 | 83.9 | 72.0 |
| SENTRY | **61.8** | 77.4 | 80.1 | 66.3 | 71.6 | 74.7 | 66.8 | **63.0** | 80.9 | 74.0 | **66.3** | 84.1 | 72.3 |
| FixBi | 58.1 | 77.3 | 80.4 | 67.7 | 79.5 | 78.1 | 65.8 | 57.9 | 81.7 | 76.4 | 62.9 | **86.7** | 72.7 |
| CST | 59.0 | **79.6** | **83.4** | **68.4** | 77.1 | 76.7 | **68.9** | 56.4 | **83.0** | 75.3 | 62.2 | 85.1 | 72.9 |
| CoVi | 58.5 | 78.1 | 80.0 | 68.1 | 80.0 | 77.0 | 66.4 | 60.2 | 82.1 | **76.6** | 63.6 | 86.5 | 73.1 |
| GGF | 59.4 | 75.6 | 81.7 | 67.6 | 77.6 | 78.0 | 67.4 | 61.0 | 82.7 | 75.9 | 62.4 | 85.4 | 72.9 |
| E-SUOT | 61.6 | 79.3 | 81.8 | 67.6 | 77.7 | 78.1 | 67.4 | 61.2 | 82.9 | 76.3 | 62.5 | 85.2 | **73.5** |
| Win Counts | 9 | 9 | 8 | 6 | 7 | 8 | 7 | 9 | 9 | 8 | 7 | 6 | 10 |

*Kindly Note*: **Bolded** and underlined results are the first and second best results, respectively.

### 4.3 INVESTIGATION OF THE UOT FORMULATION

While our main contribution lies in introducing the semi-dual UOT formulation to analyze and improve the flow-based GDA approach, we further investigate "why the UOT-based method performs better than the vanilla OT formulation". To this end, we conduct experiments under label-shift and missing-class scenarios using the Portrait dataset (binary classification task). Specifically, we re-sample the target domain to vary the class prior $p(y = 1)$; when $p(y = 1) = 0.0$ or $p(y = 1) = 1.0$, a missing-class situation is realized. The comparison results between vanilla OT and UOT are reported in Table 3. Here, "E-SOT" denotes entropy-regularized semi-dual optimal transport. From the table, we observe that in the missing-class cases ($p(y = 1) = 0.0$ or $p(y = 1) = 1.0$), the vanilla OT formulation not only fails to improve but even degrades the classifier's performance in the target domain. Moreover, under moderate label shift ($p(y = 1)$ between 0.3 and 0.9), the performance of vanilla OT fluctuates strongly and lacks stability, whereas UOT consistently improves performance. These observations demonstrate that incorporating the unbalanced OT formulation provides a more robust and effective approach for handling domain adaptation under label distribution mismatch.

Table 3: Comparison of vanilla and unbalanced OT formulations for GDA task on Portraits dataset.

| Method | $p(y = 1) = 0.0$ | | $p(y = 1) = 0.1$ | | $p(y = 1) = 0.2$ | | $p(y = 1) = 0.3$ | | $p(y = 1) = 0.4$ | |
|---|---|---|---|---|---|---|---|---|---|---|
| | Accuracy (%) | Δ | Accuracy (%) | Δ | Accuracy (%) | Δ | Accuracy (%) | Δ | Accuracy (%) | Δ |
| Initial | 35.4 | - | 41 | - | 47.1 | - | 53.2 | - | 59.6 | - |
| E-SOT | 55.4$_{\pm3.15E-1}$ | ↑56.41% | 61.1$_{\pm2.01E-1}$ | ↑48.94% | 67.1$_{\pm2.45E-1}$ | ↑42.55% | 56.7$_{\pm2.16E-1}$ | ↑6.54% | 57.7$_{\pm1.66E-1}$ | ↓3.22% |
| E-SUOT | 64.5$_{\pm9.09E-2}$ | ↑82.32% | 78.2$_{\pm6.55E-2}$ | ↑90.50% | 74.5$_{\pm1.02E-1}$ | ↑58.28% | 79.8$_{\pm8.24E-2}$ | ↑49.92% | 77.7$_{\pm3.39E-2}$ | ↑30.42% |

| Method | $p(y = 1) = 0.6$ | | $p(y = 1) = 0.7$ | | $p(y = 1) = 0.8$ | | $p(y = 1) = 0.9$ | | $p(y = 1) = 1.0$ | |
|---|---|---|---|---|---|---|---|---|---|---|
| | Accuracy (%) | Δ | Accuracy (%) | Δ | Accuracy (%) | Δ | Accuracy (%) | Δ | Accuracy (%) | Δ |
| Initial | 73.8 | - | 80 | - | 85.9 | - | 91.4 | - | 97.1 | - |
| E-SOT | 75.2$_{\pm3.08E-2}$ | ↑1.95% | 74.1$_{\pm4.80E-2}$ | ↓7.33% | 80.0$_{\pm2.21E-2}$ | ↓6.89% | 89.9$_{\pm1.92E-2}$ | ↓1.65% | 96.0$_{\pm2.93E-2}$ | ↓1.08% |
| E-SUOT | 79.1$_{\pm2.56E-2}$ | ↑7.24% | 84.0$_{\pm3.46E-2}$ | ↑5.05% | 87.8$_{\pm1.12E-2}$ | ↑2.18% | 91.8$_{\pm1.71E-3}$ | ↑0.45% | 97.6$_{\pm3.23E-3}$ | ↑0.54% |

*Kindly Note*: Δ denotes performance change percentage compared to E-SUOT with entropy regularization and KL divergence. The accuracy is reported as mean (%) $\pm$ 1 × standard deviation error. For source domain, $p(y = 1) = 0.63$. The acronym E-SOT stands for entropy-regularized semi-dual optimal transport.

### 4.4 ABLATION STUDIES

We perform ablation studies from two perspectives: the training strategy for $\boldsymbol{T}_\theta$ and the choice of $f$-divergence. For the *training strategy*, we 1). examine the effect of removing the entropy regularization term—reducing the method to the adversarial training strategy in Eq. (7), and 2). evaluate a barycentric projection approach analogous to flow matching (Lipman et al., 2023), where the transport plan is first estimated and then used to project source samples toward the target, subsequently being refined during training. For the objective *functional*, we study different parameterizations of $f^\star$, such as employing non-decreasing convex functions like 1) `Softplus`, and also compare the 2) $\chi^2$ divergence and the 3) identity function. More detailed information on these experiments' implementation is provided in Appendix E.3. The ablation study results are summarized in Table 4.

Table 4: Ablation study results on GDA setting with standard deviation and $p$-value.

| Dataset | | Portraits | | | MNIST 45° | | | MNIST 60° | | |
|---|---|---|---|---|---|---|---|---|---|---|
| Metric | | Accuracy (%) | Δ | $p$-value | Accuracy (%) | Δ | $p$-value | Accuracy (%) | Δ | $p$-value |
| Training | Adversarial KL | 74.8*$_{\pm3.10E0}$ | ↓13.4% | 1.83E-03 | 52.0*$_{\pm3.60E0}$ | ↓27.8% | 3.05E-04 | 34.9*$_{\pm4.10E0}$ | ↓31.5% | 4.59E-03 |
| | Barycentric KL | 83.9*$_{\pm9.42E-1}$ | ↓3.0% | 4.05E-03 | 62.5*$_{\pm1.06E0}$ | ↓13.3% | 1.77E-04 | 38.3*$_{\pm4.46E0}$ | ↓24.8% | 6.57E-03 |
| Functional | Entropy Softplus | 80.1*$_{\pm3.53E0}$ | ↓7.3% | 2.24E-02 | 59.7*$_{\pm1.14E0}$ | ↓17.2% | 7.08E-05 | 38.2*$_{\pm1.10E0}$ | ↓25.1% | 6.27E-06 |
| | Entropy $\chi^2$ | 79.8$_{\pm6.07E0}$ | ↓7.7% | 9.57E-02 | 60.2*$_{\pm1.43E0}$ | ↓16.5% | 8.03E-05 | 42.4*$_{\pm3.47E0}$ | ↓16.9% | 7.19E-03 |
| | Entropy Identity | 81.2*$_{\pm1.71E0}$ | ↓6.1% | 3.13E-03 | 59.6*$_{\pm1.25E0}$ | ↓17.4% | 8.62E-05 | 39.6*$_{\pm2.34E0}$ | ↓22.3% | 2.59E-04 |
| | Entropy KL | 86.4$_{\pm8.72E-2}$ | - | - | 72.1$_{\pm4.62E-1}$ | - | - | 51.0$_{\pm5.81E-1}$ | - | - |

*Kindly Note*: "*" marks the variants that E-SUOT outperforms significantly at $p$-value $< 0.05$ over paired sample $t$-test. Δ denotes performance change percentage compared to E-SUOT with entropy regularization and KL divergence. The accuracy is reported as mean (%) $\pm$ 1 × standard deviation error.

From Table 4, we find that adversarial training performs the worst, underscoring the importance of entropy regularization for model training in Section 3.3. While barycentric mapping is competitive, it struggles on complex datasets such as MNIST 45° and MNIST 60°, highlighting the need for the semi-dual formulation. Additionally, alternatives to KL divergence—especially `Softplus`—cause significant performance drops, emphasizing the importance of proper divergence selection. We also observe that replacing KL divergence with alternatives such as $\chi^2$ divergence, the identity function, or particularly Softplus results in substantial performance degradation, further illustrating that choosing a suitable discrepancy to drive the evolution of source domain to target domain is critical for promising the performance of GDA.

## 4.5 SENSITIVITY ANALYSIS

From Figs. 3(a) to 3(d), we systematically investigate the sensitivity of our E-SUOT model with respect to key hyperparameters, including batch size $\mathcal{B}$, discretization step size $\eta$, simulation steps $T$, and entropy regularization strength $\epsilon$ on the Portraits and MNIST 45° datasets.

Specifically, as shown in Fig. 3(a), we observe that increasing the batch size $\mathcal{B}$ initially improves model performance; however, after a certain point, further increasing the batch size leads to a performance decline. This pattern suggests that, in the simulation of WGF-based approaches (including ours), careful selection of batch size is crucial: if $\mathcal{B}$ is too small, stochastic sampling noise may dominate and degrade the results; conversely, excessively large $\mathcal{B}$ can cause the model to overfit and diminish its performance. A similar trend is found when varying the discretization step size $\eta$, as illustrated in Fig. 3(b). A small step size may prevent the simulation trajectory from adequately reaching the target distribution within a finite number of steps, limiting learning efficiency. On the other hand, a step size that is too large introduces significant discretization error, which again results in poor model performance. Furthermore, as demonstrated in Fig. 3(c), increasing the number of simulation steps $T$ also produces a non-monotonic effect: beyond a certain threshold, more steps actually undermine performance. This is likely because aligning the feature/target distributions too strictly does not necessarily correspond to optimal performance in the target domain, thus further justifying our introduction of divergence-based regularization to relax strict alignment constraints compared to traditional OT-based methods. Finally, as shown in Fig. 3(d), the entropy regularization parameter $\epsilon$ also significantly influences results. We observe that varying $\epsilon$ can lead to diverse performance outcomes, highlighting the importance of properly investigating and tuning the entropy regularization strength in practical applications. In conclusion, our sensitivity study underscores the importance of carefully selecting the batch size $\mathcal{B}$, step size $\eta$, and end time $T$ for E-SUOT performance, and further indicates that the entropy regularization strength $\epsilon$ is dataset-dependent and thus warrants systematic validation on the target dataset to achieve optimal E-SUOT performance.

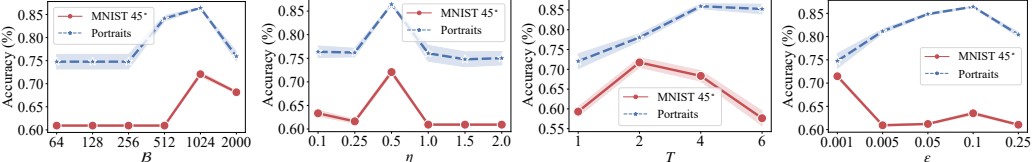

(a) Accuracy (%) along $\mathcal{B}$. (b) Accuracy (%) along $\eta$. (c) Accuracy (%) along $T$. (d) Accuracy (%) along $\epsilon$.

Figure 3: Sensitivity Analysis Results on Portrait and MNIST 45° Datasets.

## 4.6 COMPUTATIONAL TIME COMPARISON

In this subsection, we further analyze the empirical time complexity of the proposed E-SUOT approach in comparison with alternative methods on the GDA task. The computational time results are presented in Fig. 4.

As shown in Fig. 4, the GOAT approach is the most time-consuming on larger datasets, while GGF takes more time on smaller datasets; both consistently rank among the top two in terms of computation cost. This can be attributed to their inherent algorithmic structures: GOAT involves solving the exact optimal transport problem, which becomes computationally prohibitive as the dataset size increases. In contrast, GGF relies on the forward Euler method, which requires a very small step size—and therefore a large number of iterations—to avoid significant simulation errors, resulting in higher computational overhead even on smaller datasets. Notably, the computational time of our proposed E-SUOT remains stable as dataset size grows. This effi-

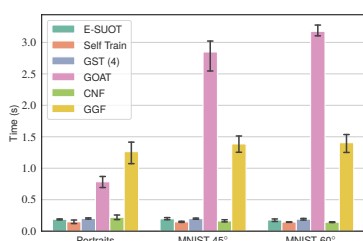

Figure 4: Computational time (s).

ciency stems from directly parameterizing the transport map using a single forward pass through a neural network and the JKO scheme, a variant of backward discretization approach, requiring only a few steps to achieve the desired performance.

## 5 RELATED WORKS

### 5.1 GRADUAL DOMAIN ADAPTATION

GDA seeks to bridge the distributional gap between source and target domains by leveraging a sequence of intermediate domains, thereby enabling more fine-grained adaptation. Early works have explored self-training strategies (Kumar et al., 2020), adversarial objectives (Wang et al., 2020), and provided generalization bounds under gradual distribution shifts (Kumar et al., 2020; Dong et al., 2022; Wang et al., 2022). However, these approaches often depend on the availability of discrete intermediate domains (Chen & Chao, 2021). To address this, optimal transport approaches (Abnar et al., 2021; He et al., 2024) have been leveraged to construct intermediate domains along the Wasserstein geodesic, ensuring minimal distributional discrepancy in the adaptation process. More recently, flow-based GDA has emerged, which explicitly models domain evolution and synthesizes continuous intermediate distributions via parametric flows. For instance, Sagawa & Hino (2025) uses continuous normalizing flows to parameterize domain trajectories as ODEs in the data space, while Zhuang et al. (2024) incorporates label information into this evolution and employs gradient flows to realize the steepest transformation from source to target domain. Nevertheless, flow-based methods still require explicit estimation of the target domain's PDF to guide the evolution, and inaccuracies in this estimation can lead to performance drops in target domain. To address this limitation, we reformulate the flow-based approach from a semi-dual formulation (see Proposition 1), which unifies the flow-based and optimal transport methods. Building on this, we further propose a convergence-guaranteed approach with the help of entropy regularization (Proposition 3) and analyze its generalization error (see Theorem 6). during the evolution of the flow and proposed gradient

### 5.2 SEMI-DUAL FORMULATION OF GRADIENT FLOWS

Gradient flow (Santambrogio, 2017), which seeks to optimize a specified functional in the space of probability measures, has played a critical role in both sampling and optimization algorithm design. For gradient flows induced by $f$-divergences (with the KL divergence being the notable example), such as Langevin sampling (Welling & Teh, 2011), have been extensively explored to generate samples that progressively transition from the source domain toward the target domain. However, these methods typically assume access to an exact (unnormalized) PDF for the target distribution (Liu & Wang, 2016; Liu, 2017), which is often infeasible in practice when only samples are available. To overcome this, several approaches have explored dual formulations of $f$-divergence (Nguyen et al., 2007; 2010), which avoid explicit density estimation for the target domain and instead optimize primal formulation (Korotin et al., 2023; Rout et al., 2022; Fan et al., 2022; Gazdieva et al., 2023; Choi et al., 2023; 2024). These dual-formulation methods, however, generally require adversarial optimization characterized by a composite "sup-inf" structure in order to properly approximate the dual objective when implemented with neural networks (Nowozin et al., 2016; Arjovsky et al., 2017). Our work differs from these approaches in two key aspects. First, we provide a theoretical analysis from the perspective of the non-uniqueness of optimal solutions in Proposition 2, highlighting that such adversarial formulations can suffer from this issue, which may hinder training stability. Building upon this insight, we introduce the entropy regularization that transforms the adversarial game into an alternative paradigm in Proposition 3, and further prove that this regularization ensures the stability via the uniqueness of the optima in Proposition 4 and convergence in Theorem 5.

## 6 CONCLUSIONS

In this paper, we addressed the challenge in flow-based GDA, namely the reliance on explicit estimation of the target domain PDF inherited from traditional $f$-divergence formulations. To overcome this, we reformulated the flow simulation as an optimization problem augmented with a Wasserstein regularization term. Building on this, we derived a novel semi-dual formulation that avoids explicit estimation of the target density. However, we observed that the resulting semi-dual structure introduces instability due to its composite 'sup-inf' structure. To address this, we proposed an entropy regularization term that eliminates the inner inf operator, thereby restoring stability and ensuring uniqueness of the optimal solution. Based on these insights, we developed a new GDA framework called "E-SUOT" and provided theoretical guarantees for its convergence and generalization. Finally, extensive experiments validate the effectiveness and practical advantages of our approach.

## ETHICS STATEMENT

The authors have read and comply with the ICLR Code of Ethics. This research does not involve human subjects or personally identifiable information, and uses only publicly available datasets under their respective licenses. We do not foresee harmful or dual-use implications from the proposed methods. There are no conflicts of interest or undisclosed sponsorship.

## REPRODUCIBILITY STATEMENT

The anonymous downloadable source code is available at: https://anonymous.4open.science/r/E_SUOT_GDA-9240/. For theoretical results, the derivations proof of the claims are included in Appendix C. Based on this, a detailed overall workflow for the proposed E-SUOT is summarized in Appendix D. For datasets used in our experiments, we provide a complete description of the dataset statistics and processing work flow in Appendix E.

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

# Contents for Appendix

# A  NOMENCLATURE

Table A.1: Technical terminology table.

| Symbol | Description | Symbol | Description |
|---|---|---|---|
| $T$ | Intermediate domain number | $\mathscr{A}$ | Upper bound on the $L_2$ norm of the first variation of the KL divergence |
| $\Delta$ | Performance difference | $\mathscr{B}$ | Upper bound on the $L_2$ norm of the gradient of the first variation of the KL divergence |
| $\|\|\|_2$ | $L_2$ norm | $\mathscr{H}_0$ | Light-tail constant |
| $\arg\min$ | Argument of the minimum | $\nabla$ | Gradient operator |
| $\boldsymbol{T}$ | Transportation map | $\omega$ | Parameter of classifier |
| $\boldsymbol{T}^{-1}$ | Inverse transformation of the transportation map | $\phi$ | Parameter of potential function |
| $\delta$ | Variation operator | $\pi$ | Transportation plan |
| $\det$ | Determinant | $\rho^*(x)$ | Optimal PDF |
| $\epsilon$ | Entropy regularization coefficient | $\sup$ | Supremum |
| $\eta$ | Discretization stepsize | $\theta$ | Parameter of transportation map |
| $\exp$ | Exponential function | $\delta$ | Dirac delta mass |
| $\inf$ | Infimum | $\varepsilon$ | Generalization error |
| $\iota$ | Lipschitz constant of the loss function | $\widehat{h}$ | Logit layer of classifier |
| $\kappa(x,y)$ | Reference joint PDF | $\widehat{y}$ | Predicted label |
| $\lambda_1$ | Coefficient of unbalanced optimal transport | $\zeta$ | Lipschitz constant bound for hypotheses in $\mathcal{H}$ |
| $\lambda_2$ | Coefficient of unbalanced optimal transport | $c$ | Cost matrix |
| $\mathbb{D}_f[\rho(x), p_T(x)]$ | $f$-divergence of distribution $q(x)$ with-respect-to distribution $p(x)$ | $f^\star(x)$ | Convex conjugate function of $f(x)$ |
| $\mathbb{D}_{\mathrm{KL}}[\rho(x), p_T(x)]$ | Kullback-Leibler divergence of distribution $q(x)$ with-respect-to distribution $p(x)$ | $h$ | Classifier |
| $\mathbb{E}_q(x)[f(x)]$ | Expectation of function $f(x)$ with-respect-to distribution $q(x)$ | $t$ | Time index |
| $\mathbb{I}$ | indicator function | $u$ | Kantorovich potential function |
| $\mathcal{B}$ | Batch size | $v_t$ | Velocity field |
| $\mathcal{C}$ | Cumulative domain transportation and label continuity costs along the adaptation path | $w$ | Kantorovich potential function |
| $\mathcal{H}$ | Hypothesis class for classifier | $x$ | Input |
| $\mathcal{L}$ | Loss function | $y$ | Label |
| $\mathcal{P}_2(\mathbb{R}^{\mathrm{D}})$ | D-dimensional Wasserstein space | GDA | Gradual domain adaptation |
| $\mathcal{S}_{\mathrm{stat}}$ | Statistical error term. | JKO | Jordan-Kinderlehrer-Otto |
| $\mathcal{T}$ | Set of transportation map | KL divergence | Kullback-Leibler divergence |
| $\mathcal{W}_p$ | $p$-Wasserstein distance | OT | Optimal transport |
| $\mathcal{X}$ | Input space | PDF | Probability density function |
| $\mathcal{Y}$ | Label space | UDA | Unsupervised domain adaptation |
| N | Sample size | UOT | Unbalanced optimal transport |
| d | Differential operator | | |
| softmax | softmax function | | |

# B    MATHEMATICAL BACKGROUND ON OPTIMAL TRANSPORT

We begin by reviewing the relevant background of optimal transport, based on references (Villani et al., 2009; Peyré et al., 2019). Assume continuous variables with densities: source $\rho(x)$ supported on $\mathcal{X}$, target $\xi(y)$ supported on $\mathcal{Y}$, and a cost $c(x, y) \geq 0$. We search for a joint probability density function which is called transport plan $\pi(x, y) \geq 0$ such that:

$$\int \pi(x, y)\, \mathrm{d}y = \rho(x), \tag{B.1a}$$

$$\int \pi(x, y)\, \mathrm{d}x = \xi(y), \tag{B.1b}$$

and minimize expected cost:

$$\inf_{\pi \geq 0} \iint c(x, y)\, \pi(x, y)\, \mathrm{d}y\, \mathrm{d}x, \tag{B.2}$$

where $c(x, y)$ is the cost function, for example, squared Euclidean norm: $c(x, y) = \|x - y\|_2^2$. Notably, when $c(x, y)$ is chosen as the squared Euclidean distance, the resulting optimal transport cost corresponds to the squared Wasserstein-2 distance between the two PDFs.

Introducing potentials $u(x)$ and $w(y)$ as Lagrange multipliers for the marginal constraints, we get:

$$\sup_{u,w} \left[ \int u(x)\, \rho(x)\, \mathrm{d}x + \int w(y)\, \xi(y)\, \mathrm{d}y \right] \quad \text{s.t.} \quad u(x) + w(y) \leq c(x, y) \ \forall x, y. \tag{B.3}$$

Intuitively, $u$ and $w$ are "prices"; the constraint ensures the total price never exceeds the cost function. In addition, $u$ and $w$ are also called "(Kantorovich) potential" in optimal transport.

Based on this, we can eliminate one potential via the $c$-transform as follows:

$$w^c(x) := \inf_y c(x, y) - w(y). \tag{B.4}$$

Based on this, we get the semi-dual formulation of optimal transport problem (Korotin et al., 2021; 2023; Choi et al., 2023; 2024; 2025) which maximizes over one potential:

$$\sup_w \int w^c(x)\, \rho(x)\, \mathrm{d}x + \int w(y)\, \xi(y)\, \mathrm{d}y. \tag{B.5}$$

Notably, when total mass may differ or we allow creation/destruction of mass, we can relax marginal constraints using the $f$-divergence-based penalty terms (Chizat et al., 2018; Zhang et al., 2022). Specifically, we still want to optimize $\pi(x, y) \geq 0$, but we will penalize deviations of the induced marginals $\tilde{\rho}(x) := \int \pi(x, y)\, \mathrm{d}y$ and $\tilde{\xi}(y) := \int \pi(x, y)\, \mathrm{d}x$ from $\rho(x)$ and $\xi(y)$:

$$\min_{\pi \geq 0} \iint c(x, y)\, \pi(x, y)\, \mathrm{d}y\, \mathrm{d}x + \lambda_1 \, \mathbb{D}_f(\tilde{\rho}(x), \rho(x)) + \lambda_2 \, \mathbb{D}_f(\tilde{\xi}(y), \xi(y)), \tag{B.6}$$

where $\mathbb{D}_f(\tilde{\rho}(x), \rho(x)) = \int \rho(x)\, f\big(\frac{\tilde{\rho}(x)}{\rho(x)}\big)\, \mathrm{d}x$ and $\lambda_{1,2} > 0$.

In addition, using the convex conjugate $f^\star$, the dual problem becomes

$$\max_{u,w} - \int \rho(x)\, f_1^\star\big(-u(x)\big)\, \mathrm{d}x - \int \xi(y)\, f_2^\star\big(-w(y)\big)\, \mathrm{d}y \quad \text{s.t.} \quad u(x) + w(y) \leq c(x, y)\, \forall x, y, \tag{B.7}$$

where $f_1$, $f_2$ are the chosen divergences on each side.

Similarly, we can eliminate one potential via the $c$-transform as follows:

$$\max_w \ - \int \rho(x)\, f_1^\star\big(-w^c(x)\big)\, \mathrm{d}x - \int \xi(y)\, f_2^\star\big(-w(y)\big)\, \mathrm{d}y, w^c(x) = \inf_y \{c(x, y) - w(y)\}. \tag{B.8}$$

Based on this, we obtain the semi-dual formulation of the unbalanced optimal transport problem.

# C  THEORETICAL DERIVATION

## C.1  DERIVATION OF EQ. (4)

In this subsection, we want to derive the following equivalent relationship in the main content to uphold the rigor of our manuscript:

$$x_{t+\eta} = x_t - \eta \, \nabla \frac{\delta \mathbb{D}_f[p(x_t), p_T]}{\delta p(x_t)} \Rightarrow p(x_{t+\eta}) = \underset{\rho(x) \in \mathcal{P}_2(\mathbb{R}^D)}{\arg\min} \frac{1}{2\eta} \, \mathcal{W}_2^2(\rho(x), p(x_t)) + \mathbb{D}_f[\rho(x), p_T(x)]. \tag{C.1}$$

Notably, the optimization problem given by the right-hand-side of the abovementioned equation is also called Jordan-Kinderlehrer-Otto canonical form (Jordan et al., 1998; Caluya & Halder, 2020; 2022) or minimum movement scheme (Park et al., 2023). Before conducting the derivation, it is necessary to introduce the definition of Wasserstein distance. The squared 2-Wasserstein distance $\mathcal{W}_2^2$ can be defined by finding a transport map $\boldsymbol{T} : \mathbb{R}^D \to \mathbb{R}^D$ that minimizes the average cost of transporting mass from $\rho(x)$ to $\xi(x)$ as follows:

$$\mathcal{W}_2^2(\rho, \xi) = \underset{\boldsymbol{T}:\boldsymbol{T}_\# \rho(x) = \xi(x)}{\inf} \int \|x - \boldsymbol{T}(x)\|_2^2 \, \rho(x) \mathrm{d}x, \tag{C.2}$$

where $\boldsymbol{T}_\#$ indicates the pushforward measure, and the expression for $\boldsymbol{T}(x)$ is defined as follows:

$$\boldsymbol{T}(x) = x + \eta v_t(x). \tag{C.3}$$

Meanwhile, during the transportation, the differential equation that delineates PDF of the evolution process driven by Eq. (1) is called *continuity equation*, defined as follows:

$$\frac{\partial \rho(x_t)}{\partial t} = -\nabla \cdot [v_t(x_t)\rho(x_t)]. \tag{C.4}$$

Building on Eqs. (C.3) and (C.4), and discretizing the continuity equation in the time domain using the forward Euler scheme (Butcher, 2016; Evans, 2022), we obtain:

$$\rho(x) = \rho(x_t) - \eta \nabla \cdot (\rho(x_t)v_t(x_t)) + \mathcal{O}(\eta^2). \tag{C.5}$$

Taking the functional derivative of $\mathbb{D}_f[\rho(x), p_T(x)]$ with respect to $\rho(x)$, we get:

$$\begin{aligned}
\frac{\mathrm{d}}{\mathrm{d}\eta} \mathbb{D}_f[\rho(x), p_T(x)] &= \frac{\mathrm{d}}{\mathrm{d}\eta} \int p_T(x) f\left(\frac{\rho(x)}{p_T(x)}\right) \, dx \\
&= \int p_T(x) \frac{\mathrm{d}}{\mathrm{d}\eta} f\left(\frac{\rho(x)}{p_T(x)}\right) \, dx \\
&= \int \cancel{p_T(x)} f'\left(\frac{\rho(x)}{p_T(x)}\right) \frac{1}{\cancel{p_T(x)}} \frac{\partial \rho(x)}{\partial \eta} \, dx \\
&\overset{(i)}{=} \int \frac{\delta \mathbb{D}_f}{\delta \rho(x)} \frac{\partial \rho(x)}{\partial \eta} \, dx
\end{aligned} \tag{C.6}$$

Here, step (i) is based on comparing the first variation:

$$\begin{aligned}
\delta \mathbb{D}_f[\rho; \sigma] &= \frac{\mathrm{d}}{\mathrm{d}\varepsilon}\bigg|_{\varepsilon=0} \int p_T(x) f\left(\frac{\rho(x) + \varepsilon\sigma(x)}{p_T(x)}\right) \, dx \\
&= \int p_T(x) f'\left(\frac{\rho(x)}{p_T(x)}\right) \frac{1}{p_T(x)} \sigma(x) \, dx \quad \text{(chain rule, } p_T \text{ fixed)} \\
&= \int f'\left(\frac{\rho(x)}{p_T(x)}\right) \sigma(x) \, dx,
\end{aligned}$$

with the definition of functional derivative:

$$\delta \mathbb{D}_f[\rho; \sigma] = \int \frac{\delta \mathbb{D}_f}{\delta \rho(x)} \sigma(x) \, dx,$$

where $\sigma(x)$ denotes an arbitrary perturbation function. Inserting Eq. (C.5) into Eq. (C.6), we get

$$
\begin{aligned}
\frac{\mathrm{d}}{\mathrm{d}\eta}\,\mathbb{D}_f[\rho(x), p_T(x)] &= \int \frac{\delta \mathbb{D}_f}{\delta \rho(x)}\left[-\nabla \cdot (\rho(x)v_t(x))\right]\mathrm{d}x \\
&= \int \frac{\delta \mathbb{D}_f}{\delta \rho(x)}\left[-v_t^\top(x)\nabla\rho(x) - \rho(x)\nabla \cdot v_t(x)\right]\mathrm{d}x \\
&\overset{(ii)}{=} \int \left(-\nabla \cdot \left[\tfrac{\delta \mathbb{D}_f}{\delta \rho(x)}\rho(x)v_t(x)\right] + \rho(x)\,v_t^\top(x)\,\nabla \tfrac{\delta \mathbb{D}_f}{\delta \rho(x)}\right)\mathrm{d}x \\
&\overset{(iii)}{=} \int \rho(x)\,v_t^\top(x)\,\nabla \frac{\delta \mathbb{D}_f(\rho(x), p_T(x))}{\delta \rho(x)}\,\mathrm{d}x.
\end{aligned}
\tag{C.7}
$$

Step (ii) is based on the chain rule:

$$
\begin{aligned}
\nabla \cdot \left[\tfrac{\delta \mathbb{D}_f}{\delta \rho(x)}\,\rho(x)v_t(x)\right] &= \tfrac{\delta \mathbb{D}_f}{\delta \rho(x)}\,\rho(x)\left[\nabla \cdot v_t(x)\right] \\
&+ \tfrac{\delta \mathbb{D}_f}{\delta \rho(x)}\,v_t^\top(x)\,\nabla\rho(x) \\
&+ \left[\nabla \tfrac{\delta \mathbb{D}_f}{\delta \rho(x)}\right]^\top [\rho(x)v_t(x)].
\end{aligned}
\tag{C.8}
$$

Step (iii) uses a mild regularity assumption (Abraham et al., 2012; Liu et al., 2019; Shi et al., 2022) on $\frac{\delta \mathbb{D}_f}{\delta \rho(x)}\rho(x)v_t(x)$, for example rapid decay as $x \to \infty$, so that

$$
\int -\nabla \cdot \left[\tfrac{\delta \mathbb{D}_f}{\delta \rho(x)}\,\rho(x)v_t(x)\right]\mathrm{d}x = 0.
\tag{C.9}
$$

Consequently, $\mathbb{D}_f[\rho(x), p_T(x)]$ can be expanded as follows when $\eta \to 0$:

$$
\mathbb{D}_f[\rho(x), p_T(x)] = \mathbb{D}_f[p(x_t), p_T(x)] + \eta \int p(x_t)v_t^\top(x_t)\nabla \frac{\delta \mathbb{D}_f[p(x_t), p_T(x)]}{\delta p(x_t)}\mathrm{d}x.
\tag{C.10}
$$

For the squared 2-Wasserstein distance, we get:

$$
\mathcal{W}_2^2(\rho(x), p(x_t)) = \int p(x_t)\|x - \boldsymbol{T}^*(x_t)\|_2^2\mathrm{d}x = \eta^2 \int p(x_t)\|v_t^*(x_t)\|_2^2\mathrm{d}x \le \eta^2 \int p(x_t)\|v_t(x_t)\|_2^2\mathrm{d}x,
\tag{C.11}
$$

where $\boldsymbol{T}^*(x)$ and $v_t^*(x)$ are the optimal transportation map and optimal velocity field. Since $v_t(x)$ is not the optimal velocity filed, we obtain the last inequality. Based on Eqs. (C.10) and (C.11), we finally reach the following result:

$$
\begin{aligned}
&\mathbb{D}_f[\rho(x), p_T(x)] + \frac{1}{2\eta}\mathcal{W}_2^2(\rho(x), p(x_t)) - \mathbb{D}_f[p(x_t), p_T(x)] \\
\le &\cancel{\mathbb{D}_f[\rho(x), p_T(x)]} + \frac{\eta}{2}\mathbb{E}_{p(x_t)}[\|v_t(x_t)\|_2^2] + \eta \int \cdot[p(x_t)v_t^\top(x_t)\nabla\frac{\delta \mathbb{D}_f[p(x_t), p_T(x)]}{\delta p(x_t)}]\mathrm{d}x - \cancel{\mathbb{D}_f[\rho(x), p_T(x)]} \\
\le &\frac{\eta}{2}\underbrace{\mathbb{E}_{p(x_t)}[\|\nabla\frac{\delta \mathbb{D}_f[p(x_t), p_T(x)]}{\delta p(x_t)}]\|_2^2]}_{\ge 0} + \frac{\eta}{2}\mathbb{E}_{p(x_t)}[\|v_t(x_t)\|_2^2] + \eta \int p(x_t)v_t^\top(x_t)\nabla\frac{\delta \mathbb{D}_f[p(x_t), p_T(x)]}{\delta p(x_t)}\mathrm{d}x \\
= &\frac{\eta}{2}\mathbb{E}_{p(x_t)}\{\|v_t(x_t) + \nabla\frac{\delta \mathbb{D}_f[p(x_t), p_T(x)]}{\delta p(x_t)}\|_2^2\}.
\end{aligned}
\tag{C.12}
$$

Consequently, the optimal velocity field that reduces the upper bound of the optimization problem defined by the right-hand-side of Eq. (4) can be given as follows:

$$
v_t^*(x_t) = -\nabla\frac{\delta \mathbb{D}_f[p(x_t), p_T(x)]}{\delta p(x_t)},
\tag{C.13}
$$

which implies that the left-hand side of Eq. (C.1) is a sufficient condition for the optimality of its right-hand side.

## C.2 DERIVATION OF PROPOSITION 1

**Proposition** (1). *Consider the following primal problem:*

$$\mathcal{L}^{Primal} = \underset{\rho(x)\in\mathcal{P}_2(\mathbb{R}^D)}{\arg\min} \frac{1}{2\eta}\mathcal{W}_2^2(\rho(x), p(x_t)) + \mathbb{D}_f[\rho(x), p_T(x)]. \tag{C.14}$$

*This problem is equivalent to the following semi-dual formulation:*

$$\mathcal{L}^{SemiDual} = \sup_w \mathbb{E}_{p(x_t)}\left[\inf_{\boldsymbol{T}}\left(\|\boldsymbol{T}(x_t) - x_t\|_2^2 - w(\boldsymbol{T}(x_t))\right)\right] - \mathbb{E}_{p_T(x)}[f^\star(-w(x))], \tag{C.15}$$

*where $w : \mathbb{R}^D \to \mathbb{R}$ is a measurable continuous function, $\boldsymbol{T} : \mathbb{R}^D \to \mathbb{R}^D$ is the transport map, and $f^\star$ denotes the convex conjugate of $f$, defined as $f^\star(z) := \sup_{y\geq 0}(zy - f(y))$.*

*Proof.* Eq. (C.14) can be reformulated as follows:

$$\inf_{\pi\in\mathbb{R}_+^{D\times D}} \quad \frac{1}{2\eta}\iint\|x_t - x\|_2^2\pi(x_t, x)\mathrm{d}x_t\mathrm{d}x + \int f\left(\frac{\rho(x)}{p_T(x)}\right)p_T(x)\mathrm{d}x, \tag{C.16a}$$

$$\text{s.t.} \quad p(x_t) = \int\pi(x_t, x)\mathrm{d}x, \quad \rho(x) = \int\pi(x_t, x)\mathrm{d}x_t. \tag{C.16b}$$

Based on this, we introduce the Lagrangian multiplier Biegler (2010) $u(x_t)$ and $w(x)$ to handle the equality constraints given by Eq. (C.16b) as follows:

$$\begin{aligned}
\mathcal{L} =& \frac{1}{2\eta}\iint\|x_t - x\|_2^2\pi(x_t, x)\mathrm{d}x_t\mathrm{d}x + \int f\left(\frac{\rho(x)}{p_T(x)}\right)p_T(x)\mathrm{d}x \\
&+ \int u(x_t)[p(x_t) - \int\pi(x_t, x)\mathrm{d}x]\mathrm{d}x_t + \int w(x)[\rho(x) - \int\pi(x_t, x)\mathrm{d}x_t]\mathrm{d}x \\
=& \iint[\frac{1}{2\eta}\|x_t - x\|_2^2 - u(x_t) - w(x)]\pi(x_t, x)\mathrm{d}x_t\mathrm{d}x \\
&+ \int u(x_t)p(x_t)\mathrm{d}x_t + \int w(x)\rho(x) + f\left(\frac{\rho(x)}{p_T(x)}\right)p_T(x)\mathrm{d}x.
\end{aligned} \tag{C.17}$$

On this basis, the dual function can be given as follows due to the linear independent structure of problem defined by Eq. (C.17):

$$\begin{aligned}
g(u, w) =& \inf_{\pi(x_t, x)}\iint\left[\frac{1}{2\eta}\|x_t - x\|_2^2 - u(x_t) - w(x)\right]\pi(x_t, x)\,\mathrm{d}x_t\,\mathrm{d}x \\
&+ \int u(x_t)p(x_t)\,\mathrm{d}x_t + \inf_{\rho(x)}\int\left[w(x)\frac{\rho(x)}{p_T(x)} + f\left(\frac{\rho(x)}{p_T(x)}\right)\right]p_T(x)\,\mathrm{d}x \\
=& \inf_{\pi(x_t, x)}\iint\left[\frac{1}{2\eta}\|x_t - x\|_2^2 - u(x_t) - w(x)\right]\pi(x_t, x)\,\mathrm{d}x_t\,\mathrm{d}x \\
&+ \int u(x_t)p(x_t)\,\mathrm{d}x_t - \int p_T(x)\,f^\star(-w(x))\,\mathrm{d}x.
\end{aligned} \tag{C.18}$$

where the last line uses the Legendre–Fenchel conjugate (Touchette, 2005; Caluya & Halder, 2020). Writing $y(x) = \rho(x)/p_T(x)$ and using separability, we have

$$\begin{aligned}
\inf_{\rho(x)}\int\left[w(x)\frac{\rho(x)}{p_T(x)} + f\left(\frac{\rho(x)}{p_T(x)}\right)\right]p_T(x)\,\mathrm{d}x &= \int\inf_{y\geq 0}\left(w(x)\,y + f(y)\right)p_T(x)\,\mathrm{d}x \\
&= -\int\sup_{y\geq 0}\left((-w(x))\,y - f(y)\right)p_T(x)\,\mathrm{d}x \\
&= -\int p_T(x)\,f^\star\left(-w(x)\right)\mathrm{d}x.
\end{aligned} \tag{C.19}$$

Suppose that $\frac{1}{2\eta}\|x_t - x\|_2^2 - u(x_t) - w(x) < 0$ for some pair $(x_t, x)$. In this case, concentrating all the mass of $\pi(x_t, x)$ at this point drives the Lagrangian in Eq. (C.18) to $-\infty$. To avoid such degenerate solutions, it is necessary to impose the condition $\frac{1}{2\eta}\|x_t - x\|_2^2 - u(x_t) - w(x) \geq 0$ almost everywhere. Consequently, the dual problem can be written as

$$\sup_{u(x_t) + w(x) \leq \frac{1}{2\eta}\|x_t - x\|_2^2 \ \pi\text{-a.e.}} \left\{ \int u(x_t) p(x_t) \, dx_t - \int p_T(x) f^\star(-w(x)) \, dx \right\}. \tag{C.20}$$

Equivalently, introducing the convex indicator function $\ell$, this becomes

$$\sup_{u,w} \left\{ \int u(x_t) p(x_t) \, dx_t - \int p_T(x) f^\star(-w(x)) \, dx - \ell\left( u(x_t) + w(x) \leq \frac{1}{2\eta}\|x_t - x\|_2^2 \right) \right\}. \tag{C.21}$$

Since $f^\star$ is convex, non-decreasing, and differentiable, and because $\|x_t - x\|_2^2 \geq 0$, the choice $u(x_t) \equiv -1$ and $w(x) \equiv -1$ ensures all terms in Eq. (C.21) are finite. By Fenchel–Rockafellar's theorem (Bauschke & Combettes, 2017), strong duality therefore holds. Moreover, by complementary slackness the optimal plan $\pi^*$ assigns zero mass to pairs where $\frac{1}{2\eta}\|x_t - x\|_2^2 - u^*(x_t) - w^*(x) > 0$, implying that $\frac{1}{2\eta}\|x_t - x\|_2^2 = u^*(x_t) + w^*(x)$ $\pi^*$-almost everywhere. Hence,

$$u^*(x_t) = \inf_x \left( \frac{1}{2\eta}\|x_t - x\|^2 - w^*(x) \right).$$

Substituting this into the dual yields the semi-dual formulation

$$\sup_{w(x)} \left\{ \int \inf_x \left[ \frac{1}{2\eta}\|x_t - x\|_2^2 - w(x) \right] p(x_t) \, dx_t - \int p_T(x) f^\star(-w(x)) \, dx \right\}. \tag{C.22}$$

Defining the transport map via the $c$-transform as

$$\boldsymbol{T}^*(x_t) \in \arg\min_x \left( \frac{1}{2\eta}\|x_t - x\|_2^2 - w(x) \right)$$

$$\iff \inf_x \left( \frac{1}{2\eta}\|x_t - x\|_2^2 - w(x) \right) = \frac{1}{2\eta}\|x_t - \boldsymbol{T}^*(x_t)\|_2^2 - w(\boldsymbol{T}^*(x_t)), \tag{C.23}$$

and substituting Eq. (C.23) into Eq. (C.22), we obtain the final semi-dual objective

$$\mathcal{L}^{\text{SemiDual}} = \sup_w \mathbb{E}_{p(x_t)} \left[ \|\frac{1}{2\eta}\boldsymbol{T}^*(x_t) - x_t\|_2^2 - w(\boldsymbol{T}^*(x_t)) \right] - \mathbb{E}_{p_T(x)}[f^\star(-w(x))], \tag{C.24}$$

It should be pointed out that there is no closed-form expression of the optimal $\boldsymbol{T}^*(x_t)$ for each $w(x)$ (Korotin et al., 2023; Choi et al., 2023). Hence, the optimization $\boldsymbol{T}(x_t)$ for each $w(x)$ is required, and we reach the final semi-dual objective as follows based on Eq. (C.24):

$$\mathcal{L}^{\text{SemiDual}} = \sup_w \mathbb{E}_{p(x_t)} \left[ \inf_{\boldsymbol{T}} \left( \frac{1}{2\eta}\|\boldsymbol{T}(x_t) - x_t\|_2^2 - w(\boldsymbol{T}(x_t)) \right) \right] - \mathbb{E}_{p_T(x)}[f^\star(-w(x))].$$

$$\square$$

## C.3 DERIVATION OF PROPOSITION 2

**Proposition** (2). *The semi-dual formulation in Eq. (7) admits non-unique optimal solutions.*

*Proof.* Consider the discrete optimal transport setting with a single source point ($x_t$ in Eq. (7)) and two symmetric target points ($x$ in Eq. (7)). Augment the dual objective with an $f$-divergence term acting only on the target potential $w$, but not on the source potential $u$. Then the dual optimizer is not unique.

Specifically, let:

- **Source space:** $x_t = \{a\}$ with $p(x_t) \approx \delta_a$.

- **Target space:** $x = \{b_1, b_2\}$ with $\rho(x) = \frac{1}{2}\delta_{b_1} + \frac{1}{2}\delta_{b_2}$.

- **Cost constant on pairs:** $\|a - b_1\|_2^2 = \|a - b_2\|_2^2 = K$ for some fixed $K \in \mathbb{R}$.

The dual problem obtained from the primal with an additional term $\int f\left(\frac{\rho(x)}{p_T(x)}\right) p_T(x) \, \mathrm{d}x$ acting only on the target side admits multiple optimal solutions $(u, w)$; in particular, uniqueness fails.

The demonstration process can be summarized as follows:

1) At the beginning, let us recall the feasibility for the multipliers $u$ and $w$:

$$u(a) + w(b_j) \leq \|a - b_j\|_2^2 = K, \quad \forall j \in \{1, 2\}. \tag{C.25}$$

Based on this, we can define a shifted source potential $\tilde{u} := u - K$ and keep $\tilde{w} := w$. Hence, the feasibility in Eq. (C.25) can be given as follows:

$$\tilde{u}(a) + \tilde{w}(b_j) \leq 0, \quad \forall j \in \{1, 2\}, \tag{C.26}$$

where the dual objective differs from the original by a global additive constant (independent of $(\tilde{u}, \tilde{w})$), hence the set of maximizers is unaffected by this normalization. As such, without loss of generality, it suffices to analyze the case $K = 0$. For notational simplicity we drop tildes and write

$$u + w_j \leq 0, \quad \forall j \in \{1, 2\}. \tag{C.27}$$

2) Eliminating $u$ and obtaining a piecewise-linear term Since $p(a) = 1$ and $\rho(b_1) = \rho(b_2) = \frac{1}{2}$, the dual objective function (up to an additive constant) can be reformulated as follows:

$$\max_{u, w_1, w_2} \quad u + \frac{1}{2}w_1 + \frac{1}{2}w_2 - \frac{1}{2}f^\star(-w_1) - \frac{1}{2}f^\star(-w_2), \tag{C.28}$$

subject to $u \leq -w_1$ and $u \leq -w_2$. At optimum the constraint in $u$ is tight, hence we have the following result:

$$u = -\min\{w_1, w_2\}. \tag{C.29}$$

Substituting back yields an equivalent maximization over $(w_1, w_2)$:

$$\Phi(w_1, w_2) := -\min\{w_1, w_2\} + \frac{1}{2}w_1 + \frac{1}{2}w_2 - \frac{1}{2}f^\star(-w_1) - \frac{1}{2}f^\star(-w_2). \tag{C.30}$$

On this basis, we can define the "hinge" (V-shaped) linear part as follows:

$$L(w_1, w_2) := -\min\{w_1, w_2\} + \frac{1}{2}w_1 + \frac{1}{2}w_2 = \begin{cases} \frac{1}{2}(w_2 - w_1), & w_1 \leq w_2, \\ \frac{1}{2}(w_1 - w_2), & w_2 \leq w_1, \end{cases} \tag{C.31}$$

so that $L(w_1, w_2) = \frac{1}{2}|w_1 - w_2|$ and in particular $L(r, r) = 0$ for all $r$.

Consequently, we have:

$$\Phi(w_1, w_2) = \frac{1}{2}|w_1 - w_2| - \frac{1}{2}f^\star(-w_1) - \frac{1}{2}f^\star(-w_2). \tag{C.32}$$

3) Notably, on the diagonal $w_1 = w_2 = r$, we have the following result:

$$\Phi(r, r) = -f^\star(-r). \tag{C.33}$$

Since $f^\star$ is strictly convex, the one-dimensional problem $\max_t \Phi(r, r)$ has a unique maximizer $r^*$. Now let us consider antisymmetric perturbations around the diagonal:

$$w_1 = r^* + \delta, \qquad w_2 = r^* - \delta, \qquad \delta \in \mathbb{R}. \tag{C.34}$$

Then we obtain the following result:

$$\frac{1}{2}|w_1 - w_2| = \frac{1}{2}|2\delta| = |\delta|. \tag{C.35}$$

Using the second-order Taylor expansion of the strictly convex function $f^\star$ about $-r^*$, we have for the following equality for small $|\delta|$:

$$-\frac{1}{2}f^\star(-w_1) - \frac{1}{2}f^\star(-w_2) = -f^\star(-r^*) - \frac{1}{2}\left(f^{\star\prime\prime}(-r^*)\right)\delta^2 + \mathcal{O}(\delta^2). \tag{C.36}$$

Based on this, we get:

$$\Phi(r^* + \delta,\, r^* - \delta) = |\delta| \; - \; \tfrac{1}{2} f^{*\prime\prime}(-r^*)\,\delta^2 \; - \; f^\star(-r^*) + \mathcal{O}(\delta^2). \qquad (\text{C.37})$$

It should be pointed out that, for any sufficiently small but nonzero $\delta$, the linear gain term $|\delta|$ dominates the quadratic penalty term $\tfrac{1}{2} f^{*\prime\prime}(-r^*)\delta^2$, hence

$$\Phi(r^* + \delta,\, r^* - \delta) > \Phi(r^*, r^*).$$

Consequently, the diagonal point $(w_1, w_2) = (r^*, r^*)$ is not uniquely optimal; in fact, there exists a continuum of distinct maximizers in a neighborhood along the antisymmetric direction. The corresponding $u$ is

$$u = -\min\{w_1, w_2\} = \begin{cases} -(r^* - \delta), & \delta \geq 0, \\ -(r^* + \delta), & \delta < 0, \end{cases}$$

yielding distinct optimal triples $(u, w_1, w_2)$ for different $\delta \neq 0$.

4) If the original cost is $\|a - b_j\|_2^2 = K$, recall $u = \tilde{u} + K$. Thus each optimal $(\tilde{u}, w)$ constructed above gives an optimal $(u, w)$ for the original problem by adding $K$ to $u$. As the set of optimal $w$-pairs is already non-singleton, the full optimal dual variable pair $(u, w)$ is non-unique.

In summary, our proof is based on the counter-example mentioned above. Specifically, in the symmetric two-target discrete setting, with the additional $f$-term acting only on the target potential $w$, the dual objective contains a V-shaped hinge $L(w_1, w_2) = \tfrac{1}{2}|w_1 - w_2|$ arising from eliminating $u$. This non-strict component competes with the strictly convex penalty $-\sum_j \rho(b_j)\, f^\star(-w(b_j))$. Along antisymmetric perturbations, the first-order increase from the hinge dominates the second-order decrease from the convex penalty, producing a continuum of maximizers. Hence the optimal dual variable pair is not unique. Consequently, the dual problem defined in Eq. (7) admits non-unique optimal solutions. $\qquad \square$

## C.4 DERIVATION OF PROPOSITION 3

**Proposition** (3). *Let $\kappa(x_t, x) := p(x_t)\, p_T(x)$ denote the reference joint PDF. The entropy-regularized primal problem is*

$$\mathcal{L}^{\text{E-Primal}} = \underset{\rho \in \mathcal{P}_2(\mathbb{R}^D)}{\arg\min} \; \frac{1}{2\eta}\, \mathcal{W}_2^2(\rho(x), p(x_t)) + \mathbb{D}_f\big[\rho(x), p_T(x)\big]$$
$$+ \epsilon \iint \pi(x_t, x)\,[\log \frac{\pi(x_t, x)}{\kappa(x_t, x)} - 1]\, dx_t\, dx, \qquad (\text{C.38})$$

*and is equivalent to the semi-dual optimization problem*

$$\mathcal{L}^{\text{E-SemiDual}} = \sup_{w}\, -\epsilon\, \mathbb{E}_{p(x_t)}[\log \mathbb{E}_{p_T(x)}(\exp(\tfrac{w(x) - \frac{1}{2\eta}\|x - x_t\|_2^2}{\epsilon}))] \; - \mathbb{E}_{p_T(x)}[f^\star(-w(x))], \qquad (\text{C.39})$$

*where $f^\star$ denotes the convex conjugate of $f$.*

*Proof.* Define $c(x_t, x) := \frac{1}{2\eta}\|x_t - x\|_2^2$ as the quadratic transport cost. Introducing Lagrange multipliers $u(x_t): \mathbb{R}^D \to \mathbb{R}$ (for the $x_t$-marginal) and $w(x): \mathbb{R}^D \to \mathbb{R}$ (for the $x$-marginal). The Lagrangian of Eq. (C.38) is

$$\mathcal{L}(\pi, \rho; u, w) = \iint c(x_t, x)\, \pi(x_t, x)\, dx_t\, dx + \epsilon \iint \pi(x_t, x)\Big[\log \frac{\pi(x_t, x)}{\kappa(x_t, x)} - 1\Big] dx_t\, dx$$
$$+ \int f\Big(\frac{\rho(x)}{p_T(x)}\Big) p_T(x)\, dx + \int u(x_t)\Big[p(x_t) - \int \pi(x_t, x)\, dx\Big] dx_t \qquad (\text{C.40})$$
$$+ \int w(x)\Big[\rho(x) - \int \pi(x_t, x)\, dx_t\Big] dx.$$

Grouping $\pi$-, $\rho$- and constant terms yields

$$
\begin{aligned}
\mathcal{L} = \iint & \big(c(x_t, x) - u(x_t) - w(x)\big)\, \pi(x_t, x)\, dx_t\, dx \\
& + \epsilon \iint \pi(x_t, x)\Big[\log\frac{\pi(x_t, x)}{\kappa(x_t, x)} - 1\Big] dx_t\, dx \\
& + \int \Big(w(x)\rho(x) + f\big(\tfrac{\rho(x)}{p_T(x)}\big)p_T(x)\Big) dx + \int u(x_t)p(x_t)\, dx_t.
\end{aligned}
\tag{C.41}
$$

Define $a(x_t, x) := c(x_t, x) - u(x_t) - w(x)$. For each fixed $(x_t, x)$, minimize

$$
\phi(y) := a\, y + \epsilon\Big(y \log \tfrac{y}{\kappa} - y\Big), \qquad y \geq 0.
$$

The first-order condition $a + \epsilon \log(y/\kappa) = 0$ gives

$$
y^\star = \kappa\, e^{-a/\epsilon} = \kappa\, e^{(u+w-c)/\epsilon}.
\tag{C.42}
$$

Substituting back yields

$$
\inf_{y \geq 0} \phi(y) = -\epsilon\, \kappa\, e^{-a/\epsilon} = -\epsilon\, \kappa\, \exp\Big(\tfrac{u+w-c}{\epsilon}\Big).
\tag{C.43}
$$

Hence

$$
\inf_{\pi \geq 0}\Big\{\pi\text{-terms of Eq. (C.41)}\Big\} = -\epsilon \iint \kappa(x_t, x)\, \exp\Big(\tfrac{u(x_t)+w(x)-c(x_t,x)}{\epsilon}\Big)\, dx_t\, dx.
\tag{C.44}
$$

For $\rho$, by Legendre–Fenchel conjugate (Touchette, 2005; Caluya & Halder, 2020), we have:

$$
\inf_{\rho(x) \geq 0}\Big\{w(x)\rho(x) + f\big(\tfrac{\rho(x)}{p_T(x)}\big)p_T(x)\Big\} = -p_T(x)\, f^\star\big(-w(x)\big).
\tag{C.45}
$$

Integrating over $x$ gives

$$
\inf_{\rho} \int [w\, \rho(x_t) + f\big(\tfrac{\rho(x_t)}{p_T(x)}\big)\, p_T(x)] dx = -\int p_T(x)\, f^\star\big(-w(x)\big)\, \mathrm{d}x.
$$

Combining Eq. (C.44) and Eq. (C.45), we obtain

$$
\begin{aligned}
g(u, w) = &-\epsilon \iint \kappa(x_t, x)\, \exp\Big(\tfrac{u(x_t)+w(x)-c(x_t,x)}{\epsilon}\Big) dx_t\, dx \\
& -\int p_T(x)\, f^\star\big(-w(x)\big)\, dx + \int u(x_t)p(x_t)\, dx_t.
\end{aligned}
\tag{C.46}
$$

Using $\kappa = p(x_t) \cdot p_T(x)$, define

$$
A(x_t) := \int \exp\Big(\tfrac{w(x)-c(x_t,x)}{\epsilon}\Big)\, p_T(x)\, \mathrm{d}x.
\tag{C.47}
$$

Then

$$
\iint \kappa\, \exp[\tfrac{(u(x_t) + w(x) - c(x_t, x))}{\epsilon}]\mathrm{d}x_t\mathrm{d}x = \int p(x_t)\, A(x_t)\, e^{\frac{u(x_t)}{\epsilon}}\, \mathrm{d}x_t.
$$

Thus, Eq. (C.46) can be reformulated as follows:

$$
g(u, w) = \int \Big[p(x_t)\, u(x_t) - \epsilon\, p(x_t)\, A(x_t)\, e^{u(x_t)/\epsilon}\Big] \mathrm{d}x_t - \int p_T(x)\, f^\star(-w(x))\, \mathrm{d}x.
\tag{C.48}
$$

For each $x_t$, consider

$$
\psi_{x_t}(u) := p(x_t)\, u - \epsilon\, p(x_t)\, A(x_t)\, e^{\frac{u(x_t)}{\epsilon}}.
$$

The first-order condition

$$
\frac{\mathrm{d}}{\mathrm{d}u}\psi_{x_t}(u) = p(x_t) - p(x_t)A(x_t)e^{\frac{u(x_t)}{\epsilon}} = 0
$$

gives

$$e^{u^{\star}(x_t)/\epsilon} = \frac{1}{A(x_t)} \quad \Longleftrightarrow \quad u^{\star}(x_t) = -\epsilon \log A(x_t). \tag{C.49}$$

Substituting back,

$$\sup_{u} \psi_{x_t}(u) = \epsilon \, p(x_t)\big(-\log A(x_t) - 1\big).$$

Summing over $x_t$ and discarding the constant $-\epsilon \int p(x_t) \, \mathrm{d}x_t = -\epsilon$ (independent of $w(x)$), we obtain the semi-dual

$$\sup_{w} -\epsilon \, \mathbb{E}_{p(x_t)}\big[\log A(x_t)\big] - \mathbb{E}_{p_T(x)}[f^{\star}(-w(x))], \tag{C.50}$$

with $A(x_t)$ defined in Eq. (C.47). $\qquad\qquad\square$

## C.5 DERIVATION OF PROPOSITION 4

**Proposition** (4). *The semi-dual formulation in Eq.* (9) *admits a unique optimal solution.*

*Proof.* Let the entropy-regularized dual objective in Eq. (9) be

$$g(w) = -\epsilon \, \mathbb{E}_{p(x_t)}\{\log \mathbb{E}_{p_T(x)}[\exp(\frac{w(x) - \|x - x_t\|_2^2}{\epsilon})]\} - \mathbb{E}_{p_T(x)}[f^{\star}(-w(x))], \tag{C.51}$$

where $f^{\star}$ is assumed to be strictly convex and proper, and $\epsilon > 0$.

We seek to show that $g(w)$ is a strictly concave functional on an appropriate space of measurable functions $w$, thus its maximizer (if it exists) is unique.

Our proof can be given by the following steps

1) Define for fixed $x_t$:

$$\Phi_\epsilon(w; x_t) := -\epsilon \log \mathbb{E}_{p_T(x)}[\exp(\frac{w(x) - \|x - x_t\|_2^2}{\epsilon})], \tag{C.52}$$

   The mapping $w \mapsto \mathbb{E}_{p_T(x)}[\exp(\frac{w(x) - C(x,x_t)}{\epsilon})]$ is log-convex by Hölder's inequality, and therefore, $w \mapsto \Phi_\epsilon(w; x_t)$ is strictly concave, except in directions where $w$ differs only by an additive constant almost everywhere. Taking the expectation over $x_t$ preserves strict concavity unless $w$ is constant almost everywhere.

2) The term $-\mathbb{E}_x[f^{\star}(-w(x))]$ is strictly concave with respect to $w$ because $f^{\star}$ is strictly convex. Specifically, for any distinct $w_1 \neq w_2$, strict convexity of $f^{\star}$ gives for all $\lambda \in (0, 1)$,

$$-\mathbb{E}_x[f^{\star}(-((1-\lambda)w_1(x) + \lambda w_2(x)))] > -(1-\lambda)\mathbb{E}_x[f^{\star}(-w_1(x))] - \lambda \mathbb{E}_x[f^{\star}(-w_2(x))]$$

   provided $w_1(x) \neq w_2(x)$ on a set of positive measure.

3) Since the sum of a strictly concave function and a concave function is strictly concave, it follows that the full dual objective $g(w)$ is strictly concave on the set of admissible functions.

As a result, $g(w)$ admits at most one maximizer, and the proposition is proved. $\qquad\square$

## C.6 DERIVATION OF THEOREM 5

**Theorem** (5). *The optimal solution $\rho^*(x)$ to problem defined in Eq.* (8) *satisfies the following bound:*

$$\mathbb{D}_f[\rho^*(x), p_T(x)] \leq \mathcal{W}_2(p(x_t), p_T(x)). \tag{C.53}$$

According to the definition of the JKO proximal recursion Eqs. (6) and (8), each update $p(x_{t+1})$ satisfies the following inequality (see Theorem 4.0.4 of reference (Ambrosio et al., 2005)):

$$\frac{1}{2\eta}\mathcal{W}_2^2(p(x_{t+1}), p(x_t)) + \mathbb{D}_f[p(x_{t+1}), p_T(x)] \leq \mathbb{D}_f[p(x_t), p_T(x)],$$

where $\mathbb{D}_f[p(x_t), p_T(x)] \geq 0$ and attains its minimum at the target distribution $p_T(x)$. This inequality implies that the total energy decreases at every step, thereby reducing the Wasserstein distance to $p_T(x)$ when $\mathbb{D}_f[p(x_{t+1}), p_T(x)]$ is geodesically convex:

$$\mathcal{W}_2\big(p(x_{t+1}), p_T(x)\big) \leq \mathcal{W}_2\big(p(x_t), p_T(x)\big) - \Delta_t, \quad \Delta_t \geq 0. \tag{C.54}$$

Hence, as $t$ increases, based on Eqs. (C.53) and (C.54) the upper bound on $\mathbb{D}_f[\rho^*(x_t), p_T(x)]$ is gradually tightened:

$$\mathbb{D}_f[\rho^*(x_{t+1}), p_T(x)] \leq \mathbb{D}_f[\rho^*(x_t), p_T(x)] - \Delta_t,$$

which shows that the transported probability density $\rho(x)$ progressively becomes more similar to the target $p_T(x)$.

*Proof.* To facilitate reading, we define the signal as follows:

$$\mathcal{W}_{2,\epsilon}^2(\rho, \xi) := \inf_{\pi \in \Pi(\rho, \xi)} \iint \|x - y\|_2^2 \pi(x, y)\mathrm{d}x\mathrm{d}y$$
$$+ \epsilon \iint \pi(\rho(x), p(x_t))[\log \pi(\rho(x), p(x_t)) - 1]\mathrm{d}x\mathrm{d}x_t, \tag{C.55}$$

The dual representation of the $f$-divergence based on the Legendre–Fenchel conjugate is:

$$\mathbb{D}_f[\rho(x), p_T(x)] = \sup_{v(x)} \left\{ \mathbb{E}_{\rho(x)}[v(x)] - \mathbb{E}_{p_T(x)}[f^\star(v(x))] \right\}. \tag{C.56}$$

Thus, the problem defined in Eq. (8) can be written as:

$$\inf_{\rho(x)} \mathcal{W}_{2,\epsilon}(\rho(x), p(x_t)) + \sup_{v(x)}\{\mathbb{E}_{\rho(x)}[v(x)] - \mathbb{E}_{p_T(x)}[f^\star(v(x))]\} \tag{C.57}$$

Interchanging $\min_{\rho(x)}, \sup_{v(x)}$ by the convexity-concavity and Sion's theorem (Sion, 1958; Simons, 1995), we obtain the following result:

$$\sup_{v(x)} -\mathbb{E}_{p_T(x)}[f^\star(v(x))] + \inf_{\rho(x)}\{\mathcal{W}_{2,\epsilon}(\rho(x), p(x_t)) + \mathbb{E}_{\rho(x)}[v(x)]\} \tag{C.58}$$

The inner minimization with respect to $\rho(x)$ is precisely the entropic optimal transport problem in the semi-dual form for PDFs $\rho(x)$ and $p(x_t)$:

$$\min_{\rho(x)} \mathcal{W}_{2,\epsilon}(\rho(x), p(x_t)) + \mathbb{E}_{\rho(x)}[v(x)] \tag{C.59}$$

whose optimal value equals

$$\mathbb{E}_{p(x_t)}[-\epsilon \log \int \exp(\frac{v(x) - c(x_t, x)}{\epsilon})\mathrm{d}y]. \tag{C.60}$$

This follows from standard duality in entropic optimal transport.

Plug the expression above into the main problem:

$$\sup_{v(x)} \mathbb{E}_{p(x_t)}[-\epsilon \log \int \exp(\frac{v(x) - c(x_t, x)}{\epsilon})\mathrm{d}y] - \mathbb{E}_{p_T(x)}[f^\star(v(x))]. \tag{C.61}$$

This is the desired semi-dual form.

At optimality, plug in any variation $v = v^* + \delta\psi$ into $g(w)$ and take derivative w.r.t. $\delta$ at 0, then set to zero. The calculation is:

$$0 = \frac{\partial}{\partial \delta}g(v^* + \delta\psi)\Big|_{\delta=0} = \mathbb{E}_{p(x_t)}\left[\frac{\int \psi(x) \exp\left(\frac{v^*(x) - c(x_t, x)}{\epsilon}\right)\mathrm{d}x}{\int \exp\left(\frac{v^*(x) - c(x_t, x)}{\epsilon}\right)\mathrm{d}x}\right] - \mathbb{E}_{p_T(x)}\left[(f^\star)'(v^*(x))\psi(x)\right], \tag{C.62}$$

which for all test functions $\psi(x)$ implies

$$\underbrace{\int p(x_t) \frac{\exp\left(\frac{v^*(x) - c(x_t, x)}{\epsilon}\right)}{\int \exp\left(\frac{v^*(x) - c(x_t, x)}{\epsilon}\right) \mathrm{d}x} \mathrm{d}x_t}_{:= \tilde{p}_T(x)} = p_T(x)(f^\star)'(v^*(x)).$$

That is, the pushforward of $p(x_t)$ under the mapping:

$$\boldsymbol{T}^*(x|x_t) = \frac{\exp\left(\frac{v^*(x) - c(x_t, x)}{\epsilon}\right)}{\int \exp\left(\frac{v^*(x) - c(x_t, x)}{\epsilon}\right) \mathrm{d}x},$$

which indicates that

$$\tilde{p}_T(x) = p_T(x)(f^\star)'(v^*(x)). \tag{C.63}$$

So, $\rho^*(x) = \tilde{p}_T(x)$ is the marginal of the optimal transport $\pi^*$ as claimed.

Since the value of the primal objective at $\rho(x) = p_T(x)$ gives an upper bound:

$$\mathbb{D}_f[\rho^*(x)\|p_T(x)] + \mathcal{W}_{2,\epsilon}(\rho^*(x), p(x_t)) \leq \mathcal{W}_{2,\epsilon}(p_T, p(x_t)). \tag{C.64}$$

So in particular, we get:

$$\mathbb{D}_f[\rho^*(x)\|p_T(x)] \leq \mathcal{W}_{2,\epsilon}(p(x_t), p_T(x)). \tag{C.65}$$

In addition, we notice that the following inequality holds for $\epsilon > 0$:

$$\epsilon \iint \pi(\rho(x), p(x_t))[\log \pi(\rho(x), p(x_t)) - 1]\mathrm{d}x\mathrm{d}x_t \leq 0. \tag{C.66}$$

Plugging Eq. (C.66) into Eq. (C.65), we arrive at the desired result. □

## C.7 DERIVATION OF THEOREM 6

**Theorem** (6). *Under mild assumptions, the E-SUOT-based GDA ensures that the target domain generalization error is upper-bounded by the following inequality:*

$$\varepsilon_{p_T}(h_T) \leq \varepsilon_{p_0}(h_0) + \varepsilon_{p_0}(h_T^*) + \iota\zeta\mathcal{C} + \mathcal{S}_{stat}, \tag{C.67}$$

*where $\iota$ is the Lipschitz constant of the loss function, $\zeta$ is the Lipschitz constant bound for hypotheses in $\mathcal{H}$, $\mathcal{C}$ aggregates the cumulative domain transportation and label continuity costs along the adaptation path, and $\mathcal{S}_{stat}$ is the statistical error term.*

Before formally proving the theorem, we introduce the following assumptions, which are mild and commonly satisfied in practical domain adaptation scenarios:

(A. 1) The loss function $\mathcal{L}(\cdot, y)$ is $\iota$-Lipschitz with respect to its first argument; that is, for any $a, a'$ and fixed $y$, we have:

$$|\mathcal{L}(a, y) - \mathcal{L}(a', y)| \leq \iota|a - a'|. \tag{C.68}$$

(A. 2) Each hypothesis $h \in \mathcal{H}$ is $\zeta$-Lipschitz, i.e., for any $x, x'$, we have:

$$|h(x) - h(x')| \leq \zeta\|x - x'\|. \tag{C.69}$$

(A. 3) The labeling function $q_t$ along the adaptation path is such that $|q_t(x) - q_{t-1}(x)|$ is small for most $x$, to ensure local continuity.

(A. 4) The sequence of domains $(p_0, p_1, \ldots, p_T)$ is induced by E-SUOT-based GDA transport, so that the total cumulative cost $\mathcal{C}$ as defined below is finite.

(A. 5) At every step, empirical risk minimization over sufficient samples ensures a small empirical-to-expected error gap, leading to a statistical error term $\mathcal{S}_{stat}$.

(A. 6) The sample size for each domain is large enough to make $\mathcal{S}_{stat}$ negligible in the asymptotic regime.

Notably, Assumption (A.1) is standard and well-justified for classification tasks employing the categorical cross-entropy loss. More specifically, the gradient of $\mathcal{L}_{\mathrm{CE}}$ with respect to the logits $a$ is bounded as

$$\left|\frac{\partial \mathcal{L}_{\mathrm{CE}}}{\partial a_j}\right| = \left|[\mathrm{softmax}(a)]_j - \mathbb{I}(j = y)\right| \le 1,$$

for any class index $j$, since each softmax component lies within $[0, 1]$, and the indicator function $\mathbb{I}(j = y)$ takes values in $\{0, 1\}$. By the classical Lagrange mean value theorem (see, Theorem 4 in (Thomas et al., 2014)), this bounded-gradient property implies that $\mathcal{L}_{\mathrm{CE}}$ is globally Lipschitz continuous with respect to its first argument on any bounded input domain, with a Lipschitz constant of at most 1. In practice, this assumption can be further facilitated by applying weight normalization or spectral-norm regularization, which help maintain bounded network outputs and thereby make the Lipschitz condition more readily satisfied during optimization.

Furthermore, Assumptions (A.2), (A.5) and (A.6) are standard and generally hold for commonly used loss functions and hypothesis classes. Unless the loss or model is exceptionally non-standard, these can be stated directly with the theorem and do not require additional justification.

Assumption (A.3) holds in cases where the labeling function changes smoothly along the adaptation path. For our construction, since the intermediate domains are generated by incremental, continuous transformations (e.g., gradual style or environmental shifts), the underlying semantics of inputs remain stable, and thus $\mathbb{E}_{p_{t-1}(x)}\left[q_t(x) - q_{t-1}(x)\right]$ is small for every $t$. This situation typically occurs under covariate shift, where only the input distribution evolves while class definitions stay fixed. However, we acknowledge that this assumption may not hold in fine-grained recognition settings or tasks with rapidly changing label semantics, where small variations in features can lead to distinct class assignments. In such cases, the theoretical guarantees derived under Assumption (A.3) would apply only locally, within regions where the labeling function remains approximately smooth. However, the assumption may not hold in tasks with abrupt semantic boundaries—such as fine-grained classification—where visually similar samples can belong to distinct categories. Our analysis therefore applies when the domain evolution does not induce significant concept shift.

As for Assumption (A.4), in our E-SUOT-based GDA, each domain is generated via an iterative unbalanced optimal transport step that progressively reduces the transport cost as we proved in Theorem 6. This guarantees that the cumulative cost $\mathcal{C}$ is finite, as can be bounded analytically. In summary, all the above assumptions are justified in our setting. Based on these assumptions, we now proceed with the formal proof.

*Proof.* Our goal is to bound the target risk $\varepsilon_{p_T}(h_T)$. Consider the telescoping sum along the domain adaptation path:

$$\varepsilon_{p_T}(h_T) = \varepsilon_{p_0}(h_0) + \left[\varepsilon_{p_T}(h_T) - \varepsilon_{p_0}(h_0)\right]. \tag{C.70}$$

To make the recursion explicit, rewrite this as:

$$\varepsilon_{p_T}(h_T) = \varepsilon_{p_0}(h_0) + \sum_{t=1}^{T} \left[\varepsilon_{p_t}(h_t) - \varepsilon_{p_{t-1}}(h_{t-1})\right]. \tag{C.71}$$

For each $t \in \{0, \dots, T-1\}$, we observe that

$$\varepsilon_{p_t}(h_t) - \varepsilon_{p_{t-1}}(h_{t-1})$$
$$= \underbrace{\left[\varepsilon_{p_t}(h_t) - \varepsilon_{p_t}(h_{t-1})\right]}_{\text{optimization error}} + \underbrace{\left[\varepsilon_{p_t}(h_{t-1}) - \varepsilon_{p_{t-1}}(h_{t-1})\right]}_{\text{domain shift term}} + \underbrace{\left[\varepsilon_{p_{t-1}}(h_{t-1}) - \varepsilon_{p_{t-1}}(h_t)\right]}_{\le 0 \text{ by ERM}} \tag{C.72}$$

In practice, the last term is non-positive since 'empirical risk minimization' (Vapnik, 1999; Shalev-Shwartz & Ben-David, 2014; Zhuang et al., 2024) ensures moving toward lower risk, so we can drop it for an upper bound.

By the Lipschitz property of $\mathcal{L}$ and $h$,

$$\left|\varepsilon_{p_t}(h) - \varepsilon_{p_{t-1}}(h)\right| \le \iota\zeta \cdot \mathcal{W}_1(p_{t-1}, p_t). \tag{C.73}$$

Suppose the true label function $q_t$ changes along the path. Following standard analysis, this gives an additional cost due to the label discrepancy:

$$\iota\, \mathbb{E}_{p_t(x)}|q_t(x) - q_{t-1}(x)|. \tag{C.74}$$

Therefore, each step can be bounded by

$$|\varepsilon_{p_t}(h_t) - \varepsilon_{p_{t-1}}(h_{t-1})| \leq \iota\zeta\mathcal{W}_1(p_{t-1}, p_t) + \iota\,\mathbb{E}_{p_t(x)}|f_t(x) - f_{t-1}(x)| + s_t \tag{C.75}$$

where $s_t$ denotes the statistical error at step $t$.

Let

$$\mathcal{C} := \sum_{t=0}^{T-1}\left[\mathcal{W}_1(p_{t-1}, p_t) + \frac{1}{\zeta}\mathbb{E}_{p_t(x)}|q_t(x) - q_{t-1}(x)|\right] \tag{C.76}$$

and

$$\mathcal{S}_{\text{stat}} := \sum_{t=1}^{T-1} s_t. \tag{C.77}$$

Sum these bounds for all $t \in \{0, \ldots, T-1\}$, we get:

$$\sum_{t=0}^{T-1}|\varepsilon_{p_t}(h_t) - \varepsilon_{p_{t-1}}(h_{t-1})| \leq \iota\zeta\mathcal{C} + \mathcal{S}_{\text{stat}}. \tag{C.78}$$

As the final classifier $h_T$ may not be optimally trained with respect to $p_0$, include the approximation gap:

$$\varepsilon_{p_0}(h_0) + \varepsilon_{p_0}(h_T^*) - \varepsilon_{p_0}(h_0) \tag{C.79}$$

where $h_T^*$ is the risk minimizer in $\mathcal{H}$ for $p_0$.

Finally,

$$\varepsilon_{p_T}(h_T) \leq \varepsilon_{p_0}(h_0) + \varepsilon_{p_0}(h_T^*) + \iota\zeta\mathcal{C} + \mathcal{S}_{\text{stat}},$$

as desired. $\qquad\square$

While Theorem 6 provides a clean decomposition, the last two terms are not directly computable from data. To bridge this gap between theory and practice, we attempt to estimate them using the following strategies:

- **Loss Lipschitz constant $\iota$:** According to the analysis of Assumption (A.1), we conclude that $\iota < 1$ with the help of applying weight normalization or spectral-norm regularization.

- **Hypothesis Lipschitz constant $\zeta$.** This bounds how sensitively hypotheses $h \in \mathcal{H}$ react to input perturbations. In our implementation process, we use the multi-layer-perceptron with ReLU activated function. Thus, the Lipschitz constant $\zeta$ for the classfier can be estimated as follows:

$$\zeta \leq \prod_{\ell=1}^{L}\|W_\ell\|_2, \tag{C.80}$$

where $W_\ell$ is the weight of the linear layer, $\|W_\ell\|_2$ is the spectral norm of $W_\ell$. Similarily, $\zeta$ can be controlled by enforcing spectral normalization or weight normalization on layers.

- **Cumulative cost $\mathcal{C}$:** Recall Eq. (C.76), we observe that the 1-Wasserstein distance term $\mathcal{W}_1(p_{t-1}, p_t)$ can be approximated using sample-based optimal transport distances, for example, Sinkhorn distance (Cuturi, 2013), computed on intermediate feature representations. Besides, the inter-step labeling-function shift term $\mathbb{E}_{p_t(x)}|q_t(x) - q_{t-1}(x)|$ in Eq. (C.76) can be approximated in practice via pseudo-labels predicted by the models at steps $t-1$ and $t$, effectively quantifying how much pseudo-labels change along the adaptation path.

- **Statistical error $\mathcal{S}_{\text{stat}}$:** The term $\mathcal{S}_{\text{stat}} = \sum_{t=1}^{T-1} s_t$ collects the statistical deviations between empirical and population risks at each adaptation step. Under mild assumptions, including $G$-Lipschitz losses and norm-constrained neural networks, standard uniform-convergence bounds based on Rademacher complexity yield the following result, adapted from Section 9.4 of Bach (2024):

$$s_t = \mathcal{O}\left(\frac{1}{\sqrt{N}}\right), \tag{C.81}$$

Consequently, for all $t \in \{0, 1, \ldots, T-1\}$, we can estimate $\mathcal{S}_{\text{stat}}$ as follows:

$$\mathcal{S}_{\text{stat}} = \mathcal{O}\left(\frac{T}{\sqrt{N}}\right). \tag{C.82}$$

### C.8 DISCUSSIONS ON THE SELECTION OF STEP SIZE $\eta$

Let us recall Eqs. (B.6) and (6) as follows:

$$\begin{cases} \mathcal{L}^{\text{Primal}} = \arg\min_{\rho(x) \in \mathcal{P}_2(\mathbb{R}^D)} \frac{1}{2\eta} \mathcal{W}_2^2(\rho(x), p(x_t)) + \mathbb{D}_f[\rho(x), p_T(x)], \\ \min_{\pi \geq 0} \iint c(x,y)\,\pi(x,y)\,\mathrm{d}y\,\mathrm{d}x + \lambda_1\,\mathbb{D}_f(\tilde{\rho}(x), \rho(x)) + \lambda_2\,\mathbb{D}_f(\tilde{\xi}(y), \xi(y)). \end{cases} \tag{C.83}$$

Since Eq. (6) is a variant of Eq. (B.6), where $\lambda_1 \equiv 0$. Based on this, we may raise one question: How to select the $\lambda_2$, i.e. $\eta$?

Since our target is decreasing the functional $\mathbb{D}_f[\rho(x), p_T(x)]$ along the simulation process, thus one of the key factor is that the selection of the $\eta$ can decrease the functional $\mathbb{D}_f[\rho(x), p_T(x)]$. Take the KL divergence, the $f$-divergence we consider in the proposed E-SUOT approach, we have the following proposition for selecting the $\eta$:

**Proposition 7.** *Suppose that the $\|\frac{\delta\mathbb{D}_{\text{KL}}[\rho(x), p_T(x)]}{\delta\rho(x)}\| \leq \mathscr{A}$ and $\|\nabla\frac{\delta\mathbb{D}_{\text{KL}}[\rho(x), p_T(x)]}{\delta\rho(x)}\| \leq \mathscr{B}$, there exists a constant $\mathscr{H}_0$ control the tailness of $p_T(x)$, and let $\{\rho_t(x)|t = 1, \ldots, T\}$ denote the sequence of empirical PDF of the intermediate domain generated by the JKO recursion. When $\eta$ satisfies the following condition, the sequence of KL divergence $\{\mathbb{D}_{\text{KL}}[\rho_t(x), p_T(x)]|t = 1, \ldots, T\}$ converges to a finite value as $t \to \infty$:*

$$0 < \eta < \min(\frac{1}{\mathscr{B}}, \frac{\mathscr{H}_0}{\mathscr{A}}). \tag{C.84}$$

Before proposing the proof, we should introduce the light-tailness property on the target distribution $p_T(x)$ in order to ensure the validity of our Taylor expansion and to control higher-order discretization errors during the proof process. Specifically, we say that $p_T(x)$ is light-tailed (Ambrosio et al., 2005; Johnson & Zhang, 2018; 2021) if there exists a universal constant $\mathscr{H}_0 < \infty$ such that

$$\int \|\nabla \log p_T(x)\| \, p_T(x)\mathrm{d}x < \mathscr{H}_0. \tag{C.85}$$

We call $\mathscr{H}_0$ the "light-tail constant" of $p_T(x)$. This condition requires that the expectation (under $p_T(x)$) of the norm of the score function $\nabla p_T(x)$ is finite. Intuitively, this ensures that $p_T(x)$ decays sufficiently rapidly in the tails so that the gradients do not blow up at infinity. On this basis, the proof is articulated as follows:

*Proof.* Suppose at time $t$ and time $t + \eta$, we have:

$$x_{t+\eta} = \boldsymbol{T}(x) := x_t + \eta v_t(x_t). \tag{C.86}$$

We denote the probability distributions, before and after applying Eq. (C.86) as $\rho_t(x)$ and $\rho_{t+\eta}(x)$, respectively. The aim is to Taylor expand the evolution of

$$\mathbb{D}_{\text{KL}}[\rho_{t+\eta}(x), p_T(x)] = \int \rho_{t+\eta}(x) \log \frac{\rho_{t+\eta}(x)}{p_T(x)} \mathrm{d}x, \tag{C.87}$$

with respect to $\eta$ around $\eta = 0$. The new probability distribution, for small $\eta$, can be given as follows according to the Liouville's theorem:

$$\rho_{t+\eta}(x) = \rho_t(\boldsymbol{T}^{-1}(x)) \cdot |\det \boldsymbol{J}_{\boldsymbol{T}^{-1}}(z)| \tag{C.88}$$

where $\boldsymbol{J}_{\boldsymbol{T}^{-1}}(z)$ is the Jacobian matrix of the inverse map, and when $\eta$ is small enough, the inverse function $\boldsymbol{T}^{-1}(x)$ of function $\boldsymbol{T}(x)$ can be given as follows:

$$\boldsymbol{T}^{-1}(x) \approx x - \eta v_t(x). \tag{C.89}$$

Hence, expanding to the first order in $\eta$ can be given as follows:

$$\rho_{t+\eta}(x) \approx \rho_t(x - \eta v_t(x)) [1 - \eta \nabla \cdot \phi(z)] \approx \rho_t(x) - \eta \nabla [\rho_t(x) v_t(x)]. \tag{C.90}$$

Define $F(\eta)$:

$$F(\eta) := \mathbb{D}_{\text{KL}}[\rho_{t+\eta}(x), p_T(x)] = \int \rho_{t+\eta}(x) \log \frac{\rho_{t+\eta}(x)}{p_T(x)} \mathrm{d}x. \tag{C.91}$$

Applying Taylor's expansion at $\eta = 0$, we can obtain the following result:

$$F(\eta) = F(0) + \eta F'(0) + \mathcal{O}(\eta^2). \tag{C.92}$$

Now, we start deriving $F'(0)$. When we take the derivative inside the integral, we have:

$$F'(\eta) = \frac{\mathrm{d}}{\mathrm{d}\eta} \int \rho_{t+\eta}(x) \log \frac{\rho_{t+\eta}(x)}{p_T(x)} \mathrm{d}x = \int \frac{\mathrm{d}}{\mathrm{d}\eta} \rho_{t+\eta}(x) [1 + \log \frac{\rho_{t+\eta}(z)}{p_T(x)}] \mathrm{d}x \tag{C.93}$$

At $\eta = 0$, $\rho_{t+\eta}(x) = \rho_t(x)$:

$$F'(0) = \int \frac{\mathrm{d}}{\mathrm{d}\eta} \rho_{t+\eta}(x) \Big|_{\eta=0} [1 + \log \frac{\rho_t(x)}{p_T(x)}] \mathrm{d}x. \tag{C.94}$$

Now, using the result from the calculus of variations:

$$\frac{\mathrm{d}}{\mathrm{d}\eta} \rho_{t+\eta}(x) \Big|_{\eta=0} = -\nabla \cdot (\rho_t(x) v_t(x)) \tag{C.95}$$

Thus,

$$F'(0) = - \int \nabla \cdot (\rho_t(x) v_t(x)) [1 + \log \frac{\rho_t(x)}{p_T(x)}] \mathrm{d}x. \tag{C.96}$$

Now, use integration by parts:

$$\int -\nabla \cdot [\rho_t(x) v_t(x)] g(x) \mathrm{d}z = \int [\rho_t(x) v_t(x)]^\top \nabla g(x) \mathrm{d}x. \tag{C.97}$$

Set $g(x) = 1 + \log \frac{\rho_t(x)}{p_T(x)}$. Its gradient is:

$$\nabla g(x) = \nabla \log \rho_t(x) - \nabla \log p_T(x). \tag{C.98}$$

Hence,

$$F'(0) = \int [\rho_t(x) v_t(x)]^\top [\nabla \log \rho_t(x) - \nabla \log p_T(x)] \mathrm{d}x = \mathbb{E}_{\rho_t(x)} \left[ v_t^\top(x)(\nabla \log \rho_t(x) - \nabla \log p_T(x)) \right]. \tag{C.99}$$

But with a negative sign because the original derivative is minus divergence:

$$F'(0) = -\mathbb{E}_{\rho_t(x)} \left[ v_t^\top(x)(\nabla \log \rho_t(x) - \nabla \log p_T(x)) \right]. \tag{C.100}$$

Putting all together, we get:

$$\mathbb{D}_{\mathrm{KL}}[\rho_{t+\eta}(x), p_T(x)]$$
$$= \mathbb{D}_{\mathrm{KL}}[\rho_t(x), p_T(x)] - \eta \mathbb{E}_{\rho_t(x)} \left[ v_t^\top(x)(\nabla \log p_T(x) - \nabla \log \rho_t(x)) \right] + \mathcal{O}(\eta^2). \tag{C.101}$$

Notably, the optimal velocity field $v_t^*(x)$ for KL divergence is $-\nabla \frac{\delta \mathbb{D}_{\mathrm{KL}}[\rho_t(x)]}{\delta \rho_t(x)}$. Thus, we have:

$$\mathbb{D}_{\mathrm{KL}}[\rho_{t+\eta}(x), p_T(x)]$$
$$= \mathbb{D}_{\mathrm{KL}}[\rho_t(x), p_T(x)] \underbrace{-\eta \mathbb{E}_{\rho_t(x)} \left[ (\nabla \log p_T(x) - \nabla \log \rho_t(x))^\top v_t^*(x) \right]}_{\leq 0} + \mathcal{O}(\eta^2). \tag{C.102}$$

Since $\|\nabla \frac{\delta \mathbb{D}_{\mathrm{KL}}[\rho(x), p_T(x)]}{\delta \rho(x)}\| \leq \mathscr{B}$, there exists a positive constant $C$ such that:

$$\mathbb{D}_{\mathrm{KL}}[\rho_t(x), p_T(x)] - \eta \mathbb{E}_{\rho_t(x)} \left[ (\nabla \log p_T(x) - \nabla \log \rho_t(x))^\top v_t^*(x) \right] + \mathcal{O}(\eta^2)$$
$$\leq \mathbb{D}_{\mathrm{KL}}[\rho_t(x), p_T(x)] - \eta \mathbb{E}_{\rho_t(x)} \left[ (\nabla \log p_T(x) - \nabla \log \rho_t(x))^\top v_t^*(x) \right] + C\eta^2, \tag{C.103}$$

where constant $C$ satisfies the following condition:

$$C \propto \mathscr{B}^2. \tag{C.104}$$

To avoid $C\eta^2$ dominating the right-hand-side of Eq. (C.103), we should satisfy the following condition:

$$\frac{1}{\eta^2} \gg \mathscr{B}^2 \Rightarrow \eta \ll \frac{1}{\mathscr{B}} \Rightarrow \eta < \frac{1}{\mathscr{B}}. \tag{C.105}$$

According to the log–Sobolev inequality (Ambrosio et al., 2005; Villani et al., 2009), for the target distribution $p_T(x)$ satisfying the curvature condition controlled by $\mathscr{H}_0 > 0$ (i.e., strong

log–concavity or equivalent tailness control), there exists a log–Sobolev constant $\mathscr{H}_0$ such that for any smooth density $\rho_t(x)$,

$$\mathbb{D}_{\mathrm{KL}}[\rho_t(x), p_T(x)] \leq \frac{\mathscr{H}_0}{2} \mathbb{E}_{\rho_t(x)}\left[\|\nabla \log \tfrac{\rho_t(x)}{p_T(x)}\|^2\right]. \tag{C.106}$$

Equivalently, the following lower-bound form holds:

$$\mathbb{E}_{\rho_t(x)}\left[\|v_t^*(x)\|^2\right] \geq \frac{1}{\mathscr{H}_0} \mathbb{D}_{\mathrm{KL}}[\rho_t(x), p_T(x)], \tag{C.107}$$

where we used $v_t^*(x) = \nabla \log p_T(x) - \nabla \log \rho_t(x)$.

When the variational derivative $\frac{\delta \mathbb{D}_{\mathrm{KL}}[\rho_t(x), p_T(x)]}{\delta \rho_t(x)}$ is bounded by $\mathscr{A}$, and the spatial gradient of this functional derivative is bounded by $\mathscr{B}$, the log–Sobolev inequality admits a perturbation-corrected version:

$$\mathbb{E}_{\rho_t(x)}[\|v_t^*(x)\|^2] \geq \frac{1}{\mathscr{H}_0}\{\mathbb{D}_{\mathrm{KL}}[\rho_t(x), p_T(x)] - \frac{\mathscr{A}}{\mathscr{H}_0}\}, \tag{C.108}$$

where the correction term $\frac{\mathscr{A}}{\mathscr{H}_0}$ compensates for the bounded $\|\nabla \frac{\delta \mathbb{D}_{\mathrm{KL}}}{\delta \rho}\|$ and ensures dimensional consistency of the energy inequality. Plugging Eq. (C.108) into Eq. (C.103) gives:

$$\begin{aligned}
&\mathbb{D}_{\mathrm{KL}}[\rho_{t+\eta}(x), p_T(x)] \\
\leq &\mathbb{D}_{\mathrm{KL}}[\rho_t(x), p_T(x)] - \eta \mathbb{E}_{\rho_t}[\|v_t^*(x)\|^2] + C\eta^2 \\
\leq &\mathbb{D}_{\mathrm{KL}}[\rho_t(x), p_T(x)] - \frac{\eta}{\mathscr{H}_0}\{\mathbb{D}_{\mathrm{KL}}[\rho_t(x), p_T(x)] - \frac{\mathscr{A}}{\mathscr{H}_0}\} + C\eta^2,
\end{aligned} \tag{C.109}$$

Rearranging Eq. (C.109), we have:

$$\mathbb{D}_{\mathrm{KL}}[\rho_{t+\eta}(x), p_T(x)] \leq (1 - \tfrac{\eta}{\mathscr{H}_0}) \mathbb{D}_{\mathrm{KL}}[\rho_t(x), p_T(x)] + \tfrac{\eta \mathscr{A}}{\mathscr{H}_0^2} + C\eta^2. \tag{C.110}$$

To promise the iteration will gradually reduce $\mathbb{D}_{\mathrm{KL}}[\rho_t(x), p_T(x)]$, we should require:

$$(1 - \tfrac{\eta}{\mathscr{H}_0}) \in (0, 1) \quad \Rightarrow \quad 0 < \eta < \mathscr{H}_0. \tag{C.111}$$

According to Eq. (C.105), ignoring $C\eta^2$, we obtain the equilibrium point, corresponding to the steady state as $t \to \infty$:

$$\mathbb{D}_{\mathrm{KL}}[\rho_\infty(x), p_T(x)] = \frac{\mathscr{A}}{\mathscr{H}_0}. \tag{C.112}$$

Next, to ensure that each discrete update indeed decreases the KL divergence, we impose

$$\mathbb{D}_{\mathrm{KL}}[\rho_{t+\eta}(x), p_T(x)] < \mathbb{D}_{\mathrm{KL}}[\rho_t(x), p_T(x)]. \tag{C.113}$$

Substituting Eq. (C.110) into Eq. (C.113) yields the following result:

$$-\frac{\eta}{\mathscr{H}_0} \mathbb{D}_{\mathrm{KL}}[\rho_t(x), p_T(x)] + \frac{\eta \mathscr{A}}{\mathscr{H}_0^2} + C\eta^2 < 0. \tag{C.114}$$

Dividing both sides by $\eta > 0$ and rearranging terms gives the following equation:

$$\mathbb{D}_{\mathrm{KL}}[\rho_t(x), p_T(x)] > \frac{\mathscr{A}}{\mathscr{H}_0} + C\eta \mathscr{H}_0. \tag{C.115}$$

In the late stage of GDA task, $\mathbb{D}_{\mathrm{KL}}[\rho_t(x), p_T(x)]$ approaches its equilibrium value $\mathbb{D}_{\mathrm{KL}}[\rho_\infty(x), p_T(x)] = \frac{\mathscr{A}}{\mathscr{H}_0}$, so that:

$$\mathbb{D}_{\mathrm{KL}}[\rho_t, p_T] - \mathbb{D}_{\mathrm{KL}}[\rho_\infty, p_T] \approx \mathbb{D}_{\mathrm{KL}}[\rho_\infty, p_T] = \frac{\mathscr{A}}{\mathscr{H}_0}. \tag{C.116}$$

Hence, the typical contraction strength per update is of order:

$$\frac{\eta}{\mathscr{H}_0}\left(\mathbb{D}_{\mathrm{KL}}[\rho_t, p_T] - \mathbb{D}_{\mathrm{KL}}[\rho_\infty, p_T]\right) \approx \frac{\eta}{\mathscr{H}_0}\frac{\mathscr{A}}{\mathscr{H}_0}. \tag{C.117}$$

To guarantee that the quadratic residual $C\eta^2$ does not dominate the contraction term in Eq. (C.117), we require:

$$C\eta^2 \ll \frac{\eta}{\mathscr{H}_0}\frac{\mathscr{A}}{\mathscr{H}_0} \implies \eta \ll \frac{\mathscr{H}_0}{\mathscr{A}}. \tag{C.118}$$

That is, the discretization step must satisfy the stability condition since $\mathscr{A} > 0$:

$$0 < \eta < \frac{\mathscr{H}_0}{\mathscr{A}} < \mathscr{H}_0, \tag{C.119}$$

which ensures that the numerical update is dominated by the contraction term rather than by the additive bias or high-order error. Based on Eqs. (C.105) and (C.119), we arrive at the desired result.

$\square$

# D    DETAILED ALGORITHM OF THE E-SUOT FRAMEWORK

While Algorithm 1 outlines the general workflow for generating the intermediate domain, it does not specify how E-SUOT can be applied to the GDA task. To bridge this gap, we first present the complete workflow for E-SUOT-based GDA in Algorithm 2.

Building on this foundation, the complete workflow for E-SUOT-based gradual domain adaptation is summarized in Algorithm 2 based on Algorithm 1. Notably, our algorithm decouples the training of the transport function $T_\theta$ from the fine-tuning of the classifier $h_\omega$. This separation allows the intermediate domain to be generated offline and subsequently used for online inference, potentially reducing overall computation time comparable to traditional GDA approaches.

---

**Algorithm 2** Overall Workflow for Construing E-SUOT-based Gradual Domain Adaptation

---

**Input:** Source domain samples: $\{(x_0^{(i)}, y_0^{(i)})\}_{i=1}^{N}$, target domain samples: $\{(x_T^{(i)}, y_T^{(i)})\}_{i=1}^{N}$, entropy regularization strength: $\epsilon$, step size: $\eta$, number of intermediate domain $T-1$, neural network batch size $\mathcal{B}$, and neural network training epochs: $\mathcal{E}$.
**Output:** Classifier in target domain $h_{\omega,T}$.

1: Initialize the classifier $h_{\omega,0}$: $h_{\omega,0} \leftarrow \arg\min_\omega \mathcal{L}_{\mathrm{CE}}(x_0, h_{\omega,t}, y_0)$.
2: Train $\mathcal{T} = \{T_{\theta,t}\}_{t=1}^{T-1}$: $\mathcal{T} \leftarrow$ Algorithm 1.
3: **for** $t = 0$ **to** $T-1$ **do**
4:     Obtain the intermediate domain data $\{(x_{t+1}^{(i)}, y_{t+1}^{(i)})\}$: $x_{t+1}^{(i)} \leftarrow T_{\theta,t}(x_t^{(i)})$ and $y_{t+1}^{(i)} \leftarrow y_t^{(i)}$ for all $i \in \{1, \ldots, N\}$.
5:     Finetune the classifier $h_{\omega,t+1}$: $h_{\omega,t+1} \leftarrow \arg\min_\omega \mathcal{L}_{\mathrm{CE}}(x_{t+1}, h_{\omega,t}, y_{t+1})$.
6: **end for**

---

# E    DETAILED INFORMATION FOR EXPERIMENTS

## E.1    DATASET DESCRIPTIONS

- **Portraits:** Portraits is a binary gender classification dataset comprising 37,921 front-facing portrait images collected between 1905 and 2013. Following the chronological split protocol of (Kumar et al., 2020), we divide the data into a source domain (the earliest 2,000 images), intermediate domains (14,000 images not utilized in this work), and a target domain (the subsequent 2,000 images), similar to the setting in reference (Zhuang et al., 2024).

- **Rotated MNIST:** Rotated MNIST is a variant of the standard MNIST dataset Deng (2012) in which images are rotated to create domain adaptation challenges. As described in He et al. (2024); Kumar et al. (2020), we use 4,000 source images and 4,000 target images, with the target images rotated by $45°$ to $60°$.

- **Office-Home:** Office-Home is a domain adaptation benchmark dataset consisting of approximately 15,500 images categorized into 65 object classes commonly found in office and home environments (Venkateswara et al., 2017). The dataset encompasses four visually distinct domains—*Artistic (Ar), Clipart (Cl), Product (Pr),* and *Real-World (Rw).*

Following common domain adaptation protocols, one domain is selected as the source domain while another serves as the target domain, resulting in a total of 12 domain transfer tasks (e.g., Ar→Rw, Cl→Pr, etc.).

### E.2 EXPERIMENTAL SETTINGS

#### E.2.1 PRELIMINARIES OF SELF-TRAINING METHOD

Self-training is a classical semi-supervised learning strategy that leverages the model's own predictions on unlabeled data to iteratively improve its performance. Given a model $h_\omega$ trained on labeled source data, we use it to generate pseudo-labels for unlabeled samples in a target or auxiliary dataset $\mathcal{D}_{\text{aux}}$. Each unlabeled input $x_i \in \mathcal{D}_{\text{aux}}$ is assigned a pseudo-label $\tilde{y}_i = \text{sign}(h_\omega(x_i))$, indicating a positive or negative prediction. A new model $h_{\omega'}$ is then trained to minimize the empirical loss on this pseudo-labeled dataset:

$$\text{ST}(\theta, \mathcal{D}_{\text{aux}}) = \arg\min_{h'_\omega \in \mathcal{H}} \frac{1}{N_{\text{aux}}} \sum_{x_i \in \mathcal{D}_{\text{aux}}} \mathcal{L}(h'_\omega(x_i), \text{sign}(h_\omega(x_i))), \tag{E.1}$$

where ST is the abbreviation of self-training, $N_{\text{aux}}$ denotes the sample size of auxiliary dataset $\mathcal{D}_{\text{aux}}$. This procedure can be iteratively repeated, replacing $\theta$ with the newly optimized $\theta'$ to refine the pseudo-labels over time.

Intuitively, self-training alternates between

1) Producing pseudo-labels using the current classifier.

2) Retraining the model on these pseudo-labels, thereby progressively refining the decision boundary.

In practice, confidence thresholding or pseudo-label sharpening can be incorporated to reduce noise accumulation and improve stability.

#### E.2.2 TRAINING & EVALUATION PROTOCOLS

For GDA and UDA task, we follow the standard domain adaptation protocol, where model training and hyperparameter tuning are performed using labeled source data merely since the validation on the target domain is infeasible in unsupervised adaptation setting. All results are reported on the target domian dataset without using target labels for validation or early stopping.

For classifier training under this protocol, let $\widehat{h}_{\omega,t}(x_t)$ denote the logits produced by the classifier parameterized by $\omega$ at time step $t$. We define

$$h_{\omega,t}(x_t) = \text{softmax}(\widehat{h}_{\omega,t}(x_t)) \tag{E.2}$$

as the corresponding class-probability vector. Given the ground-truth label $y_t$, the categorical cross-entropy loss is formulated as follows:

$$\mathcal{L}_{\text{CE}}(x_t, \omega, y_t) = -\sum_{i=1}^{\mathcal{B}} y_t^{(i)} \log h_{\omega,t}^{(i)}(x_t) = -\sum_{i=1}^{\mathcal{B}} y_t^{(i)} \log[\text{softmax}(\widehat{h}_{\omega,t}(x_t))^{(i)}]. \tag{E.3}$$

On this basis, we use the classification accuracy (denoted as "Accuracy") as the evaluation metric, defined as the proportion of correctly predicted target samples:

$$\text{Accuracy} = [\frac{1}{N_{\text{tgt}}} \sum_{i=1}^{N_{\text{tgt}}} \mathbb{I}(\hat{y}_i = y_i)] \times 100\%, \tag{E.4}$$

where $N_{\text{tgt}}$ is the number of target test samples, $\hat{y}_i$ denotes the predicted label of the $i$-th sample, $y_i$ is the corresponding ground-truth label, and $\mathbb{I}(\cdot)$ is the indicator function that equals 1 when the condition holds and 0 otherwise. A larger accuracy value corresponds to better adaptation performance.

### E.2.3 GDA TASK

The official implementations of GOAT (He et al., 2024) and CNF (Sagawa & Hino, 2025) are used in our experiments. Additionally, we employ UMAP (McInnes et al., 2018) to reduce the dimensionality of the three GDA datasets to 8 in order to align with the experimental tuple provided by (Zhuang et al., 2024). The experiments are conducted on a workstation equipped with two NVIDIA RTX 4090 GPUs under five different random seeds at least three times. The overall hyper-parameters we use in our GDA task are summarized in Table E.1.

Table E.1: Hyperparameters for E-SUOT on GDA task.

| Datasets | $\eta$ | $\mathcal{B}$ | $\epsilon$ | $T$ |
|---|---|---|---|---|
| Portraits | 0.5 | 1024 | 0.1 | 5 |
| MNIST $45°$ | 0.5 | 1024 | 0.01 | 5 |
| MNIST $60°$ | 0.5 | 2048 | 0.005 | 5 |

In all experiments, we parameterize the classifier $h_\phi$ as a three-layer multi-layer perceptron (MLP) at each step, utilizing ReLU activation functions and a hidden dimension of 100 for each layer. For both $\boldsymbol{T}_\theta$ and $w_\phi$, we employ a two-layer MLP with the SiLU activation function and incorporate a skip connection to enable a residual structure (He et al., 2016). All models are optimized by the Adam optimizer (Kingma & Ba, 2015) with learning rate at 0.0001. For all three GDA datasets, we apply UMAP (McInnes et al., 2018) to reduce their dimensionality to eight. For the classifier $h_\omega$, we use a two-layer MLP with ReLU activations and 128 hidden units. All baseline models are trained on features embedded by the UMAP.

### E.2.4 UDA TASK

For the UDA task, we adopt the Office-Home dataset (Venkateswara et al., 2017) as the benchmark to evaluate the performance of the proposed E-SUOT framework. Following the standard unsupervised domain adaptation (UDA) protocol, model training and hyperparameter tuning are performed solely using the labeled source data, without access to target labels for validation or early stopping. All results are reported on the target domain dataset.

We compare E-SUOT with a diverse set of representative UDA approaches, including DANN (Ganin & Lempitsky, 2015), MSTN (Xie et al., 2018), GVB-GD (Cui et al., 2020), RSDA (Gu et al., 2020; 2022), LAMBDA (Le et al., 2021), SENTRY (Prabhu et al., 2021), FixBi (Na et al., 2021), CST (Liu et al., 2021a), CoVi (Na et al., 2022), and GGF (Zhuang et al., 2024). For baselines including DANN, MSTN, GVB-GD, RSDA, LAMBDA, SENTRY, FixBi, CST, and CoVi, we directly report the publicly available results from their original papers under identical experimental settings (i.e., the same dataset and evaluation protocol). For the GGF method, as the public release of GGF provides only code for GDA task, we re-implemented the missing components according to the paper's description and ran the experiments locally to maintain consistency with the original experimental protocol.

In addition, similar to GGF, our E-SUOT framework builds upon CoVi as the backbone feature extractor. The extracted features are embedded into an eight-dimensional space using UMAP. The classifiers $h_\omega$ for GGF and E-SUOT are implemented as two-layer ReLU MLPs with 256 hidden units. We set the $\eta$, $\mathcal{B}$, $\epsilon$, and $T$ as 0.5, 1024, 0.001, and 4, respectively. Both GGF and E-SUOT models are trained under the same conditions for fair comparison.

### E.3 DETAILED INFORMATION FOR ABLATION STUDIES

For the ablation study, we ablate two module namely the training strategy of $\boldsymbol{T}_\theta$ and the objective functional. The detailed information are elaborated in this part.

For "Training Strategy", the detailed experimental protocols are given as follows:

- **Adversarial Training:** In our adversarial training scheme, we optimize Eq. (7). Building on (Korotin et al., 2021; 2023; Choi et al., 2023; 2024), the training of $\boldsymbol{T}_\theta$ is formulated adversarially, as summarized in Algorithm 3. In *Line 5*, the penalty term $\frac{1}{2\eta}\|x_{t-1} - \boldsymbol{T}_{\theta,t-1}(x_{t-1})\|_2^2$ is omitted since it is constant with respect to $w_{\phi,t-1}$.
- **Barycentric-based Training:** We propose the algorithm for barycentric-based training in Algorithm 4. For barycentric-based training, rather than first compute the transport map, we attempt to compute the optimal transport map $\pi^*$ between $\rho(x_{t-1})$ and $p_T(x)$ as we demonstrate in *Line 4*. Based on this, we make barycentric projection (Courty et al., 2017b; Perrot et al., 2016) using this $\pi^*$ to obtain the proxy points (Liu et al., 2021b; 2023)

for transport map learning as we demonstrate in *Line 5*. Finally, the transport map $\boldsymbol{T}_{\theta,t-1}$ is constructed based on these points, similar to the flow matching (Lipman et al., 2023), as we demonstrate in *Line 6*.

---

**Algorithm 3** Adversarial Training for $\{\boldsymbol{T}_{\theta,t}\}_{t=1}^{T-1}$.

---

**Input:** Intermediate domain samples: $\{(x_{t-1}^{(i)}, y_{t-1}^{(i)})\}_{i=1}^{N}$ for all $t \in \{1, \dots, T\}$, target domain samples: $\{(x_T^{(i)}, y_T^{(i)})\}_{i=1}^{N}$, entropy regularization strength: $\epsilon$, step size: $\eta$, neural network batch size $\mathcal{B}$, and neural network training epochs: $\mathcal{E}$.
**Output:** The transportation map at $t - 1$: $\boldsymbol{T}_{\theta,t-1}$.

1: Initialize
2: **for** $e = 1$ **to** $\mathcal{E}$ **do**
3:      Sample a batch $\{x_{t-1}^{(i)}\}_{i=1}^{\mathcal{B}} \sim \{(x_{t-1}^{(i)}, y_{t-1}^{(i)})\}_{i=1}^{N}$ and $\{x_T^{(i)}\}_{i=1}^{\mathcal{B}} \sim \{(x_T^{(i)}, y_T^{(i)})\}_{i=1}^{N}$.
4:      Update $w_{\phi,t-1}$ by: $\phi \leftarrow \arg\min_\phi \frac{1}{\mathcal{B}} \sum_{i=1}^{\mathcal{B}} -\frac{1}{2\eta} \|x_{t-1} - \boldsymbol{T}_{\theta,t-1}(x_{t-1})\|_2^2 + w_{\phi,t-1}(\boldsymbol{T}_\theta(x_t^{(i)})) + \frac{1}{\mathcal{B}} \sum_{j=1}^{\mathcal{B}} f^\star(-w_{\phi,t-1}(x_T^{(j)}))$.
5:      Sample a batch $\{x_{t-1}^{(i)}\}_{i=1}^{\mathcal{B}} \sim \{(x_{t-1}^{(i)}, y_{t-1}^{(i)})\}_{i=1}^{N}$.
6:      Update $\boldsymbol{T}_{\theta,t-1}$ by: $\theta \leftarrow \arg\min_\theta \frac{1}{\mathcal{B}} \sum_{i=1}^{\mathcal{B}} \frac{1}{2\eta} \|x_{t-1}^{(i)} - \boldsymbol{T}_{\theta,t-1}(x_{t-1}^{(i)})\|_2^2 - w_{\phi,t-1}(\boldsymbol{T}_{\theta,t-1}(x_{t-1}^{(i)}))$.
7: **end for**

---

**Algorithm 4** Barycentric-based training for $\{\boldsymbol{T}_{\theta,t}\}_{t=1}^{T-1}$.

---

**Input:** Intermediate domain samples: $\{(x_{t-1}^{(i)}, y_{t-1}^{(i)})\}_{i=1}^{N}$ for all $t \in \{1, \dots, T\}$, target domain samples: $\{(x_T^{(i)}, y_T^{(i)})\}_{i=1}^{N}$, entropy regularization strength: $\epsilon$, step size: $\eta$, neural network batch size $\mathcal{B}$, and neural network training epochs: $\mathcal{E}$.
**Output:** The transportation map at $t - 1$: $\boldsymbol{T}_{\theta,t-1}$.

1: Initialize
2: **for** $e = 1$ **to** $\mathcal{E}$ **do**
3:      Sample a batch $\{x_{t-1}^{(i)}\}_{i=1}^{\mathcal{B}} \sim \{(x_{t-1}^{(i)}, y_{t-1}^{(i)})\}_{i=1}^{N}$ and $\{x_T^{(i)}\}_{i=1}^{\mathcal{B}} \sim \{(x_T^{(i)}, y_T^{(i)})\}_{i=1}^{N}$.
4:      Obtain the optimal transport map $\pi^*(x_{t-1}, x_T)$ by: $\pi^*(x_{t-1}, x_T) \leftarrow \inf_\pi \frac{1}{2\eta} \mathcal{W}_2^2(\rho(x_{t-1}), p_T(x)) + \epsilon \iint \pi(x_{t-1}, x_T)[\log \pi(x_{t-1}, x_T) - 1]\mathrm{d}x_{t-1}\mathrm{d}x_T + \mathbb{D}_f[\rho(x_{t-1}), p_T(x)]$.
5:      Obtain the projected samples $\tilde{x}_t$ via $\pi^*(x_{t-1}, x_T)$: $\tilde{x}_t = x_{t-1}\pi^*(x_{t-1}, x_T)$:
6:      Update $\boldsymbol{T}_{\theta,t-1}$ by: $\theta \leftarrow \frac{1}{\mathcal{B}} \sum_{i=1}^{\mathcal{B}} \|\tilde{x}_t^{(i)} - \boldsymbol{T}_{\theta,t-1}(x_{t-1}^{(i)})\|_2^2$
7: **end for**

---

For "Objective Functional", the detailed experimental protocols are given as follows:

- $\chi^2$ **Divergence:** The expression for $\chi^2$ divergence can be given as follows:

$$\mathbb{D}_{\chi^2}[\rho(x_t), p_T(x)] \int p_T(x)[\frac{\rho(x_t)}{p_T(x)} - 1]^2 \mathrm{d}x_t, \quad \text{where} \quad f(x) = (x - 1)^2. \quad \text{(E.5)}$$

  Based on this, the corresponding conjugate function $f^\star$ can be given as follows:

$$f^\star(x) = \begin{cases} \frac{1}{4}x^2 + x, & \text{if } x \geq -2 \\ -1, & \text{if } x < -2 \end{cases}. \quad \text{(E.6)}$$

- **Identity:** For the identity function, we remove the $f$-divergence-based regularization term during the construction of E-SUOT framework. Based on this, the training objective for $w_\phi$ is reformulated as follows:

$$\mathcal{L}_{\text{Identity}}^{\text{E-SemiDual}} = \sup_w -\epsilon \mathbb{E}_{p(x_t)}\{\log \mathbb{E}_{p_T(x)}[\exp(\frac{w(x) - \frac{1}{2\eta}\|x - x_t\|_2^2}{\epsilon})]\} + \mathbb{E}_{p_T(x)}[w(x)], \quad \text{(E.7)}$$

- **Softplus:** We directly parameterize the $f^\star(x)$ using the smooth, convex, and non-decreasing softplus function as follows:

$$f^\star(x) = \log(1 + \exp(x)). \tag{E.8}$$

# F  ADDITIONAL EXPERIMENTAL RESULTS

## F.1  IMPACT OF EARLY TRANSPORT STEPS ON ADAPTATION PERFORMANCE

The multi-step structure of E-SUOT involves a series of learned transport maps $\{T_{\theta,0}, T_{\theta,1}, \ldots\}$, where each step aims to progressively align intermediate feature distributions between domains. While this design promotes smooth domain alignment, it also raises an important question: how sensitive the overall adaptation performance is to the quality of early transport maps, and whether suboptimal early mappings introduce cumulative errors that affect subsequent steps.

To investigate this, we conduct an experiment on the Portraits dataset, where we selectively disable the training of certain transport maps to simulate incomplete or inaccurate early-stage optimization. Two complementary training strategies are designed:

- **Forward strategy:** Progressively remove the training of early transport maps $(T_{\theta,0}, T_{\theta,1}, \ldots)$ while keeping the later ones active, thereby testing whether missing early steps hinder later GDA performance.
- **Backward strategy:** progressively disable the training of later maps $(T_{\theta,4}, T_{\theta,3}, \ldots)$ while retaining the trained early steps, examining whether well-trained initial stages are sufficient to sustain strong performance.

Table F.1 summarizes the results. We observe that in the "Forward" direction, excluding early transport steps causes a substantial accuracy drop (up to 17.7%), indicating that early-stage mappings are crucial for forming a reliable transport foundation. In contrast, the "Backward" experiments show that once these early steps are properly optimized, subsequent refinements yield consistent performance gains, suggesting that later transport maps mainly provide fine adjustments on top of an already well-aligned feature space.

Table F.1: Performance of the E-SUOT vary different training stage on the Portraits dataset.

| Direction | | | Forward | | | | | | | Backward | | | | |
|---|---|---|---|---|---|---|---|---|---|---|---|---|---|---|
| Time Index | $t=0$ | $t=1$ | $t=2$ | $t=3$ | $t=4$ | Accuracy (%) | $\Delta$ | $t=0$ | $t=1$ | $t=2$ | $t=3$ | $t=4$ | Accuracy (%) | $\Delta$ |
| | ✗ | ✓ | ✓ | ✓ | ✓ | $76.4_{\pm 2.06\text{E-}2}$ | ↑7.2% | ✓ | ✗ | ✗ | ✗ | ✗ | $81.5_{\pm 1.70\text{E-}2}$ | ↑14.4% |
| Training | ✗ | ✗ | ✓ | ✓ | ✓ | $75.0_{\pm 2.48\text{E-}2}$ | ↑5.3% | ✓ | ✓ | ✗ | ✗ | ✗ | $83.2_{\pm 1.13\text{E-}2}$ | ↑16.8% |
| Status | ✗ | ✗ | ✗ | ✓ | ✓ | $74.4_{\pm 1.92\text{E-}2}$ | ↑4.4% | ✓ | ✓ | ✓ | ✗ | ✗ | $83.9_{\pm 8.10\text{E-}3}$ | ↑17.8% |
| | ✗ | ✗ | ✗ | ✗ | ✓ | $74.0_{\pm 1.64\text{E-}2}$ | ↑3.9% | ✓ | ✓ | ✓ | ✓ | ✗ | $84.0_{\pm 6.56\text{E-}3}$ | ↑17.9% |
| | ✗ | ✗ | ✗ | ✗ | ✗ | $58.6_{\pm 1.87\text{E-}2}$ | ↓17.7% | ✓ | ✓ | ✓ | ✓ | ✓ | $86.4_{\pm 8.72\text{E-}2}$ | ↑21.5% |

*Kindly Note*: $\Delta$ denotes performance change percentage of the initial classifier accuracy.

Overall, the results highlight that the early transport steps are the key drivers of successful adaptation. When the early mappings are well trained, the rest of the chain benefits from a stabilized feature representation, leading to larger and more consistent improvements. This behavior parallels diffusion-like processes, where the early transport transformations largely determine the shape and quality of the final distribution, as also illustrated in Fig. 3 in reference (Caluya & Halder, 2020) and Fig. 1 in reference (Liu & Wang, 2016).

## F.2  PERFORMANCE IMPROVEMENT VARYING DIFFERENT UDA FEATURE EXTRACTORS

In addition to the results reported in Table 2, we further assess the effectiveness of the proposed E-SUOT framework by integrating it with four representative UDA backbones—MSTN, RSDA, FixBi, and CoVi—on the Office-Home dataset. To facilitate a fair and consistent comparison, we apply UMAP-based dimensionality reduction to obtain an 8-dimensional embedding. The corresponding results are summarized in Table F.2.

Table F.2: Accuracy (%) improvement over different UDA feature extractors.

| Method | Ar→Cl | Ar→Pr | Ar→Rw | Cl→Ar | Cl→Pr | Cl→Rw | Pr→Ar | Pr→Cl | Pr→Rw | Rw→Ar | Rw→Cl | Rw→Pr | Avg. |
|---|---|---|---|---|---|---|---|---|---|---|---|---|---|
| MSTN | 49.8 | 70.3 | 76.3 | 60.4 | 68.5 | 69.6 | 61.4 | 48.9 | 75.7 | 70.9 | 55 | 81.1 | 65.7 |
| E-SUOT+MSTN | 57.8 | 75.9 | 79.6 | 65.5 | 75.9 | 74.8 | 64.5 | 58.5 | 81.4 | 73.7 | 59.5 | 84.4 | 71 |
| Δ | ↑16.1% | ↑8.0% | ↑4.3% | ↑8.4% | ↑10.8% | ↑7.5% | ↑5.0% | ↑19.6% | ↑7.5% | ↑3.9% | ↑8.2% | ↑4.1% | ↑8.1% |
| RSDA | 53.2 | 77.7 | 81.3 | 66.4 | 74 | 76.5 | 67.9 | 53 | 82 | 75.8 | 57.8 | 85.4 | 70.9 |
| E-SUOT+RSDA | 61.5 | 78.8 | 81.7 | 67.6 | 77.3 | 77.6 | 67.2 | 61 | 82.7 | 76 | 62.4 | 85.3 | 73.3 |
| Δ | ↑15.7% | ↑1.4% | ↑0.5% | ↑1.8% | ↑4.4% | ↑1.5% | ↓1.0% | ↑15.2% | ↑0.9% | ↑0.3% | ↑7.9% | ↓0.1% | ↑3.3% |
| FixBi | 58.1 | 77.3 | 80.4 | 67.7 | 79.5 | 78.1 | 65.8 | 57.9 | 81.7 | 76.4 | 62.9 | 86.7 | 72.7 |
| E-SUOT+FixBi | 61.7 | 79.1 | 81.7 | 67.6 | 77.6 | 78.2 | 67.3 | 61.3 | 82.7 | 76 | 62.5 | 85.3 | 73.4 |
| Δ | ↑6.2% | ↑2.3% | ↑1.6% | ↓0.1% | ↓2.4% | ↑0.1% | ↑2.3% | ↑5.9% | ↑1.2% | ↓0.5% | ↓0.6% | ↓1.6% | ↑1.0% |
| CoVi | 58.5 | 78.1 | 80 | 68.1 | 80 | 77 | 66.4 | 60.2 | 82.1 | 76.6 | 63.6 | 86.5 | 73.1 |
| E-SUOT+CoVi | 61.6 | 79.3 | 81.8 | 67.6 | 77.7 | 78.1 | 67.4 | 61.2 | 82.9 | 76.3 | 62.5 | 85.2 | 73.5 |
| Δ | ↑5.3% | ↑1.5% | ↑2.2% | ↓0.7% | ↓2.9% | ↑1.4% | ↑1.5% | ↑1.7% | ↑1.0% | ↓0.4% | ↓1.7% | ↓1.5% | ↑0.5% |

*Kindly Note*: Δ denotes performance change percentage of the vanilla UDA feature extractor compared to E-SUOT.

As shown in Table F.2, E-SUOT consistently enhances the performance of all baseline methods, achieving an average improvement of 8.1%, 3.3%, 1.0%, and 0.5% for MSTN, RSDA, FixBi, and CoVi, respectively. The largest relative gain (*+16.1%*) is observed on the challenging "Ar→Cl" transfer, indicating that E-SUOT is particularly effective when the domain gap is large. Even when combined with more recent and competitive adaptation approaches (e.g., FixBi, CoVi), E-SUOT still yields moderate yet consistent improvements, demonstrating its strong complementary capability rather than competing nature. Overall, the consistent gains across different backbones highlight the adaptability, scalability, and robustness of the E-SUOT framework in handling UDA task.

### F.3 INVESTIGATION ON THE UMAP'S DIMENSION

We further conduct experiments on the sensitivity of the UMAP dimension to investigate the performance of E-SUOT under different input dimensions. Specifically, we vary the UMAP's reduction dimension and investigate the model performance. Since the backbone dimension is 256, setting the dimension to 256 indicates that we did not use the UMAP embedding feature.

From Fig. F.1, we observe that as the embedding dimension changes, the classification accuracy tends to fluctuate within a relatively small range across all target domains. In detail, for each target domain, the performance remains quite stable as we vary the UMAP dimension from 4 to 256—no drastic drops are observed. In some domains (e.g., Ar in Fig. F.1(a), Pr in Fig. F.1(c) and Rw in Fig. F.1(d)), the accuracy curves are almost flat, indicating the proposed method is insensitive to UMAP dimension choices in these cases. For domain Cl in Fig. F.1(b), although the standard deviation is higher, the main trend is still relatively stable. It suggests that E-SUOT retains strong robustness to the UMAP embedding dimension and does not rely heavily on fine-tuning this hyper-parameter. In summary, the sensitivity analysis shows that E-SUOT remains robust to variations in the UMAP embedding dimension.

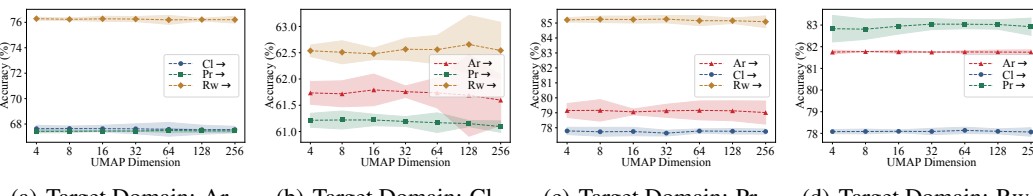

| (a) Target Domain: Ar. | (b) Target Domain: Cl. | (c) Target Domain: Pr. | (d) Target Domain: Rw. |

Figure F.1: Sensitivity of E-SUOT performance to the UMAP embedding dimension on the Office-Home dataset under the UDA setting. For dimension 256, the vanilla backbone features are used without UMAP. The feature extractor is pre-trained using the CoVI method. The shaded area indicates the ± 5.0 times standarad deviation error.

## G  LIMITATIONS & FUTURE DIRECTIONS AND BROADER IMPACT

### G.1  LIMITATIONS & FUTURE DIRECTIONS

The limitations and future research directions of this work can be summarized as follows:

- **Consideration of Label Information:** In this work, we focused primarily on feature adaptation and did not explicitly incorporate label or discriminator information into the adaptation process. As a result, the performance of the proposed E-SUOT framework may degrade under scenarios involving significant covariate shift (Sugiyama et al., 2007; Sugiyama & Kawanabe, 2012). An important direction for future research is to integrate label information into the transportation process, for example, classifier guidance approach (Courty et al., 2017a; Dhariwal & Nichol, 2021; Bonet et al., 2025; Zhuang et al., 2024), which could further enhance model robustness and adaptation performance.

- **Regularization for Transport Plan:** To facilitate computation, we introduced entropy regularization on the transport plan; however, this may introduce potential instability or blur sparsity in the map (Yin et al., 2025). Future work may explore alternative regularization strategies (Courty et al., 2014; 2017b), such as group sparsity (to better incorporate label priors) or Laplacian regularization (to preserve local relationships), in order to further stabilize training and improve the properties of the learned potential function $w$.

- **Exploration of Other Discrepancy:** In this work, we adopted the Wasserstein distance as the primary metric for measuring domain discrepancy. However, other discrepancy measures, such as the Fisher-Rao distance (Zhang et al., 2022; Wang et al., 2023; Zhu, 2025), could also be explored to enable more flexible or principled adaptation approaches. Future work may investigate the use of alternative metrics (Neklyudov et al., 2023; Skreta et al., 2025) to further improve the effectiveness of the quality of intermediate domain thereby improving the performance of GDA.

- **Assumption of Label Invariance along the Transport Path:** The current formulation assumes that labels remain invariant during adaptation, i.e., $y_{t+1} \leftarrow y_t$, and thus primarily focuses on aligning the marginal feature distributions $p(x_t)$. This assumption may limit performance under pronounced label shift scenarios, where the conditional relationship $p(y|x)$ varies across domains, a case often encountered in unbalanced or fine-grained settings. Although our framework can be extended by incorporating classifier uncertainty or pseudo-label refinement (drawing inspiration from self-training schemes such as Eq. (3)–(4) in reference (Kumar et al., 2020)), handling substantial concept drift remains an open challenge. Future work may consider integrating adaptive label transport (Courty et al., 2017a) or uncertainty-aware pseudo-labeling (Kumar et al., 2020; Zhuang et al., 2024) to explicitly account for label-shift dynamics along the adaptation trajectory.

- **Higher Efficiency Utilization of Neural Networks:** In our current design, each stage requires training three separate networks, namely $w_\phi$, $T_\theta$, and $h_\omega$, to generate each intermediate domain. Although this strategy can save computation time compared to existing approaches that perform intermediate-domain generation online during the domain adaptation stage, it may still be suboptimal in terms of overall training efficiency in the offline stage. A promising future direction is to reformulate the training of $T_\theta$ into a more parameter-efficient form, such as adopting a LoRA-style adaptation (Hu et al., 2022) or using reparameterization trick to parameterize the difference between different stage (Choi et al., 2024). For $w_\phi$ and $h_\omega$, one possible improvement is to fine-tune only the last layer (Harrison et al., 2024; Brunzema et al., 2025), which could further reduce the offline training cost.

### G.2  BROADER IMPACT STATEMENT

GDA addresses a critical challenge in machine learning: transferring knowledge from a labeled source domain to an unlabeled target domain when there is a substantial gap between the two. Rather than relying on abrupt, one-shot shifts—which are often brittle in the face of large distributional discrepancies—GDA interpolates through a series of intermediate domains, allowing for a smoother and more effective adaptation process. This paradigm has direct implications for many real-world applications. For example, in recommender systems, GDA enables knowledge transfer to serve cold-start users or to integrate new items, and in language processing it allows models trained on high-resource languages to adapt more robustly to low-resource languages. By constructing and

navigating intermediate distributions, GDA provides a principled foundation for bridging domain gaps and ensuring stable model performance under challenging conditions. Our work advances the field of GDA by unifying flow-based methods and optimal transport within the semi-dual formulation, identifying fundamental issues of stability and generalization that have limited previous approaches. We further propose theoretically-grounded regularization strategies that improve the robustness and reliability of the adaptation process. These advances not only deepen the theoretical understanding of GDA but also offer practical benefits for deploying adaptable machine learning systems in diverse settings. We believe our findings will help catalyze the development of more general, stable, and information-preserving domain adaptation methods, with impact across fields ranging from recommendation and computational linguistics to broader AI applications.

## H   LLM USAGE STATEMENT

In accordance with the conference guidelines, we disclose our use of Large Language Models (LLMs) in the preparation of this paper as follows:

We used LLMs (specifically, OpenAI GPT-4.1, GPT-5 and Google Gemini 2.5) *solely for checking grammar errors and improving the readability of the manuscript*. The LLMs *were not involved in research ideation, the development of research contributions, experiment design, data analysis, or interpretation of results*. All substantive content and scientific claims were created entirely by the authors. The authors have reviewed all LLM-assisted text to ensure accuracy and originality, and take full responsibility for the contents of the paper. The LLMs are not listed as an author.

