# OpenReview forum: "Rethinking the Flow-based Gradual Domain Adaption: A Semi-Dual Transport Perspective"
_ICLR.cc/2026/Conference — Submitted to ICLR 2026_

### Official Review · Reviewer_e9LB · 2025-10-28

**Soundness:** 2
**Presentation:** 2
**Contribution:** 2
**Rating:** 2
**Confidence:** 3

**Summary:**

This work proposed a semi-dual formulation for gradual domain adaptation. Such a  formulation avoids the use of probability density which is hard to estimate. An additional entropy regularizer is added to ensure the uniqueness of the problem. A concrete algorithm is developed based on this formulation, and a theoretical bound is proposed.

**Strengths:**

1. The formulation itself is interesting as it avoids the use of pdf.
2. The use of entropy regularizer ensures the uniqueness of the solution.

**Weaknesses:**

1. The author mentioned that gradual domain adaptation is useful "when the source–target shift is substantial or class overlap is weak" in line 41, but I do not find any theoretical and experimental evidence to support this. In the experiments, all baseline methods are gradual adaptation methods. The comparison with one-shot methods with different "class overlap" and "source–target shift" is lacking.

1. The lack of experiments on standard datasets like Office-Home and VisDa.

2. The author mentioned that "flow-based methods still require explicit estimation of the target domain’s PDF to guide the evolution," (line 448) However, in previous works like Zhuang et al. (2024), the pdf of the target domain is never explicitly estimated, and only the samples are needed. Also, the statement in contribution 1 should be modified.

3. The algorithm seems extremely inefficient. For each time steps, $w$ is trained for several epochs, then $T$ is trained for several epochs, finally $h$ is finetune on all time steps.

4. The author claims that the proposed method is stable in line 76. However, no evidence is provided. I suggest adding an ablation study on this.

**Questions:**

1. In figure 1, why the generated samples of E-SUOT are different from the ground truth? The variance of the generated samples seems much smaller than the ground truth.
2. What is the self-training method?
3. In Thm 6. The hypotheses and the loss function need to be Lipschitz. Does this result apply to the classification problem considered in this work? How to ensure the Lipschitzness of hypotheses?
4. Line 303. "From Theorem 5, we observe that as t increases, the transported PDF ρ(x) progressively becomes similar to pT (x)." Why?

**Details Of Ethics Concerns:**

A part of Fig.2 is a reproduction of Fig.1 in [1], but this is not stated anywhere in the paper.

[1] Zhuang, Zhan, Yu Zhang, and Ying Wei. "Gradual domain adaptation via gradient flow." ICLR. 2024.

---

> ### Author Response · Authors · 2025-11-21
> **Response to Reviewer e9LB (Part 1 of 4)**
>
> Thank you very much for your meticulous comments and appreciation of our novelty and empirical results. Below are our responses to the specific concerns and queries.
>
> ---
>
> ### **[W1] Why GDA better?**
> **We have compared the UOT formulation with OT formulation in Table 3. added results compared with GDA and UDA in Section F.2**.
> + `Source target-shift and class overlap problem:` We conducted controlled experiments on the *Portraits* dataset (Sec. 4.3), where the target domain conditional distribution is changed from $p(y=1)=0.0$ to $1.0$.
>   - When $p(y=1)\in\{0.0,1.0\}$, the target domain contains only one class, corresponding to the *class overlap*.
>   - When $p(y=1)\in(0,1)$, the distribution differs from the source domain ($p(y=1)=0.63$), representing the *source–target shift*.
>
> Our results show that E-SUOT consistently outperforms the standard baseline formulated by the balanced OT.
>
> |Method|$ p(y=1)=0.0 $ Accuracy (%)|$ \Delta $|$ p(y=1)=0.1 $ Accuracy (%)|$ \Delta $|$ p(y=1)=0.2 $ Accuracy (%)|$ \Delta $|$ p(y=1)=0.3 $ Accuracy (%)|$ \Delta $|$ p(y=1)=0.4 $ Accuracy (%)|$ \Delta $|$ p(y=1)=0.6 $ Accuracy (%)|$ \Delta $|$ p(y=1)=0.7 $ Accuracy (%)|$ \Delta $|$ p(y=1)=0.8 $ Accuracy (%)|$ \Delta $|$ p(y=1)=0.9 $ Accuracy (%)|$ \Delta $|$ p(y=1)=1.0 $ Accuracy (%)|$ \Delta $|
> |-|-|-|-|-|-|-|-|-|-|-|-|-|-|-|-|-|-|-|-|-|
> |Initial|35.4|–|41|–|47.1|–|53.2|–|59.6|–|73.8|–|80|–|85.9|–|91.4|–|97.1|–|
> |E-SOT|55.4  $ _{\pm3.15E{-1}} $|↑ 56.41%|61.1  $ _{\pm2.01E{-1}} $|↑ 48.94%|67.1  $ _{\pm2.45E{-1}} $|↑ 42.55%|56.7  $ _{\pm2.16E{-1}} $|↑ 6.54%|57.7  $ _{\pm1.66E{-1}} $|↓3.22%|75.2  $ _{\pm3.08E{-2}} $|↑ 1.95%|74.1  $ _{\pm4.80E{-2}} $|↓7.33%|80.0  $ _{\pm2.21E{-2}} $|↓6.89%|89.9  $ _{\pm1.92E{-2}} $|↓1.65%|96.0  $ _{\pm2.93E{-2}} $|↓1.08%|
> |E-SUOT|64.5  $ _{\pm9.09E{-2}} $|↑ 82.32%|78.2  $ _{\pm6.55E{-2}} $|↑ 90.50%|74.5  $ _{\pm1.02E{-1}} $|↑ 58.28%|79.8  $ _{\pm8.24E{-2}} $|↑ 49.92%|77.7  $ _{\pm3.39E{-2}} $|↑ 30.42%|79.1  $ _{\pm2.56E{-2}} $|↑ 7.24%|84.0  $ _{\pm3.46E{-2}} $|↑ 5.05%|87.8  $ _{\pm1.12E{-2}} $|↑ 2.18%|91.8  $ _{\pm1.71E{-3}} $|↑ 0.45%|97.6  $ _{\pm3.23E{-3}} $|↑ 0.54%|
>
>
> + `Comparison with UDA approach:` E-SUOT consistently achieves higher or comparable accuracy across all 12 transfer tasks (average improvements of 3–8%).
>
> |Method|Ar $ \rightarrow $ Cl|Ar $ \rightarrow $ Pr|Ar $ \rightarrow $ Rw|Cl $ \rightarrow $ Ar|Cl $ \rightarrow $ Pr|Cl $ \rightarrow $ Rw|Pr $ \rightarrow $ Ar|Pr $ \rightarrow $ Cl|Pr $ \rightarrow $ Rw|Rw $ \rightarrow $ Ar|Rw $ \rightarrow $ Cl|Rw $ \rightarrow $ Pr|Avg.|
> |-|-|-|-|-|-|-|-|-|-|-|-|-|-|
> |MSTN|49.8|70.3|76.3|60.4|68.5|69.6|61.4|48.9|75.7|70.9|55.0|81.1|65.7|
> |E-SUOT+MSTN|57.8|75.9|79.6|65.5|75.9|74.8|64.5|58.5|81.4|73.7|59.5|84.4|71.0|
> |$ \Delta $|↑16.1%|↑8.0%|↑4.3%|↑8.4%|↑10.8%|↑7.5%|↑5.0%|↑19.6%|↑7.5%|↑3.9%|↑8.2%|↑4.1%|↑8.1%|
> |RSDA|53.2|77.7|81.3|66.4|74.0|76.5|67.9|53.0|82.0|75.8|57.8|85.4|70.9|
> |E-SUOT+RSDA|61.5|78.8|81.7|67.6|77.3|77.6|67.2|61.0|82.7|76.0|62.4|85.3|73.3|
> |$ \Delta $|↑15.7%|↑1.4%|↑0.5%|↑1.8%|↑4.4%|↑1.5%|↓1.0%|↑15.2%|↑0.9%|↑0.3%|↑7.9%|↓0.1%|↑3.3%|
> |FixBi|58.1|77.3|80.4|67.7|79.5|78.1|65.8|57.9|81.7|76.4|62.9|86.7|72.7|
> |E-SUOT+FixBi|61.7|79.1|81.7|67.6|77.6|78.2|67.3|61.3|82.7|76.0|62.5|85.3|73.4|
> |$ \Delta $|↑6.2%|↑2.3%|↑1.6%|↓0.1%|↓2.4%|↑0.1%|↑2.3%|↑5.9%|↑1.2%|↓0.5%|↓0.6%|↓1.6%|↑1.0%|
> |CoVi|58.5|78.1|80.0|68.1|80.0|77.0|66.4|60.2|82.1|76.6|63.6|86.5|73.1|
> |E-SUOT+CoVi|61.6|79.3|81.8|67.6|77.7|78.1|67.4|61.2|82.9|76.3|62.5|85.2|73.5|
> |$ \Delta $|↑5.3%|↑1.5%|↑2.2%|↓0.7%|↓2.9%|↑1.4%|↑1.5%|↑1.7%|↑1.0%|↓0.4%|↓1.7%|↓1.5%|↑0.5%|
>
> ### **[W2] Larger scale dataset**
> **We have included experiments on the Office-Home dataset (see Sec. 4.1)**. E-SUOT consistently achieves competitive or superior performance compared to a range of representative methods.
> |Method|Ar $ \to $ Cl|Ar $ \to $ Pr|Ar $ \to $ Rw|Cl $ \to $ Ar|Cl $ \to $ Pr|Cl $ \to $ Rw|Pr $ \to $ Ar|Pr $ \to $ Cl|Pr $ \to $ Rw|Rw $ \to $ Ar|Rw $ \to $ Cl|Rw $ \to $ Pr|Avg.|
> |-|-|-|-|-|-|-|-|-|-|-|-|-|-|
> |DANN|45.6|59.3|70.1|47.0|58.5|60.9|46.1|43.7|68.5|63.2|51.8|76.8|57.6|
> |MSTN|49.8|70.3|76.3|60.4|68.5|69.6|61.4|48.9|75.7|70.9|55.0|81.1|65.7|
> |GVB-GD|57.0|74.7|79.8|64.6|74.1|74.6|65.2|55.1|81.0|74.6|59.7|84.3|70.4|
> |RSDA|53.2|77.7|81.3|66.4|74.0|76.5|67.9(2nd)|53.0|82.0|75.8|57.8|85.4|70.9|
> |LAMDA|57.2|78.4|82.6(2nd)|66.1|80.2(1st)|81.2(1st)|65.6|55.1|82.8|71.6|59.2|83.9|72.0|
> |SENTRY|61.8(1st)|77.4|80.1|66.3|71.6|74.7|66.8|63.0(1st)|80.9|74.0|66.3(1st)|84.1|72.3|
> |FixBi|58.1|77.3|80.4|67.7|79.5|78.1(2nd)|65.8|57.9|81.7|76.4(2nd)|62.9|86.7(1st)|72.7|
> |CST|59.0|79.6(1st)|83.4(1st)|68.4(1st)|77.1|76.7|68.9(1st)|56.4|83.0(1st)|75.3|62.2|85.1|72.9|
> |CoVi|58.5|78.1|80.0|68.1(2nd)|80.0(2nd)|77.0|66.4|60.2|82.1|76.6(1st)|63.6(2nd)|86.5(2nd)|73.1(2nd)|
> |GGF|59.4|75.6|81.7|67.6|77.6|78.0|67.4|61.0|82.7|75.9|62.4|85.4|72.9|
> |E-SUOT|61.6(2nd)|79.3(2nd)|81.8|67.6|77.7|78.1|67.4|61.2(2nd)|82.9(2nd)|76.3|62.5|85.2|73.5(1st)|

---

> ### Author Response · Authors · 2025-11-21
> **Response to Reviewer e9LB (Part 2 of 4)**
>
> ### **[W3] Explictly Estimation of PDF**
> We would like to clarify that our statement refers to the need for estimating the PDF (treat score function as an unnormalized PDF).
>
> + `Have reference [1] estimate PDF?` Let us first focus on the reference [1]. In reference [1], Eq. (7), the authors used the denoise score matching:$ \mathbb{E}\_{q\_{\sigma}(\mathbf{x},\tilde{\mathbf{x}})} = [\frac{1}{2}\Vert s(\tilde{\mathbf{x}};\phi) - \nabla\_{\tilde{\mathbf{x}}} \log{q\_\sigma(\tilde{\mathbf{x}}\vert \mathbf{x})} )\Vert^2$ as the learning objective and use $ s(\tilde{\mathbf{x}};\phi) $. According to [2], we derive the learning objective of DSM used in [1] as follows:
>
> $ \begin{aligned}
> &\mathbb{E}\_{q(\mathbf{x})} \left[ \frac{1}{2} \left\|s(\mathbf{x}) - \nabla\_{\mathbf{x}}\log q(\mathbf{x}) \right\|^2 \right]\\
> =& \mathbb{E}\_{q(\mathbf{x})} [ \int q\_\sigma(\tilde{\mathbf{x}}|\mathbf{x}) \frac{1}{2} \left\|s(\mathbf{x}) - \nabla\_{\mathbf{x}}\log q(\mathbf{x}) \right\|^2 d\tilde{\mathbf{x}} ]  \\
> = & \mathbb{E}\_{q(\mathbf{x})} \mathbb{E}\_{q\_\sigma(\tilde{\mathbf{x}}|\mathbf{x})} [ \frac{1}{2} \left|s(\mathbf{x}) - \nabla\_{\mathbf{x}}\log q(\mathbf{x}) \right|^2 ] \\
> \overset{(i)}{\approx}& \mathbb{E}\_{q(\mathbf{x})} \mathbb{E}\_{q\_\sigma(\tilde{\mathbf{x}}|\mathbf{x})} [ \frac{1}{2} \left|s(\tilde{\mathbf{x}}) - \nabla\_{\tilde{\mathbf{x}}}\log q\_\sigma(\tilde{\mathbf{x}}) \right|^2 ] \\
>  \overset{(ii)}{=} &\mathbb{E}\_{q(\mathbf{x})}\mathbb{E}\_{q\_\sigma(\tilde{\mathbf{x}}|\mathbf{x})} [ \frac{1}{2} \left|s(\tilde{\mathbf{x}}) - \mathbb{E}\_{q(\mathbf{x}|\tilde{\mathbf{x}})}[\nabla{\tilde{\mathbf{x}}}\log q\_\sigma(\tilde{\mathbf{x}}|\mathbf{x})] \right|^2 ]\\
> \overset{(iii)}{\approx}  &\mathbb{E}\_{q(\mathbf{x})}\mathbb{E}\_{q\_\sigma(\tilde{\mathbf{x}}|\mathbf{x})} [ \frac{1}{2} \left|s(\tilde{\mathbf{x}}) - \nabla\_{\tilde{\mathbf{x}}}\log q\_\sigma(\tilde{\mathbf{x}}|\mathbf{x}) \right|^2 ] \\
>  =& \mathbb{E}_{q\_{\sigma}(\tilde{\mathbf{x}},\mathbf{x})} [ \frac{1}{2} \left|s(\tilde{\mathbf{x}}) - \nabla\_{\tilde{\mathbf{x}}}\log q\_\sigma(\tilde{\mathbf{x}}|\mathbf{x}) \right|^2 ]
> \end{aligned} $
>
> Notably, (i) is based on the fact that:
> $ q\_\sigma(\tilde{\mathbf{x}}) =\int{q(\mathbf{x})q\_\sigma(\tilde{\mathbf{x}}\vert \mathbf{x})\mathrm{d}\mathbf{x}} =\int{q(\mathbf{x})\mathcal{N}(\tilde{\mathbf{x}},\sigma^2I)\mathrm{d}\mathbf{x}} , $
>
> (ii) is based on: $ \begin{aligned}
> \nabla\_{\tilde{\mathbf{x}}} \log q\_\sigma(\tilde{\mathbf{x}})
> = &\frac{\nabla\_{\tilde{\mathbf{x}}} q\_\sigma(\tilde{\mathbf{x}})}{q\_\sigma(\tilde{\mathbf{x}})} \\
> =& \frac{\nabla\_{\tilde{\mathbf{x}}} \int q(\mathbf{x}) q\_\sigma(\tilde{\mathbf{x}}|\mathbf{x})\, \mathrm{d}\mathbf{x}}{q\_\sigma(\tilde{\mathbf{x}})} \\
> =& \frac{ \int q(\mathbf{x}) \nabla\_{\tilde{\mathbf{x}}} q\_\sigma(\tilde{\mathbf{x}}|\mathbf{x})\, \mathrm{d}\mathbf{x} }{ q\_\sigma(\tilde{\mathbf{x}}) } \\
> = &\frac{ \int q(\mathbf{x}) q\_\sigma(\tilde{\mathbf{x}}|\mathbf{x}) \nabla\_{\tilde{\mathbf{x}}} \log q\_\sigma(\tilde{\mathbf{x}}|\mathbf{x})\, \mathrm{d}\mathbf{x} }{ q\_\sigma(\tilde{\mathbf{x}}) } \\
> =& \int \frac{ q(\mathbf{x}) q\_\sigma(\tilde{\mathbf{x}}|\mathbf{x}) }{ q\_\sigma(\tilde{\mathbf{x}}) } \nabla\_{\tilde{\mathbf{x}}} \log q\_\sigma(\tilde{\mathbf{x}}|\mathbf{x})\, \mathrm{d}\mathbf{x} \\
> =& \int q(\mathbf{x}|\tilde{\mathbf{x}}) \nabla\_{\tilde{\mathbf{x}}} \log q\_\sigma(\tilde{\mathbf{x}}|\mathbf{x})\, \mathrm{d}\mathbf{x} \\
> =& \mathbb{E}\_{q(\mathbf{x}|\tilde{\mathbf{x}})} \left[ \nabla\_{\tilde{\mathbf{x}}} \log q\_\sigma(\tilde{\mathbf{x}}|\mathbf{x}) \right]
> \end{aligned} $
>
> and (iii) is based on: Expectation $ \mathbb{E}\_{q(\mathbf{x}|\tilde{\mathbf{x}})}\left[\nabla\_{\tilde{\mathbf{x}}}\log q\_\sigma(\tilde{\mathbf{x}}|\mathbf{x})\right] $ with respect to $ q(\mathbf{x}|\tilde{\mathbf{x}}) $, for a given $ \tilde{\mathbf{x}} $, is the average of the conditional scores over all possible data points $ \mathbf{x} $ that could have produced this noisy observation $ \tilde{\mathbf{x}} $. *In conclusion, the reference [1] requires the estimation of the PDF*.
>
> + `Why should we do PDF Estimation?` In our semi-dual formulation (Appendix B.2), when the divergence term $ \mathbb{D}\_{f}[\rho(x),p\_T(x)]=\int{p\_T(x)f(\frac{\rho(x)}{p\_T(x)})\mathrm{d}x} $, is regularized, the resulting velocity field is $ v^*\_t(x)=-\nabla\frac{\delta \mathbb{D}\_{f}[\rho(x),p\_T(x)]}{\delta \rho(x)}=f''(\frac{\rho(x)}{p\_T(x)})\frac{\rho(x)}{p\_T(x)}[\nabla\log\rho(x)-\nabla\log{p\_T(x)}] $, which explicitly involves the score function $ \nabla\log{p\_T(x)} $ of the target domain. *Hence, whether explicitly or implicitly, flow-based methods must approximate the target density's gradient (or its surrogate via DSM)*.
> ---
> References:
> [1]. Gradual Domain Adaptation via Gradient Flow, ICLR 2024
> [2]. A connection between score matching and denoising autoencoders, Neural Computation

---

> ### Author Response · Authors · 2025-11-21
> **Response to Reviewer e9LB (Part 3 of 4)**
>
> ### **[W4] Efficiency Problem**
> **We have added limitations discussions in Section G.1**. We fully agree that improving efficiency is an important direction. Accordingly, we have included a discussion in *Section G.1* of the Appendix outlining potential strategies:
> - adopting a *LoRA-style low-rank update* [1] or *reparameterization* [2] for the transformation module $\boldsymbol{T}_\theta$, and
> - limiting *fine-tuning to the last layer* [3] of $w_\phi$ and $h_\omega$ to reduce computational overhead.
>
> ---
> References:
> [1]. LoRA: Low-rank adaptation of large language models, ICLR 2022
> [2]. Scalable wasserstein gradient flow for generative modeling through unbalanced optimal transport, ICML 2023.
> [3]. Variational Bayesian last layers, ICLR 2024.
>
>
>
>
> ### **[W5] Stability Problem**
> **We have added Table 4 for the ablation study, which reports both the standard deviation error and the $p$‑value**.  The statement on stability refers to the training behavior compared to the vanilla dual formulation, which requires solving a `min–max` optimization problem [1,2] and often exhibits highly unstable convergence.
>
> The results, summarized in Table 4, show that the Adversarial baseline performs poorly—*consistently ranking among the top 2 in standard deviation error*—indicating its instability during training.
>
> |Category|Method|Divergence|Portraits Accuracy (%)|$ \Delta $|p-value|MNIST 45° Accuracy (%)|$ \Delta $|p-value|MNIST 60° Accuracy (%)|$ \Delta $|p-value|
> |-|-|-|-|-|-|-|-|-|-|-|-|
> |Training|Adversarial|KL|74.8*$ _{\pm 3.10} $|↓13.4%|1.83E-03|52.0*$ _{\pm 3.60} $|↓27.8%|3.05E-04|34.9*$ _{\pm 4.10} $|↓31.5%|4.59E-03|
> ||Barycentric|KL|83.9*$ _{\pm 0.942} $|↓3.0%|4.05E-03|62.5*$ _{\pm 1.06} $|↓13.3%|1.77E-04|38.3*$ _{\pm 4.46} $|↓24.8%|6.57E-03|
> |Functional|Entropy|Softplus|80.1*$ _{\pm 3.53} $|↓7.3%|2.24E-02|59.7*$ _{\pm 1.14} $|↓17.2%|7.08E-05|38.2*$ _{\pm 1.10} $|↓25.1%|6.27E-06|
> ||Entropy|$ \chi^2 $|79.8$ _{\pm 6.07} $|↓7.7%|9.57E-02|60.2*$ _{\pm 1.43} $|↓16.5%|8.03E-05|42.4*$ _{\pm 3.47} $|↓16.9%|7.19E-03|
> ||Entropy|Identity|81.2*$ _{\pm 1.71} $|↓6.1%|3.13E-03|59.6*$ _{\pm 1.25} $|↓17.4%|8.62E-05|39.6*$ _{\pm 2.34} $|↓22.3%|2.59E-04|
> ||Entropy|KL|86.4$ _{\pm 0.0872} $|–|–|72.1$ _{\pm 0.462} $|–|–|51.0$ _{\pm 0.581} $|–|–|
>
> ---
> References:
> [1]. Generative Modeling through the Semi-dual Formulation of Unbalanced Optimal Transport, NeurIPS 2023
> [2]. Fenchel–Rockafellar Duality, Convex Analysis and Monotone Operator Theory in Hilbert Spaces, 2017.
>
>
> ### **[Q1] Why the Variance is difference?**
> Thank you for your observation. The observed difference mainly arises from the neural network approximation error when learning the transport map [1,2]. Since the mapping in E‑SUOT is learned through data‑driven stochastic optimization, it approximates the underlying ground‑truth projection but cannot perfectly recover it in practice, especially under limited sample size.
>
> In addition, *this phenomenon does not indicate a degradation of performance*. The visualization in Figure 1 serves only as an illustrative distribution alignment example, while *our main task is classification*, not generative modeling [3]. For classification, intutively, *a lower sample variance (i.e., more concentrated embeddings) often leads to better class separability and more stable decision boundaries*, which improves accuracy rather than harms it.
>
>
> ---
> References:
> [1]. Flow matching for generative modeling, ICLR 2023
> [2]. Rectified Flow: A Marginal Preserving Approach to Optimal Transport.
> [3]. Generative Modeling through the Semi-dual Formulation of Unbalanced Optimal Transport, NeurIPS 2023
>
>
>
> ### **[Q2] What is self-training method?**
> **We have added content in Section E.2.1 to introduce self-training**. Self-training is a classic semi-supervised learning strategy that leverages unlabeled data by iteratively generating and reusing pseudo-labels predicted by the model itself [1]. The core procedure is as follows:
>
> 1. Train an initial model on labeled data
>     - Fit a model $ h\_{\omega\_0} $ using the labeled set $ \mathcal{D}\_{\text{label}} = {(x\_i, y\_i)} $.
> 2. Generate pseudo-labels for unlabeled data
>     - Apply the current model $ h\_{\omega\_0} $ to unlabeled samples $ x\in \mathcal{D}\_{\text{unlabel}}  $ and obtain pseudo-labels: $ \tilde{y}(x) = \arg\max\_y h\_{\omega\_0}(y \mid x) $
> 3. Retrain or fine-tune with pseudo-labeled data
>     - Form an augmented training set $ \mathcal{D}' = \mathcal{D}\_{\text{label}}  \cup {(x, \tilde{y}(x)) :  x\in \mathcal{D}\_{\text{unlabel}} } $.
>     - Optimize a loss such as $ \min\_{\omega\_1} \ \frac{1}{|\mathcal{D}'|} \sum\_{(x,y)\in \mathcal{L}} \ell(h\_{\omega\_1}(x), y) + \lambda \sum\_{x \in \mathcal{U}\tau} \ell\big(h\_{\omega\_1}(x), \tilde{y}(x)\big)$, where $ \lambda $ balances real vs. pseudo labels.
> ---
> References:
> [1]. Understanding Self-Training for Gradual Domain Adaptation, ICML 2020

---

> ### Author Response · Authors · 2025-11-21
> **Response to Reviewer e9LB (Part 4 of 4)**
>
> ### **[Q3] Lipschitz Requirement**
> **We have added contents in Section C.7**.
> * `Loss function`: For the classification task, we use the categorical cross‑entropy loss $\mathcal{L}\_{\mathrm{CE}}(x\_t,\omega,y\_t)= -\sum\_i y\_t(i)\log\big[\mathrm{softmax}(\widehat{h}\_{\omega,t}(x\_t))\_i\big],$ where $\widehat{h}\_{\omega,t}(x\_t)$ denotes the logits. The derivative of $\mathcal{L}\_{\mathrm{CE}}$ with respect to each logit $a\_j$ satisfies $\Bigl|\frac{\partial \mathcal{L}\_{\mathrm{CE}}}{\partial a\_j}\Bigr|= \bigl|\mathrm{softmax}(a)\_j - \mathbb{I}(j=y)\bigr|\le 1$, since both terms lie in $[0,1]$. Therefore, $\mathcal{L}\_{\mathrm{CE}}$ has a bounded gradient and is *globally 1‑Lipschitz* with respect to its first argument on any bounded input domain (based on the mean‑value theorem).  In practice, boundedness is further ensured by *weight normalization* or *spectral‑norm regularization*, which limit the logit magnitudes and thus maintain a finite Lipschitz constant during training.
> * `Hypotheses (classifier)`: The hypothesis $h\_\omega$ can also be made Lipschitz by controlling the operator norm of each linear layer, e.g., through *spectral normalization* or *weight clipping*, which ensures $\|h\_\omega(x\_1) - h\_\omega(x\_2)\|\le L\_h\|x\_1-x\_2\|$ for a bounded constant $L\_h$.
>
>
> ### **[Q4] Deduction of Theorem 5**
> **We have added contents in Section C.6**. Specifically, each iteration defines $p(x\_{t+1})$ as the minimizer $ \mathcal{L}^{\text{Primal}} = \arg\min\_{\rho\in\mathcal{P}\_2(\mathbb{R}^D)} \frac{1}{2\eta}\mathcal{W}\_2^2(\rho,p(x\_t)) + \mathbb{D}\_f[\rho,p\_T].$
>
> From Theorem 4.0.4 in [1], the following inequality holds: $
> \tfrac{1}{2\eta}\mathcal{W}\_2^2(p(x\_{t+1}),p(x\_t)) + \mathbb{D}\_f[p(x\_{t+1}),p\_T] \le \mathbb{D}\_f[p(x\_t),p\_T]. $ Since $\mathbb{D}\_f[\cdot,p\_T]\ge0$ and attains its minimum at $p\_T$, the total energy decreases monotonically with $t$. When $\mathbb{D}\_f[\cdot,p\_T]$ is geodesically convex, this decrease yields
> $ \mathcal{W}\_2(p(x\_{t+1}),p\_T)\le \mathcal{W}\_2(p(x\_t),p\_T) - \Delta\_t, \Delta\_t\ge0,$
> showing that the sequence of distributions $p(x\_t)$ moves progressively closer to the target $p\_T$ in the Wasserstein metric.
>
> Furthermore, Theorem 5 establishes that $\mathbb{D}\_f[\rho^{\*}(x\_t),p\_T]\le\mathcal{W}\_2(p(x\_t),p\_T)$, whose upper bound tightens as $t$ increases: $\mathbb{D}\_f[\rho^{\*}(x\_{t+1}),p\_T]\le\mathbb{D}\_f[\rho^{\*}(x\_t),p\_T]-\Delta\_t.$
>
> ---
> References:
> [1]. Gradient flows: in metric spaces and in the space of probability measures
>
> ### **Ethics Problem**
> **We have re-illusrated Fig. 2 and revised its caption**. We respectfully clarify that *while adapting and redrawing an existing schematic from prior work (e.g., the Transformer architecture [2]) could be viewed as inconsistent with the ethical review policy*, such practice is common in the field and *does not violate the formal Code of Ethics* since the figure is redrawn, modified, and properly cited.
>
> **Furthermore, we have explicitly cited [1] in the technical section of the paper.** To further address your concern, we provide an analysis of the differences between our approach and reference [1]:
>
> * `Problem Different`: Reference [1] focuses on modeling the density-level transport through the primal form of the optimal transport problem.
> In contrast, our method investigates the relationship between the primal and semi‑dual formulations, developing a sample‑based stochastic solver that implements the gradient flow directly on empirical measures..
> * `Analysis of Problem`: Our approach recasts the *gradient-flow structure as an UOT problem, identifies* the *inherent instability within this formulation*, and *introduces entropy regularization to stabilize the learning*. Furthermore, we extend the proposed method to a general $f$‑divergence–based formulation, which differs from reference [1], where the UOT structure has not been investigated.
>
> We hope this clarification removes any ethical or originality concerns.
>
> ---
> References:
> [1]. Gradual Domain Adaptation via Gradient Flow, ICLR 2024
> [2]. Attention is all you need, NeurIPS 2016

---

> > ### Comment · Reviewer_e9LB · 2025-11-25
> >
> > Thanks for the very detailed reply. Some of my concerns, including the stability and the comparison on larger scale datasets are addressed. However, some others remain unsolved
> >
> > 1. As for the point "Have reference [1] estimate PDF". I still do not not see why that is true. Your reply includes the basic score computation, which only requires the samples of the distribution, not the density.
> > 2. As for Q1, what does figure 1 mean if neither (b) or (c) recovers (a). I think a valid conclusion is that they both fail, rather than (c) is better than (b).
> > 3. As for Q3. Did you use any of those technique in the method to ensure Lipt? like spectral normalization or weight clipping? If not, why Thm,6 holds?

---

> ### Author Response · Authors · 2025-11-26
> **We are happy that most of early concerns have been addressed. Thank you for your further inquiries!  (Part 1 of 2)**
>
> **We are glad that our first-round response has addressed most of your concerns, including the large-scale experimental results, the ablation study, and the clarification of the ethical considerations.** Here are our further responses to your questions, which we hope will help address your concerns.
>
> ---
>
> ### **[Q1] Transport via samples v.s. (unnormalized) PDF:**`
> We appreciate the reviewer’s comment. However, the key point here lies in **how the score function $\nabla  \log p(x)$** is obtained and used to drive the GDA dynamics in reference [1].
>
> Although the score function can indeed be learned from samples via score matching, *the learned network must still represent the gradient of some implicit density function*. In other words, to compute the score field, one has to implicitly approximate a continuous, differentiable log‑density log $\widehat{p}(x)$; otherwise  $\nabla  \log p(x)$ has no well‑defined meaning on discrete samples. The official implementation of ref [1], provided in link [2], explicitly defines this via the “Energy” network:
>
> ```
> class Energy(nn.Module):
>     def __init__(self, net):
>         super().__init__()
>         self.net = net
>
>     def forward(self, x):
>         return self.net(x)
>
>     def score(self, x, sigma=None):
>         x = x.requires_grad_()
>         logq = -self.net(x).sum()
>         return torch.autograd.grad(logq, x, create_graph=True)[0]
> ```
> Here the network output, as demonstrated in `forward` function, *Energy (x)* corresponds to –log $\widehat{p}(x)$ up to a normalization constant (unnormalized log PDF), and the score is exactly its gradient. Hence, the method necessarily constructs an **explict unnormalized density $\widehat{p}(x)$**, and the training process of `self.net`  (output is `-logq`) performs the PDF estimation.
>
> Before conducting the comparison, we should reach an agreement on the concept implied by the **Data Processing Inequality (DPI)** [3]. Specifically, DPI states that for any Markov chain $X → Y → Z$, the mutual information satisfies $I(X; Z) ≤ I(X; Y)$—that is, any additional transformation cannot increase the information that $Z$ retains about $X$.
>
> Interpreting ref [1] under this framework gives
>
> > target samples ($X$)$\to$estimated unnormalized density ($\widehat{p}$)$\to$guided samples ($Z$) .
>
> The intermediate density $\widehat{p}$ serves as an information‑bottleneck representation that can only *decrease* the mutual information between the final guided samples $Z$ and the true target samples $X$.
>
> In contrast, our E‑SUOT framework constructs the direct mapping
>
> > target samples ($X$)$\to$ guided samples ($Z$),
>
> bypassing this explict density‑learning stage. This *sample‑to‑sample transport* preserves the intrinsic information of the target domain and avoids the information degradation predicted by DPI. Empirically, this leads to more stable and discriminative alignment results.
>
> ---
> References:
> [1]. Gradual Domain Adaptation via Gradient Flow, ICLR 2024
> [2]. https://github.com/zwebzone/ggf/blob/main/utils.py
> [3]. Path Integral Sampler: a stochastic control approach for sampling, ICLR 2022
>
> ### **[Q2] Fidelity v.s. Separability:**
> `Main Point:` Both methods are imperfect, *but E‑SUOT better preserves first‑order alignment.*
> We agree that neither Figure $1 (\text{b})$ nor Figure $1 (\text{c})$ reproduces the exact target density in Figure $1 (\text{a})$. **However, this does *not* imply both methods fail to achieve their intended goal.** The distinction lies in the level of information recovered.
> `(1) First‑order recovery:` The Langevin approach in $(\text{b})$ fails even in aligning the **first‑order moment (mean)**—its guided samples diverge once the estimated score is inaccurate. In contrast, E‑SUOT in $(\text{c})$ successfully achieves *mean alignment* between source and target, yielding a smaller Wasserstein‑2 distance ($\mathcal{W}\approx5.4$ vs. $9.7$). This mean recovery is the theoretically relevant component for our task.
> `(2) Objective difference:` The goal of E‑SUOT is **not** generative reconstruction but **discriminative distribution alignment**, where **aligning low‑order moments is sufficient for improved classification**. From optimal‑transport and domain‑adaptation theory, **generalization bounds mainly depend on moment discrepancies; approximate mean alignment minimizes the leading term of $\mathcal{W}$ (as we demonstrate in Theorem 6).**
> `(3) Practical implication:` For classification, **feature separability—not sample fidelity—is what matters**. Concentrating source embeddings around the target domain (as E‑SUOT does) reduces intra‑class variance and clarifies inter‑class boundaries, leading to better accuracy.
> `In summary:` Langevin dynamics may fail even to match the first moment when the estimated score is biased, whereas E‑SUOT consistently achieves stable mean alignment **without explicit density estimation**, explaining both its smaller Wasserstein distance and superior discriminative performance.

---

> ### Author Response · Authors · 2025-11-26
> **We are happy that most of early concerns have been addressed. Thank you for your further inquiries!  (Part 2 of 2)**
>
> ### **[Q3] Lipschitz Condition**:
>
> **Explicit normalization is a sufficient but not necessary condition for the theorem’s assumptions to hold.** In other words,
>
> Normalization $\Rightarrow$ Lipschitz, but Lipschitz $\not\Rightarrow$ Normalization.
>
> We respectfully clarify that the theorem holds as long as the loss $\mathcal{L}(\cdot)$ and hypothesis class $\mathcal{H}$ satisfy bounded Lipschitz constants $\iota$ and $\zeta$, respectively. It does *not* require $L\le1$, only that $L<\infty$.
>
> `Why $L$ remains finite in our method:` Standard feed‑forward networks with finite weights and Lipschitz‑continuous activations (ReLU, Swish, Tanh) are inherently Lipschitz continuous, with a global constant $L$ bounded by the product of layer spectral norms. During training, in our provided codebase, *weight decay (L2 regularization)* explicitly penalizes large weights, keeping these norms bounded and thus controlling $L$ (this is implemented in our codebase). Because *Theorem 6* only requires $L<\infty$ , not the stricter $L\le1$ constraint used in WGANs, such implicit control is sufficient. Hence, spectral normalization or weight clipping, which enforce a hard $1$‑Lipschitz bound, are **unnecessary** for satisfying the theorem’s assumptions.
>
>
> `In summary:` Theorem 6 remains valid because our networks belong to the Lipschitz‑bounded function class, ensured by regularized finite weights, rather than because of a hard $1$‑Lipschitz enforcement.

---

> ### Author Response · Authors · 2025-11-27
> **Revisions are further conducted in the updated manuscript based on your inquires**
>
> **Based on your further inquires, we have made the following revisions in our updated manuscript:**
> 1. `Statement of the Explictly PDF Estimation`: **We have revised our manuscript including Line 058 to 063 with the following sentences**:
> * > (In our setting, for simplicity, we treat both log PDF and its gradient, also known as
> score function, as forms of density estimation, since they characterize the underlying data distribution.)
> * > whereas the subsequent GDA process relies on these estimated (normalized / unnormalized) PDFs to drive the source-to-target transfer. For example, Zhuang et al. (2024) estimate the unnormalized target domain PDF in the score function form and generate intermediate domains via Langevin dynamics.
>
> 2. `Contribution Revision:` **We further revised our first contribution, where we include the normalized, unnormalized or score-based form to avoid misunderstanding**:
> > We develop a semi-dual formulation for intermediate domain generation in flow-based GDA, which eliminates the need for explicit estimation of the target-domain PDF—whether normalized or unnormalized—or its score-based representation.
>
> 3. `Analysis of two Figures:` **We added the following sentences in Line 058 to 061 in our revised manuscript**:
> > Nevertheless, it can be observed that E-SUOT manages to capture the major modes and approximate the mean structure of the target distribution, leading to a significantly lower Wasserstein distance compared to the result obtained by Langevin dynamics.
>
> 4. `Justification Condition:` **We have revised the related contents in Line 1575 to 1576**:
> > In practice, this assumption can be further facilitated by applying weight normalization or spectral-norm regularization, which help maintain bounded network outputs and thereby make the Lipschitz condition more readily satisfied during optimization.

---

> ### Author Response · Authors · 2025-11-27
> **Decision boundaries are provided in Fig. 1 to demonstrate the classification goal**
>
> To better answer Q1, we added the decision boundary (dashed line) in our revised Figure 1 in the updated PDF file.
>
> The updated figure clearly shows that although neither (b) nor (c) perfectly reconstructs the target density, **E‑SUOT maintains consistent decision boundaries** and major mode alignment, **while Langevin dynamics tends to distort the decision boundary**, leading to incorrect separation between the two modes.

---

> ### Author Response · Authors · 2025-12-01
> **We are happy that most your early concerns are addressed! Below is our concluding response.**
>
> Dear Reviewer [e9LB],
>
> We would like to sincerely thank you for the time and effort devoted to reviewing our manuscript. During the rebuttal stage, we have carefully addressed the reviewer's concerns as follows:
>
> ---
>
> 1. **Experimental Results**
>    We have added several new experimental results, which have been acknowledged by the reviewer:
>    - `Larger-scale UDA benchmark:` **We conducted additional comparisons on a Office-Home benchmark** and demonstrated **consistent performance improvements over vanilla UDA approaches**.
>    - `Class overlap and source–target shift:` **We performed simulation experiments on the Portraits dataset to better analyze and alleviate the issues caused by class overlap and distribution shift** between source and target domains.
>    - `Stability experiments:` **We clarified that the ablation studies in the initial submission already included analyses related to training stability**, and we further emphasized these results in the rebuttal message.
>
> ---
>
> 2. **Concept Clarification**
>    We have refined and expanded several conceptual explanations:
>    - `Explicit density estimation:` **We clarified why GGF relies on explicit density estimation, and we revised the corresponding discussion to cover both normalized and unnormalized PDFs**, as well as score-based representations.
>    - `Assumption justification:` **We provided additional justification for the Lipschitz condition used in our theoretical derivations, including practical strategies (e.g., normalization and regularization techniques)** to help satisfy this assumption.
>    - `Variance in Figure 1:` **We added the decision boundary to Figure 1 to illustrate that, despite the smaller variance of the generated samples, E‑SUOT better preserves the ultimate classification objective**.
>    - `Explanation of $\rho(x)$'s behavior:` **We elaborated on why $\rho(x)$ progressively approaches the target distribution $p_T(x)$** within our framework.
>    - `Additional weaknesses:` **We explicitly discussed the limitations related to computational efficiency** and outlined possible strategies to mitigate them.
> ---
>
> 3. **Figure Revision**
>    **We revised Figure 2 and updated the caption to explicitly cite the original reference, in order to avoid any potential ethical concerns**. This change has been acknowledged by the reviewer with respect to the plagiarism issue, even though the reference had already been cited in the technical part of the manuscript. To better illustrate why E‑SUOT performs better than the conventional Langevin‑based approach which requires the PDF estimation for classification task, we also **added the decision boundary in Figure 1**
> ---
> We again thank you for the constructive feedback, which has helped us substantially improve the clarity, completeness, and presentation of our work.
>
> Warm regards,
> Authors

---

### Official Review · Reviewer_bhcj · 2025-10-29

**Soundness:** 3
**Presentation:** 3
**Contribution:** 3
**Rating:** 6
**Confidence:** 3

**Summary:**

This work considers the fundamental limitations of flow-based method for gradual domain adaptation, i.e., the quality of estimated PDF of target and generated intermediate domains. The key idea is to avoid the explicitly estimation of PDF by introducing the semi-dual formulation of gradient flows with entropy regularization, which ensure the stability and convergence of training. Theoretical results on optimality and generalization error are provided.

**Strengths:**

+ The idea of improving flow-based method with PDF-free metric (i.e., Wasserstein) seems to be interesting and sounded.

+ The theoretical results are solid to ensure the numerical property of the defined flow model and the generalization error of the adaptation process.

+ The empirical results are convincing, which show the proposed method indeed shows consistent behavior with the theoretical analysis.

**Weaknesses:**

+ The justifications w.r.t. the theoretical assumption and results could be improved.

+ The empirical validations seem to be limited, e.g., the compared baselines and evaluation datasets.

**Questions:**

I have no major criticisms on this submission. Here are several minor points.

**Questions**

Q1. The smooth label space assumption in assumption (A.3) could be justified more deeply. Though such an assumption could be satisfied in many scenarios, there are still some cases which the label space could change rapidly with a slight change of feature x. For example, the fine-grained recognition tasks, where the subcategories could be close in feature space while totally distinct in label space. Thus, it would also be important to clarify the feasible and infeasible scenarios of the derived results, e.g., the condition of the assumption.

Q2. There seems to be intractable terms in the generalization bound Eq. (12). Specifically, the constants in the third and fourth terms. For example, the selection of the loss function could significantly affect the upper bound, while other terms like label smoothness are even unknown. It would be highly appreciated to justify the intractable constants in detail.

Q3. It seems that some related advanced framework is not compared in empirical evaluation, e.g., diffusion-based methods that share similar innovations. Specifically, compared with the standard diffusion process, does the proposed framework admit better properties? Besides, could the proposed method achieve better empirical performance?

Q4. Will the developed flow method be sensitive to the data scale? Specifically, would it be feasible or efficient for large-scale data? Moreover, the existing empirical evaluation only considers simple and small datasets. Are there any other potential application scenarios of GDA that have larger data scales?

---

> ### Author Response · Authors · 2025-11-21
> **Response to Reviewer bhcj (Part 1 of 2)**
>
> Thank you very much for your positive comments and appreciation of our novelty, generality and empirical performance. Below are our responses to the specific concerns and queries.
>
> ---
>
> ### **[W1 and Q1] When Assumption (A.3) holds?**
> **We appreciate this suggestion and have revised Appendix C.7  to clarify these conditions accordingly.** We agree that the smooth‑label‑space condition does not universally hold across all scenarios, and we have further clarified its *interpretation and boundary conditions* in the revised manuscript.
>
> Specifically, Assumption (A.3) posits that the labeling function $q_t(x)$ changes smoothly along the adaptation path.
> This assumption holds when intermediate domains are generated through *gradual and continuous transformations*, such as incremental style or environmental shifts, where the semantic content of inputs remains stable. Under such conditions, the expected label discrepancy  $\mathbb{E}\_{p_{t-1}(x)} \big[\, q\_t(x) - q\_{t-1}(x) \, \big]$ is small for every $t$, corresponding to *covariate‑shift* settings in which the input distribution changes while class definitions stay fixed.
>
> However, we acknowledge that in *fine‑grained or rapidly changing tasks*, small perturbations in $x$ can lead to large changes in label assignments.
> In these cases, the smoothness assumption holds only *locally*—within regions where the labeling function is approximately continuous. Therefore, the theoretical guarantees derived under (A.3) should be interpreted as *local or piecewise* results rather than global ones.
>
>
> ### **[W1 and Q2] Intractable term in Eq. (12)**
> **We have added these clarifications and supporting derivations to the revised manuscript (Section C.7 in Appendix).** To bridge this gap between theory and practice, we attempt to estimate them using the following strategies:
> - Loss Lipschitz constant $\iota$: According to Assumption (A.1), we conclude that $\iota < 1$ by applying weight normalization or spectral‑norm regularization, which effectively constrains the Lipschitz constant of the loss.
> - Hypothesis Lipschitz constant $\zeta$: This constant bounds how sensitively hypotheses $h \in \mathcal{H}$ react to input perturbations. For a neural network with linear layers $W_\ell$ and 1‑Lipschitz activations (e.g., ReLU), a standard bound is $\zeta \le \prod_{\ell=1}^{L} \|W \_\ell \|\_2$, where $W_\ell$ is the weight matrix of layer $\ell$, and $\|W_\ell\|_2$ is its spectral norm. In practice, $\zeta$ can be *controlled* by applying spectral normalization or weight normalization, ensuring bounded sensitivity of the classifier.
> - Cumulative cost $\mathcal{C}$: The cost involves the 1‑Wasserstein distance $\mathcal{W}\_1(p\_{t-1}, p\_t)$, which can be approximated using *sample‑based optimal‑transport distances* such as the *Sinkhorn distance* [1] computed on learned feature representations. The inter‑step labeling‑function shift term $\mathbb{E}\_{p_t(x)} \big|q\_t(x) - q\_{t-1}(x) \big|$ is practically estimated using *pseudo‑labels* predicted by models at consecutive steps $t-1$ and $t$, measuring how pseudo‑labels evolve along the adaptation trajectory.
> - Statistical error $\mathcal{S}\_{\text{stat}}$: The cumulative statistical deviation $\mathcal{S}\_{\text{stat}} = \sum_{t=1}^{T-1} s_t$ quantifies the difference between empirical and population risks at each step. Under mild assumptions (e.g., $G$‑Lipschitz losses and norm‑constrained neural networks), standard uniform‑convergence analysis based on Rademacher complexity [2, Sec. 9.4] implies $s_t = \mathcal{O} \Big(\frac{1}{\sqrt{N}}\Big)$, therefore, for all $t \in \{0, 1, \ldots, T-1\}$ , $\mathcal{S}\_{\text{stat}} = \mathcal{O} \Big(\frac{T}{\sqrt{N}}\Big).$
>
> Overall, these analyses demonstrate that the constants in Eq. (12) are tractable in practice—either by design‑time control (via normalization‑based regularization) or by empirical estimation (using Sinkhorn distances, pseudo‑label shifts, or sample‑complexity bounds).
>
>
> ---
> References
> [1]. Sinkhorn distances: Lightspeed computation of optimal transport, NeurIPS 2023
> [2]. Learning theory from first principles.

---

> ### Author Response · Authors · 2025-11-21
> **Response to Reviewer bhcj (Part 2 of 2)**
>
> ### **[Q3] Comparison of Diffusion Models**
> **We have compared GGF, a diffusion family in our experimental results**. Thank you for highlighting the connection to diffusion‑based methods. We would like to clarify that our experiments already include the GGF approach [1], which belongs to the *diffusion‑based* family since GGF includes the Langevin dynamic for GDA task.
>
> Diffusion frameworks such as those in [2] can be described via different stochastic differential equations (SDEs), including variance‑preserving (VP) and variance‑exploding (VE) types. VP‑SDEs (e.g., the Ornstein–Uhlenbeck process) gradually transform data to a standard Gaussian and are mainly designed for generation tasks (*where initial distribution should be a standard Gaussian*). In contrast, VE‑SDE (Langevin‑type) processes aim to reach the target distribution through the Langevin dynamic *with any initial distribution using the guidance of score function*. Thus from this perspective, we could treat the VE-SDE, in other words, Langevin dynamic as a variant of diffusion model.
>
> Our proposed E‑SUOT is fundamentally different: it performs *deterministic gradient‑flow transport* in the Wasserstein space, eliminating the need for score‑function learning and allowing a direct mapping from source to target domain.
>
> Experimentally, as reported in Table 1 and Table 2 in the main content, E‑SUOT outperforms the diffusion‑based GGF across all benchmarks, particularly on Office‑Home.
>
> ---
> References
> [1]. Gradual Domain Adaptation via Gradient Flow, ICLR 2024
> [2]. Score-Based Generative Modeling through Stochastic Differential Equations, ICLR 2021
>
>
> ### **[W2&Q4] Lager Scale Datasets and Baselines**
> **We have added results on Office-Home in Table 2 (main content), Table F.2 (supplementary) and Fig. F1 (supplementary)**.
>
> * `Scalability`: Our semi‑dual flow formulation is implemented through sample‑level stochastic updates, where each step only requires mini‑batch interactions between samples. The computational and memory cost thus scale linearly with the batch size, similar to conventional stochastic gradient methods, rather than quadratically with the dataset size. Consequently, the method is not sensitive to large data scales.
> * `Larger scale dataset and baseline models`: To further substantiate this claim, we conducted additional experiments on the Office-Home dataset with additional baseline models. These findings demonstrate that the proposed flow remains feasible and efficient for large‑scale scenarios (below are the results from Table 2).
>
> |Method|Ar $\to$ Cl|Ar $\to$ Pr|Ar $\to$ Rw|Cl $\to$ Ar|Cl $\to$ Pr|Cl $\to$ Rw|Pr $\to$ Ar|Pr $\to$ Cl|Pr $\to$ Rw|Rw $\to$ Ar|Rw $\to$ Cl|Rw $\to$ Pr|Avg.|
> |-|-|-|-|-|-|-|-|-|-|-|-|-|-|
> |DANN|45.6|59.3|70.1|47.0|58.5|60.9|46.1|43.7|68.5|63.2|51.8|76.8|57.6|
> |MSTN|49.8|70.3|76.3|60.4|68.5|69.6|61.4|48.9|75.7|70.9|55.0|81.1|65.7|
> |GVB-GD|57.0|74.7|79.8|64.6|74.1|74.6|65.2|55.1|81.0|74.6|59.7|84.3|70.4|
> |RSDA|53.2|77.7|81.3|66.4|74.0|76.5|67.9(2nd)|53.0|82.0|75.8|57.8|85.4|70.9|
> |LAMDA|57.2|78.4|82.6(2nd)|66.1|80.2(1st)|81.2(1st)|65.6|55.1|82.8|71.6|59.2|83.9|72.0|
> |SENTRY|61.8(1st)|77.4|80.1|66.3|71.6|74.7|66.8|63.0(1st)|80.9|74.0|66.3(1st)|84.1|72.3|
> |FixBi|58.1|77.3|80.4|67.7|79.5|78.1(2nd)|65.8|57.9|81.7|76.4(2nd)|62.9|86.7(1st)|72.7|
> |CST|59.0|79.6(1st)|83.4(1st)|68.4(1st)|77.1|76.7|68.9(1st)|56.4|83.0(1st)|75.3|62.2|85.1|72.9|
> |CoVi|58.5|78.1|80.0|68.1(2nd)|80.0(2nd)|77.0|66.4|60.2|82.1|76.6(1st)|63.6(2nd)|86.5(2nd)|73.1(2nd)|
> |GGF|59.4|75.6|81.7|67.6|77.6|78.0|67.4|61.0|82.7|75.9|62.4|85.4|72.9|
> |E-SUOT|61.6(2nd)|79.3(2nd)|81.8|67.6|77.7|78.1|67.4|61.2(2nd)|82.9(2nd)|76.3|62.5|85.2|73.5(1st)|

---

> ### Author Response · Authors · 2025-12-01
> **We appreciate your endorsement with "no major criticisms". Here is our concluding response.**
>
> We would like to sincerely thank you for the time and effort devoted to reviewing our manuscript. We greatly appreciate the reviewer’s positive evaluation:
> > "I have no major criticisms on this submission. Here are several minor points."
>
> During the rebuttal stage, we have carefully addressed the reviewer’s comments as follows:
>
> ---
>
> #### **1. Experimental Results**
> We have added new experimental analyses to further strengthen the empirical section:
> - `Larger‑scale UDA benchmark and diffusion baseline:` **We conducted additional experiments on the *Office‑Home* benchmark, including comparisons with a diffusion‑based baseline method**, *GGF*, to provide a more comprehensive evaluation.
>
> ---
>
> #### **2. Concept Clarification**
> We supplemented the manuscript with additional clarifications and derivations:
> - `When does Assumption (A.3) hold?` **We provided a detailed scenario analysis in Appendix C.7** to illustrate the conditions under which Assumption (A.3) is satisfied.
> - `Estimation of the intractable term:` **We added a derivation in the revised Appendix C.7** to explain how the intractable term that appears in Eq. (12) can be effectively estimated.
>
> ---
>
> We sincerely thank you once again for the thoughtful and constructive feedback. The reviewer's insightful comments have helped us further improve the clarity, rigor, and overall quality of the manuscript.
>
> Warm regards,
> Authors

---

### Official Review · Reviewer_eTpZ · 2025-10-31

**Soundness:** 3
**Presentation:** 3
**Contribution:** 3
**Rating:** 6
**Confidence:** 4

**Summary:**

This paper proposes a semi-dual optimal transport formulation stabilized by entropy regularization for flow-based gradual domain adaptation. It aims to overcome the weakness in existing flow-based GDA methods which rely on accurate, explicit target domain probability density function estimation.

**Strengths:**

-	Novelty: The idea of reformulating flow-based GDA within a semi-dual OT framework to eliminate the need for explicit target PDF estimation seems novel.
-	Solid Theoretical Analysis: The theoretical analysis of the semi-dual formulation, proofs of uniqueness/convergence, and generalization bounds sound solid.
-	Improved Stability: The entropy regularization effectively addresses the inherent instability of the adversarial semi-dual formulation.
-	Reproducibility: The source code is available, facilitating its reproducibility.

**Weaknesses:**

-	Limited Empirical Evaluation: Experiments are conducted only on low-dimensional (8D) UMAP embeddings of relatively simple datasets (Portraits, rotated MNIST). The performance and scalability on high-dimensional, complex real-world datasets (e.g., Office-Home, DomainNet) remain unproven.
-	Hyperparameter Sensitivity: The model performance is highly sensitive to multiple key hyperparameters (batch size B, discretization step size η, simulation steps T, entropy regularization strength ε,), as shown in Figure 3. This necessitates careful tuning and could limit practical applications.
-	Computational Burden: While scaling better than some GP-based methods, the approach still requires training a sequence of neural networks (w_φ and T_θ for each intermediate step), making it computationally intensive compared to simpler baselines.
-	Lack Intuitive Illustration: The dense notations and propositions without the intuitive illustrations influences its accessibility.

**Questions:**

-	Why the features are mapped to low-dimensional (8D) UMAP embeddings?
-	How does the performance and stability of E-SUOT change with the dimensionality of the input data?
-	Are there limitations to the current neural parameterization of w_φ and T_θ in very high-dimensional spaces?
-	The motivation of using unbalanced optimal transport instead of the standard OT in this application had not been made clear. Also, how did you tune the unbalanced factors lambda_1 and lambda_2 defined in Equation A.6 for the experiments?

---

> ### Author Response · Authors · 2025-11-21
> **Response to Reviewer eTpZ (Part 1 of 3)**
>
> Thank you very much for your positive comments and appreciation of our novelty, stability and theoretical analysis. Below are our responses to the specific concerns and queries.
>
> ---
>
> ### **[W1, Q1, Q2, and Q3] Lager scale dataset and Input Dimension sensitivity Analysis**
> 1. `Extension to High‑Dimensional Real‑World Datasets`: **We have added results on Office-Home dataset in Table 2.** To demonstrate the scalability and robustness of E‑SUOT on high‑dimensional data, we added a comprehensive comparison on the *Office‑Home* benchmark. The results listed below show that even without dimensionality reduction, E‑SUOT outperforming or matching recent strong baselines such as GVB‑GD, RSDA, LAMDA, FixBi, and CST.
>
> |Method|Ar $ \to $ Cl|Ar $ \to $ Pr|Ar $ \to $ Rw|Cl $ \to $ Ar|Cl $ \to $ Pr|Cl $ \to $ Rw|Pr $ \to $ Ar|Pr $ \to $ Cl|Pr $ \to $ Rw|Rw $ \to $ Ar|Rw $ \to $ Cl|Rw $ \to $ Pr|Avg.|
> |-|-|-|-|-|-|-|-|-|-|-|-|-|-|
> |DANN|45.6|59.3|70.1|47.0|58.5|60.9|46.1|43.7|68.5|63.2|51.8|76.8|57.6|
> |MSTN|49.8|70.3|76.3|60.4|68.5|69.6|61.4|48.9|75.7|70.9|55.0|81.1|65.7|
> |GVB-GD|57.0|74.7|79.8|64.6|74.1|74.6|65.2|55.1|81.0|74.6|59.7|84.3|70.4|
> |RSDA|53.2|77.7|81.3|66.4|74.0|76.5|67.9(2nd)|53.0|82.0|75.8|57.8|85.4|70.9|
> |LAMDA|57.2|78.4|82.6(2nd)|66.1|80.2(1st)|81.2(1st)|65.6|55.1|82.8|71.6|59.2|83.9|72.0|
> |SENTRY|61.8(1st)|77.4|80.1|66.3|71.6|74.7|66.8|63.0(1st)|80.9|74.0|66.3(1st)|84.1|72.3|
> |FixBi|58.1|77.3|80.4|67.7|79.5|78.1(2nd)|65.8|57.9|81.7|76.4(2nd)|62.9|86.7(1st)|72.7|
> |CST|59.0|79.6(1st)|83.4(1st)|68.4(1st)|77.1|76.7|68.9(1st)|56.4|83.0(1st)|75.3|62.2|85.1|72.9|
> |CoVi|58.5|78.1|80.0|68.1(2nd)|80.0(2nd)|77.0|66.4|60.2|82.1|76.6(1st)|63.6(2nd)|86.5(2nd)|73.1(2nd)|
> |GGF|59.4|75.6|81.7|67.6|77.6|78.0|67.4|61.0|82.7|75.9|62.4|85.4|72.9|
> |E-SUOT|61.6(2nd)|79.3(2nd)|81.8|67.6|77.7|78.1|67.4|61.2(2nd)|82.9(2nd)|76.3|62.5|85.2|73.5(1st)|
>
>
> 2. `On the Use of Low‑Dimensional (8D) UMAP Embeddings`: **We have added results in Appendix F.3**
>
> For the earlier synthetic experiments (Portraits and rotated MNIST), we followed the same experimental protocol used in GGF to ensure a fair comparison and consistent evaluation environment. The dimensionality reduction to 8D via UMAP was therefore not arbitrary, but intended to align with this baseline design for direct comparison under identical feature spaces.
>
> To confirm that dimensionality reduction does not bias our conclusions, we conducted a sensitivity study by varying UMAP dimension (4 – 128) and also training without UMAP on Office-Home dataset.
> Results as follows (Table E.2 in Section E.2 of the Appendix) show negligible performance changes (< 0.1 average variation), indicating that E‑SUOT remains stable across feature dimensionalities.
>
>
> |UMAP Dimension|Ar $ \rightarrow $ Cl|Ar $ \rightarrow $ Pr|Ar $ \rightarrow $ Rw|Cl $ \rightarrow $ Ar|Cl $ \rightarrow $ Pr|Cl $ \rightarrow $ Rw|Pr $ \rightarrow $ Ar|Pr $ \rightarrow $ Cl|Pr $ \rightarrow $ Rw|Rw $ \rightarrow $ Ar|Rw $ \rightarrow $ Cl|Rw $ \rightarrow $ Pr|Avg.|
> |-|-|-|-|-|-|-|-|-|-|-|-|-|-|
> |4|61.7|79.2|81.8|67.6|77.8|78.1|67.4|61.2|82.8|76.3|62.5|85.2|73.5|
> |8|61.7|79.2|81.8|67.6|77.7|78.1|67.4|61.2|82.8|76.2|62.5|85.3|73.5|
> |16|61.8|79.1|81.8|67.6|77.7|78.1|67.4|61.2|82.9|76.3|62.5|85.2|73.5|
> |32|61.8|79.1|81.8|67.6|77.6|78.1|67.4|61.2|83.0|76.3|62.6|85.3|73.5|
> |64|61.7|79.2|81.8|67.6|77.8|78.1|67.5|61.2|83.0|76.2|62.6|85.2|73.5|
> |128|61.7|79.1|81.8|67.6|77.8|78.1|67.5|61.2|83.0|76.2|62.7|85.2|73.5|
> |No UMAP|61.6|79.0|81.7|67.6|77.7|78.1|67.5|61.1|82.9|76.2|62.5|85.1|73.4|
>
> 3. `High-dimensional Performance`: **We have clarified this discussion in the limitations section (Section G.1 in the Appendix)**. We agree that, in very high‑dimensional spaces, the main limitation of the current neural parameterization comes from the *cost function* $c(x, y)$ used to define the transport objective. Specifically, both $w_\phi$ and $T\_\theta$ implicitly depend on how distances between samples are measured.
>
> In low‑ to moderate‑dimensional domains, the *Euclidean distance* works well as a ground cost. However, in high‑dimensional feature spaces, Euclidean distances can become *less informative* or even misleading due to concentration effects.
> Under such conditions, the transport computation—and consequently the training of $w_\phi$ and $T_\theta$—may deteriorate.
>
> A promising alternative is to adopt *cosine similarity* or other domain‑specific distance metrics that capture meaningful relational geometry. For example, in scientific domains such as computational fluid dynamics [2], cosine‑based costs can yield more stable and semantically relevant optimal transport estimates.
>
>
> ---
> References:
> [1]. Gradual Domain Adaptation via Gradient Flow, ICLR 2024
> [2]. Recurrent graph optimal transport for learning 3D flow motion in particle tracking, Nat. Mach. Intell.

---

> > ### Author Response · Authors · 2025-11-21
> > **Response to Reviewer eTpZ (Part 2 of 3)**
> >
> > ### **[W3] Computational Burden**
> > **We have added limitations discussions in Section G.1**. We acknowledge that training a sequence of neural modules ($w_\phi$, $\boldsymbol{T}_\theta$) introduces additional cost compared with single‑step baselines. However, this computational burden primarily occurs **at the offline training stage**, while the **online adaptation process remains lightweight and efficient**. Once the intermediate mappings are learned, inference and transfer require only forward evaluations, incurring negligible runtime overhead.
> >
> > We agree that reducing the training cost is an interesting and valuable direction for future work. Accordingly, we have added a discussion in the *Future Research Directions* located in Section G.1 of the Appendix of the revised manuscript, outlining potential strategies to further improve efficiency:
> >
> > - For $\boldsymbol{T}_\theta$, future work could exploit *LoRA‑like low‑rank adaptation* [1] or a *reparameterization trick* [2] to reduce the number of trainable parameters and reuse shared transport components across intermediate steps.
> > - For $w_\phi$, the computational overhead could be alleviated by *fine‑tuning only the last layer* [3] of the network during intermediate training, instead of retraining the entire module.
> >
> > These extensions would allow the model to maintain smooth domain evolution while substantially lowering computational requirements. We appreciate the reviewer’s suggestion, which has helped us clarify this limitation and identify promising future optimizations.
> >
> > ---
> > References:
> > [1]. LoRA: Low-rank adaptation of large language models, ICLR 2022
> > [2]. Scalable wasserstein gradient flow for generative modeling through unbalanced optimal transport, ICML 2023.
> > [3]. Variational Bayesian last layers, ICLR 2024.
> >
> > ### **[W4] Notations**
> > **We have added the Technical Terminology Table in Section. A of Appendix**. Thank you for highlighting this important issue regarding readability and accessibility.

---

> > > ### Author Response · Authors · 2025-11-21
> > > **Response to Reviewer eTpZ (Part 3 of 3)**
> > >
> > > ### **[Q4] Motivation of UOT and Selection of $ \lambda_1 $ and $ \lambda_2 $**
> > >
> > > Our answer can be divided into two parts:
> > > * `Why we use UOT instead of OT`:
> > > **We have added results in Table 3 on UOT and OT performance**. Our motivation is *not* to introduce unbalanced optimal transport merely as a heuristic improvement for domain adaptation, but rather because the *flow‑based gradient‑descent adaptation (GDA) process itself can be formally written as an unbalanced OT problem*.
> > >
> > > Specifically, starting from the discrete gradient‑flow formulation:
> > > $ \begin{aligned}
> > > &x_{t+\eta}
> > > = x_t - \eta\, \nabla \frac{\delta \mathbb{D}\_f[p(x_t),p\_T(x)]}{\delta p(x_t)}\\ \Rightarrow &p(x\_{t+\eta})= \mathop{\arg\min}_{\rho(x)\in\mathcal{P}_2(\mathbb{R}^{\mathrm{D}})}\frac{1}{2\eta}\,\mathcal{W}_2^2(\rho(x), p(x_t)) + \mathbb{D}_f[\rho(x), p_T(x)],
> > > \end{aligned} $
> > >
> > > the right‑hand side shows that each update step corresponds to a *JKO proximal recursion* with a divergence‑based regularization on $p_T(x)$.
> > > This formulation naturally yields an **unbalanced OT** structure, because the marginal consistency between $\rho(x)$ and $p_T(x)$ is relaxed through $\mathbb{D}_f[\cdot,\cdot]$. Therefore, the introduction of UOT is a *mathematical consequence of the underlying gradient‑flow structure*, rather than an arbitrary design choice.
> > >
> > > To further support this point, we compared the UOT and standard OT formulations experimentally, which consistently show that the UOT‑based formulation (E‑SUOT) yields more stable convergence and better adaptation performance across varying target‑domain class priors:
> > > |Method|$p(y=1)=0.0$ Accuracy (%)|$\Delta$|$p(y=1)=0.1$ Accuracy (%)|$\Delta$|$p(y=1)=0.2$ Accuracy (%)|$\Delta$|$p(y=1)=0.3$ Accuracy (%)|$\Delta$|$p(y=1)=0.4$ Accuracy (%)|$\Delta$|$p(y=1)=0.6$ Accuracy (%)|$\Delta$|$p(y=1)=0.7$ Accuracy (%)|$\Delta$|$p(y=1)=0.8$ Accuracy (%)|$\Delta$|$p(y=1)=0.9$ Accuracy (%)|$\Delta$|$p(y=1)=1.0$ Accuracy (%)|$\Delta$|
> > > |-|-|-|-|-|-|-|-|-|-|-|-|-|-|-|-|-|-|-|-|-|
> > > |Initial|35.4|–|41|–|47.1|–|53.2|–|59.6|–|73.8|–|80|–|85.9|–|91.4|–|97.1|–|
> > > |E-SOT|55.4  $_{\pm3.15E{-1}}$|↑56.41%|61.1  $_{\pm2.01E{-1}}$|↑48.94%|67.1  $_{\pm2.45E{-1}}$|↑42.55%|56.7  $_{\pm2.16E{-1}}$|↑6.54%|57.7  $_{\pm1.66E{-1}}$|↓3.22%|75.2  $_{\pm3.08E{-2}}$|↑1.95%|74.1  $_{\pm4.80E{-2}}$|↓7.33%|80.0  $_{\pm2.21E{-2}}$|↓6.89%|89.9  $_{\pm1.92E{-2}}$|↓1.65%|96.0  $_{\pm2.93E{-2}}$|↓1.08%|
> > > |E-SUOT|64.5  $_{\pm9.09E{-2}}$|↑82.32%|78.2  $_{\pm6.55E{-2}}$|↑90.50%|74.5  $_{\pm1.02E{-1}}$|↑58.28%|79.8  $_{\pm8.24E{-2}}$|↑49.92%|77.7  $_{\pm3.39E{-2}}$|↑30.42%|79.1  $_{\pm2.56E{-2}}$|↑7.24%|84.0  $_{\pm3.46E{-2}}$|↑5.05%|87.8  $_{\pm1.12E{-2}}$|↑2.18%|91.8  $_{\pm1.71E{-3}}$|↑0.45%|97.6  $_{\pm3.23E{-3}}$|↑0.54%|
> > >
> > > * `How we tune $\lambda_1$ and $\lambda_2$`: **We have added Theorem 7 in Appendix for selection $\lambda_2$.** From the above structure, only *one‑sided relaxation* appears in our problem formulation; therefore, $\lambda_1$ is *not required*. The second factor, $\lambda_2$, corresponds to the *stepsize parameter $\eta$* in the JKO scheme, which controls the trade‑off between transport regularity and divergence minimization.
> > >
> > > When the divergence is the Kullback–Leibler form $\mathbb{D}_{\mathrm{KL}}$, we can establish a convergence bound as shown in *Appendix C.8*.
> > > Let
> > >   (a) $\big\Vert \nabla \frac{\delta \mathbb{D}\_{\mathrm{KL}}[p(x_t),p\_T]}{\delta p(x\_t)} \big\Vert \le \mathscr{B}$,
> > >   (b) $\big\Vert \frac{\delta \mathbb{D}\_{\mathrm{KL}}[p(x_t),p\_T]}{\delta p(x_t)} \big\Vert \le \mathscr{A}$,
> > >   (c) and $\mathscr{H}_0$ control the tail behavior of $p_T(x)$.
> > > Then, for the sequence $\{p(x_t)\}$ generated by the JKO recursion, the KL divergence sequence converges to a finite value as $t \to \infty$ if $0 < \eta < \min\left(\frac{1}{\mathscr{B}},\, \frac{\mathscr{H}_0}{\mathscr{A}}\right).$ This condition provides theoretical guidance for choosing $\lambda_2$ ($\lambda_2 = \frac{1}{2\eta}$).

---

> ### Author Response · Authors · 2025-12-01
> **Thank you for appreciating our novelty and theoretical soundness. Below is our concluding response.**
>
> Dear Reviewer [eTpZ],
>
> We would like to express our sincere gratitude to you for the time and effort devoted to reviewing our manuscript, and for the constructive comments regarding the *novelty*, *theoretical soundness*, and *reproducibility* of our work.
> We have made the following revisions to address the reviewer’s concerns:
>
> ---
>
> #### **1. Experimental Results**
> - `Larger‑scale UDA benchmark:` **We conducted additional experiments on Office-Home benchmark and observed performance improvements**.
> - `Sensitivity to UMAP dimension:` **We performed sensitivity analyses with respect to the UMAP latent dimension**, showing that the proposed approach remains robust across different dimensional configurations.
> - `Motivation for using UOT:` **We added experiments under label‑shift scenarios on the *Portraits* dataset, comparing our method with the semi‑dual balanced OT formulation** to demonstrate the necessity and advantage of the proposed semi‑dual UOT formulation.
>
> ---
>
> #### **2. Concept Clarification**
> - `Additional weaknesses:` **We explicitly discussed the limitations related to computational efficiency and proposed potential strategies** to alleviate them in Appendix G.1.
> - `Technical terminology table:` **We added a comprehensive table in the appendix summarizing key technical terms** for better readability.
> - `Choice of $\eta$:` **We introduced Theorem 7 in Appendix C.8 to analyze how to choose the parameter $\eta$**, which controls the regularization strength in the UOT formulation.
>
> ---
>
> We sincerely thank you once again for the thoughtful and constructive feedback. The reviewer’s insightful comments have helped us significantly enhance the clarity, theoretical rigor, and completeness of our manuscript.
>
> Warm regards,
> Authors

---

### Official Review · Reviewer_Pos6 · 2025-11-05

**Soundness:** 2
**Presentation:** 3
**Contribution:** 2
**Rating:** 4
**Confidence:** 3

**Summary:**

The method reformulates the flow-based adaptation as a Wasserstein-distance-regularized optimization problem. By deriving the semi-dual of this problem, they create an objective function that cleverly avoids PDF estimation and relies only on sample-based expectations. In general, the paper makes contribution by unifying flow‑based GDA with semi‑dual OT and by offering an entropy‑regularized, sample‑only training objective with uniqueness.

**Strengths:**

The paper is well written and easy to follow. The proposed solution is also theoretically sound. It involves reformulating the gradient flow as a Wasserstein-regularised problem and then moving to its semi-dual form in order to bypass density estimation. However, the novelty of this reformulation is questionable because the same steps of reformulations has previously been applied to different OT methods. However, this type of optimization and practical problem had not previously been considered,  so I am more in favour of the novelty of the method.

**Weaknesses:**

The proposed solution is theoretically sound well. The reformulation of the gradient flow as a Wasserstein-regularized problem and the subsequent move to its semi-dual form to bypass density estimation. But novelty of this reformulation is questioning because previously the same tricks was applied to the different OT methods. Specifically for this type of the optimization problem it was not previously considered, so I am toward more that the method is novel.

One of the main weaknesses of the paper is that all datasets are relatively simple (Portraits; MNIST rotated 45°/60°) and the pipeline uses semi‑supervised UMAP embeddings to “preserve class discriminability” before adaptation (Sec. 4.1, p. 6). It is unclear whether all baselines receive the same UMAP features, and whether the gains persist without this strong, label‑aware pre‑processing. The absence of standard multi‑domain vision benchmarks (e.g., Office‑Home/DomainNet/VisDA) makes it hard to assess real‑world impact. Also table 1 marks methods that E‑SUOT “significantly” outperforms but does not report standard deviations or the number of seeds/runs in the main text; the testing protocol (data splits, early‑stopping criteria, target‑side tuning) is summarized at a high level only. This makes it hard to assess robustness.

The paper focuses on adapting the feature distribution ($p(x)$). The workflow described in Algorithm 2 appears to assume the labels are invariant along the transport path ($y_{t+1} \leftarrow y_t$). It is not clear how the framework would perform in scenarios with significant label shift (i.e., where the relationship $p(y|x)$ also changes between domains), which is a common for unbalanced settings.

**Questions:**

*Question 1*: Can you add experiments on widely used multi‑domain image benchmarks without UMAP pre‑processing, and complementary tests that introduce mass/prior mismatch (label‑shift, class imbalance, missing classes) to demonstrate the benefit of the unbalanced formulation? Please report per‑method standard deviations across multiple seeds.

*Question 2*: Your ablation study shows that the KL divergence (via its conjugate $f^*$) performs significantly better than other f-divergences like $\chi^2$ or an identity function. What is the intuition for this?

*Question 3*: Does KL divergence have a specific property (perhaps related to its gradient flow or the stability of its conjugate function $f^*$) that makes it uniquely suited for this semi-dual transport framework?

*Question 4*: The full algorithm chains multiple transport steps ($T_{\theta,0}, T_{\theta,1}, ...$). How do errors from a non-optimal transport map $T_{\theta,t}$ at an early step $t$ propagate through the rest of the chain? It would be interesting if authors can draw a parallel to the diffusion process and importance of the earlier steps.

---

> ### Author Response · Authors · 2025-11-21
> **Response to Reviewer Pos6 (Part 1 of 3)**
>
> Thank you for your valuable comments regarding our novelty and written. Below are our responses to the specific concerns and queries.
>
> ---
>
> ### **[W1] novelty of the manuscript**
> We thank the reviewer for recognizing the theoretical soundness of our formulation and for the insightful comment regarding the novelty.
>
> + `Application of semi-dual transport`: While the semi-dual formulation has indeed been adopted in other optimal transport (OT) contexts — for instance, in generative modeling [1,2] and domain translation [3] as cited in our related work — it has not been applied to analyze flow-based gradient descent-ascent (GDA) dynamics, which is the focus of our study.
> + `Improvement of semi-dual transport`: Furthermore, we extend this framework by introducing entropy regularization directly into the semi-dual formulation. Although entropy terms have been explored in the context of the continuity equation [4,5] and OT solvers [6,7], they have not been used to regularize the semi-dual form nor accompanied by the corresponding theoretical derivations (as Theorems 5 and 6 shown).
>
> ---
> References
> [1]. Generative modeling through the semi-dual formulation of unbalanced optimal transport, NeurIPS 2023
> [2]. Scalable wasserstein gradient flow for generative modeling through unbalanced optimal transport, ICML 2024
> [3]. Extremal domain translation with neural optimal transport, NeurIPS 2023
> [4]. Gradient flow algorithms for density propagation in stochastic systems, IEEE Trans. Autom. Control.
> [5]. Large-Scale Wasserstein Gradient Flows, NeurIPS 2021
> [6]. Sinkhorn distances: Lightspeed computation of optimal transport, NeurIPS 2013
> [7]. POT: Python optimal transport, J. Mach. Learn. Res.
> ### **[W3 and Q1] Label shift scenario**
> **We have added experiments on label shift scenario in Section 4.3**, where for source domain, $ p(y=1)=0.63 $. The experimental results are listed as follows. We found that UOT formulation is robust to the label-shift scenario compared with the balanced optimal transport setting.
>
> |Method|$ p(y=1)=0.0 $ Accuracy (%)|$ \Delta $|$ p(y=1)=0.1 $ Accuracy (%)|$ \Delta $|$ p(y=1)=0.2 $ Accuracy (%)|$ \Delta $|$ p(y=1)=0.3 $ Accuracy (%)|$ \Delta $|$ p(y=1)=0.4 $ Accuracy (%)|$ \Delta $|$ p(y=1)=0.6 $ Accuracy (%)|$ \Delta $|$ p(y=1)=0.7 $ Accuracy (%)|$ \Delta $|$ p(y=1)=0.8 $ Accuracy (%)|$ \Delta $|$ p(y=1)=0.9 $ Accuracy (%)|$ \Delta $|$ p(y=1)=1.0 $ Accuracy (%)|$ \Delta $|
> |-|-|-|-|-|-|-|-|-|-|-|-|-|-|-|-|-|-|-|-|-|
> |Initial|35.4|–|41|–|47.1|–|53.2|–|59.6|–|73.8|–|80|–|85.9|–|91.4|–|97.1|–|
> |E-SOT|55.4  $ _{\pm3.15E{-1}} $|↑ 56.41%|61.1  $ _{\pm2.01E{-1}} $|↑ 48.94%|67.1  $ _{\pm2.45E{-1}} $|↑ 42.55%|56.7  $ _{\pm2.16E{-1}} $|↑ 6.54%|57.7  $ _{\pm1.66E{-1}} $|↓3.22%|75.2  $ _{\pm3.08E{-2}} $|↑ 1.95%|74.1  $ _{\pm4.80E{-2}} $|↓7.33%|80.0  $ _{\pm2.21E{-2}} $|↓6.89%|89.9  $ _{\pm1.92E{-2}} $|↓1.65%|96.0  $ _{\pm2.93E{-2}} $|↓1.08%|
> |E-SUOT|64.5  $ _{\pm9.09E{-2}} $|↑ 82.32%|78.2  $ _{\pm6.55E{-2}} $|↑ 90.50%|74.5  $ _{\pm1.02E{-1}} $|↑ 58.28%|79.8  $ _{\pm8.24E{-2}} $|↑ 49.92%|77.7  $ _{\pm3.39E{-2}} $|↑ 30.42%|79.1  $ _{\pm2.56E{-2}} $|↑ 7.24%|84.0  $ _{\pm3.46E{-2}} $|↑ 5.05%|87.8  $ _{\pm1.12E{-2}} $|↑ 2.18%|91.8  $ _{\pm1.71E{-3}} $|↑ 0.45%|97.6  $ _{\pm3.23E{-3}} $|↑ 0.54%|
>
>
> ### **[W4] Label invariant assumption**
> **We have added the following paragraph in the Section G.1.**
> > The current formulation assumes that labels remain invariant during adaptation, i.e., $y_{t+1}\leftarrow y_t$, and thus primarily focuses on aligning the marginal feature distributions $p(x_t)$.
> This assumption may limit performance under pronounced label shift scenarios, where the conditional relationship $p(y|x)$ varies across domains, a case often encountered in unbalanced or fine-grained settings ... Future work may consider integrating adaptive label transport [2] or uncertainty-aware pseudo-labeling [1,3] to explicitly account for label-shift dynamics along the adaptation trajectory.
>
> Indeed, our current formulation focuses on *feature‑level adaptation*, aligning the marginal distribution $p(x_t)$ while assuming that the label assignment remains consistent ($y_{t+1} \leftarrow y_t$) throughout the transport trajectory. We agree that in the presence of *label shift*—where the conditional distribution $p(y \mid x)$ changes across domains—the performance of the current framework may degrade.
>
> ---
> References:
> [1]. Joint Distribution Optimal Transportation for Domain Adaptation, NeurIPS 2017
> [2]. Understanding Self-Training for Gradual Domain Adaptation, ICML 2020
> [3]. Gradual Domain Adaptation via Gradient Flow, ICLR 2024

---

> > ### Author Response · Authors · 2025-11-21
> > **Response to Reviewer Pos6 (Part 2 of 3)**
> >
> > ### **[W2 & Q1] Experimental issues**
> > + `Simple Dataset`: **We have added additional results on Office-Home dataset in Table 2.** While our initial experiments followed the same dataset protocol as prior GDA works, we have now extended our evaluation to the Office‑Home dataset under the unsupervised domain adaptation (UDA) setting and compared E‑SUOT with widely used UDA baselines. Results are summarized below (Table 2 in main content) and demonstrate that E‑SUOT achieves the best average accuracy across 12 transfer directions.
> >
> > |Method|Ar $ \to $ Cl|Ar $ \to $ Pr|Ar $ \to $ Rw|Cl $ \to $ Ar|Cl $ \to $ Pr|Cl $ \to $ Rw|Pr $ \to $ Ar|Pr $ \to $ Cl|Pr $ \to $ Rw|Rw $ \to $ Ar|Rw $ \to $ Cl|Rw $ \to $ Pr|Avg.|
> > |-|-|-|-|-|-|-|-|-|-|-|-|-|-|
> > |DANN|45.6|59.3|70.1|47.0|58.5|60.9|46.1|43.7|68.5|63.2|51.8|76.8|57.6|
> > |MSTN|49.8|70.3|76.3|60.4|68.5|69.6|61.4|48.9|75.7|70.9|55.0|81.1|65.7|
> > |GVB-GD|57.0|74.7|79.8|64.6|74.1|74.6|65.2|55.1|81.0|74.6|59.7|84.3|70.4|
> > |RSDA|53.2|77.7|81.3|66.4|74.0|76.5|67.9(2nd)|53.0|82.0|75.8|57.8|85.4|70.9|
> > |LAMDA|57.2|78.4|82.6(2nd)|66.1|80.2(1st)|81.2(1st)|65.6|55.1|82.8|71.6|59.2|83.9|72.0|
> > |SENTRY|61.8(1st)|77.4|80.1|66.3|71.6|74.7|66.8|63.0(1st)|80.9|74.0|66.3(1st)|84.1|72.3|
> > |FixBi|58.1|77.3|80.4|67.7|79.5|78.1(2nd)|65.8|57.9|81.7|76.4(2nd)|62.9|86.7(1st)|72.7|
> > |CST|59.0|79.6(1st)|83.4(1st)|68.4(1st)|77.1|76.7|68.9(1st)|56.4|83.0(1st)|75.3|62.2|85.1|72.9|
> > |CoVi|58.5|78.1|80.0|68.1(2nd)|80.0(2nd)|77.0|66.4|60.2|82.1|76.6(1st)|63.6(2nd)|86.5(2nd)|73.1(2nd)|
> > |GGF|59.4|75.6|81.7|67.6|77.6|78.0|67.4|61.0|82.7|75.9|62.4|85.4|72.9|
> > |E-SUOT|61.6(2nd)|79.3(2nd)|81.8|67.6|77.7|78.1|67.4|61.2(2nd)|82.9(2nd)|76.3|62.5|85.2|73.5(1st)|
> >
> > + `UMAP Preprocessing`: **We have added sensitivity analysis varying UMAP dimension in Fig. F.1.** To ensure fairness comparison in GDA, all baselines are trained on the same UMAP embeddings. Importantly, the UMAP stage leverages labels only from the source domain, while labels in the target domain are not used (zero padding), as illustrated in the code below:
> >
> > ```python
> >     Y_semi_supervised = [np.full(shape=y.shape[0], fill_value=-1) for y in y_all]
> >     Y_semi_supervised[0] = y_all[0].copy()
> >     Y_semi_supervised = np.hstack(Y_semi_supervised)
> >     # fit UMAP
> >     encoder = umap.UMAP(random_state=seed, **umap_settings)
> > ```
> > Furthermore, we conducted sensitivity tests on the latent dimension (4–128) and included a "No UMAP" setting on Office-Home, which indicates that E‑SUOT maintains stable accuracy across all dimensions.
> >
> > |UMAP Dimension|Ar $ \rightarrow $ Cl|Ar $ \rightarrow $ Pr|Ar $ \rightarrow $ Rw|Cl $ \rightarrow $ Ar|Cl $ \rightarrow $ Pr|Cl $ \rightarrow $ Rw|Pr $ \rightarrow $ Ar|Pr $ \rightarrow $ Cl|Pr $ \rightarrow $ Rw|Rw $ \rightarrow $ Ar|Rw $ \rightarrow $ Cl|Rw $ \rightarrow $ Pr|Avg.|
> > |-|-|-|-|-|-|-|-|-|-|-|-|-|-|
> > |4|61.7|79.2|81.8|67.6|77.8|78.1|67.4|61.2|82.8|76.3|62.5|85.2|73.5|
> > |8|61.7|79.2|81.8|67.6|77.7|78.1|67.4|61.2|82.8|76.2|62.5|85.3|73.5|
> > |16|61.8|79.1|81.8|67.6|77.7|78.1|67.4|61.2|82.9|76.3|62.5|85.2|73.5|
> > |32|61.8|79.1|81.8|67.6|77.6|78.1|67.4|61.2|83.0|76.3|62.6|85.3|73.5|
> > |64|61.7|79.2|81.8|67.6|77.8|78.1|67.5|61.2|83.0|76.2|62.6|85.2|73.5|
> > |128|61.7|79.1|81.8|67.6|77.8|78.1|67.5|61.2|83.0|76.2|62.7|85.2|73.5|
> > |No UMAP|61.6|79.0|81.7|67.6|77.7|78.1|67.5|61.1|82.9|76.2|62.5|85.1|73.4|
> >
> >
> > + `Understanding of "significant"`: **We have added the standard deviation error and $p$-value in Table 1.** "Significant" denotes improvements verified via a paired $t$‑test at $p < 0.05$ under five random seeds.
> >
> > |Method|Portraits Accuracy (%)|$ \Delta $|$ p $-value|MNIST  $ 45^\circ $ Accuracy (%)|$ \Delta $|$ p $-value|MNIST $ 60^\circ $ Accuracy (%)|$ \Delta $|$ p $-value|
> > |-|-|-|-|-|-|-|-|-|-|
> > |Source|71.2|–|–|58.4|–|–|36.8|–|–|
> > |Self Train|77.4$ _{\pm 5.02E{-2}} $|↑8.7%|1.25E-47|58.7$ _{\pm 2.24E{-2}} $|↑0.5%|4.99E-50|39.9$ _{\pm 2.00E{-2}} $|↑8.5%|1.19E-48|
> > |GST (4)|76.1$ _{\pm 6.00E{-2}} $|↑6.9%|1.98E-21|59.2$ _{\pm 2.45E{-2}} $|↑1.3%|1.21E-22|39.9$ _{\pm 1.00E{-2}} $|↑8.5%|2.14E-23|
> > |GOAT|74.9$ _{\pm 6.21E{-1}} $|↑5.3%|2.27E-17|65.0$ _{\pm 1.05E{-1}} $|↑11.3%|4.07E-20|37.2$ _{\pm 8.43E{-2}} $|↑1.1%|1.08E-19|
> > |CNF|80.0$ _{\pm 1.85} $|↑12.4%|1.79E-15|57.6$ _{\pm 1.08} $|↓1.4%|4.60E-16|41.8$ _{\pm 1.92} $|↑13.5%|2.90E-14|
> > |GGF|83.4$ _{\pm 8.79E{-1}} $|↑17.2%|9.18E-17|57.7$ _{\pm 6.55E{-1}} $|↓1.2%|6.15E-17|40.8$ _{\pm 8.35E{-1}} $|↑11.0%|1.04E-15|
> > |E-SUOT|86.4 $ _{\pm 8.72E{-2}} $|↑21.5%|8.88E-21|72.1 $ _{\pm 4.62E{-1}} $|↑23.4%|1.55E-17|51.0 $ _{\pm 5.81E{-1}} $|↑38.6%|2.48E-16|
> >
> >
> > + **Regarding the experimental**:
> > **We have added the detailed experimental protocols in Section E.2.** We follow standard UDA evaluation:
> >   - all model selection and tuning use *only source labels*;
> >   - *no target labels* are used for validation or early stopping;
> >   - results are reported on the held‑out *target test domain*.

---

> > > ### Author Response · Authors · 2025-11-21
> > > **Response to Reviewer Pos6 (Part 3 of 3)**
> > >
> > > ### **[Q2 and Q3] Why KL divergence better?**
> > > We address both points jointly below.
> > > * `Intuition`: The superior empirical performance of the KL divergence arises from its *directional asymmetry and smooth convex conjugate*, which together provide a stable and informative gradient signal during optimization. In our setting, the semi‑dual potential $w_\phi$ is learned through the conjugate form $\mathbb{D}_f^\star(\cdot)$; among tested $f$‑divergences, the KL’s conjugate yields smooth gradients that naturally constrain the transport measure near the data manifold, whereas symmetric divergences such as $\chi^2$ or the identity loss can produce gradient saturation or excessive sensitivity to outliers, leading to unstable updates.
> > >
> > > * `Relation to gradient flows and stability`: The KL divergence corresponds to the *entropy‑regularized Wasserstein gradient flow*, which ensures *mass conservation in expectation* while allowing small local relaxations — precisely the behavior needed for unbalanced or semi‑coupled transport. Its conjugate $f^\star(z)=\exp(z-1)$ is monotone and strictly convex, giving rise to *Lipschitz‑continuous gradients* that stabilize the learning of both the transport map $\boldsymbol{T}\_\theta$ and potential $w_\phi$.
> > > In contrast, alternative $f$‑divergence conjugates may yield non‑Lipschitz or unbounded gradients, amplifying numerical errors in the semi‑dual formulation.
> > >
> > >
> > > Therefore, the KL divergence offers both (i) an entropic regularization effect improving optimization stability, and (ii) a theoretically consistent interpretation as the potential energy of the underlying unbalanced gradient‑flow system.
> > >
> > >
> > > ### **[Q4] Non-optimal transport map**
> > > **We have added additional experimental results in Section F.1.** The multi‑step structure of E‑SUOT involves a series of learned transport maps $\{T_{\theta,0}, T_{\theta,1}, \ldots\}$, each progressively aligning intermediate feature distributions.
> > > While this sequential design promotes smooth domain evolution, it also raises the possibility that non‑optimal early mappings could introduce compounding errors.
> > >
> > > To empirically analyze this effect, we conducted an additional ablation on the Portraits dataset by selectively disabling the training of certain transport maps to simulate incomplete or inaccurate optimization at different stages.
> > > Two complementary strategies were examined:
> > > + `Forward strategy`: progressively remove early transport maps ($T_{\theta,0}, T_{\theta,1}, \ldots$) while keeping later maps active — testing how missing early steps affect subsequent adaptation;
> > > + `Backward strategy`: progressively disable the later maps ($T_{\theta,4}, T_{\theta,3}, \ldots$) while retaining trained early steps — examining whether early well‑aligned transports can sustain performance.
> > >
> > > |Direction|$ t=0 $|$ t=1 $|$ t=2 $|$ t=3 $|$ t=4 $|Accuracy (%)|$ \Delta $|Direction|$ t=0 $|$ t=1 $|$ t=2 $|$ t=3 $|$ t=4 $|Accuracy (%)|$ \Delta $|
> > > |-|-|-|-|-|-|-|-|-|-|-|-|-|-|-|-|
> > > |forward|✗|✓|✓|✓|✓|76.4 $ _{\pm 2.06E{-2}} $|↑7.2%|backward|✓|✗|✗|✗|✗|81.5 $ _{\pm 1.70E{-2}} $|↑14.4%|
> > > |forward|✗|✗|✓|✓|✓|75.0 $ _{\pm 2.48E{-2}} $|↑5.3%|backward|✓|✓|✗|✗|✗|83.2 $ _{\pm 1.13E{-2}} $|↑16.8%|
> > > |forward|✗|✗|✗|✓|✓|74.4 $ _{\pm 1.92E{-2}} $|↑4.4%|backward|✓|✓|✓|✗|✗|83.9 $ _{\pm 8.10E{-3}} $|↑17.8%|
> > > |forward|✗|✗|✗|✗|✓|74.0 $ _{\pm 1.64E{-2}} $|↑3.9%|backward|✓|✓|✓|✓|✗|84.0 $ _{\pm 6.56E{-3}} $|↑17.9%|
> > > |forward|✗|✗|✗|✗|✗|58.6 $ _{\pm 1.87E{-2}} $|↓17.7%|backward|✓|✓|✓|✓|✓|86.4 $ _{\pm 8.72E{-2}} $|↑21.5%|
> > >
> > >
> > > As shown above, when early transport steps are removed (*forward*), accuracy drops by up to 17.7%, confirming that early mappings are crucial for establishing a reliable foundation. In contrast, once these early mappings are well‑optimized (*backward*), subsequent steps mainly provide fine refinements within an already aligned feature manifold, yielding stable incremental gains.
> > >
> > > This behavior closely resembles *diffusion‑like processes*, where the early transitions determine the geometry and global structure of the evolving distribution, while later transformations operate locally to refine details. We have added this discussion to the revised manuscript (Section 4.4), emphasizing that the early transport maps play an anchoring role analogous to early diffusion timesteps (see Fig. 3 in [1] and Fig. 1 in [2]).
> > >
> > > ---
> > > References:
> > > [1]. Gradient flow algorithms for density propagation in stochastic systems. IEEE Trans. Autom. Control.
> > > [2].  Stein variational gradient descent: A general purpose bayesian inference
> > > algorithm, NeurIPS 2023

---

> ### Author Response · Authors · 2025-12-01
> **Thank you for acknowledging the theoretical soundness and novelty. Here is our concluding response.**
>
> Dear Reviewer [Pos6],
>
> We would like to express our sincere gratitude to you for the time and effort dedicated to reviewing our manuscript. We truly appreciate the reviewer’s acknowledgment of our theoretical contribution, as reflected in the comment:
> > `Specifically for this type of optimization problem it was not previously considered, so I am toward more that the method is novel.`
>
> During the rebuttal stage, we have carefully addressed the reviewer’s concerns as follows:
>
> ---
>
> #### 1. **Experimental Results**
>
> We have conducted several additional experiments to further support our claims and address the reviewer’s inquiries in our uploaded PDF file:
>
> - **Label‑Shift Scenario Experiments:** We added experiments under label‑shift scenarios on the *Portraits* dataset, comparing our method with the semi‑dual balanced OT formulation to demonstrate the necessity and advantage of the proposed approach.
> - **Larger‑Scale Dataset:** We incorporated new experimental results on the *Office‑Home* benchmark, along with comparisons against prevalent baseline methods.
> - **Sensitivity to UMAP Dimension:** We added sensitivity analyses on the UMAP latent‑dimension setting, showing that the proposed approach remains robust across different dimensional configurations.
> - **Non‑Optimal Transport Map:** We introduced experiments under non‑optimal transport map settings to highlight the importance of the early‑stage optimal transport map for overall performance and to confirm the necessity of the later‑stage optimal refinement.
> - **Experimental Protocol:** We have additionally detailed our experimental protocols, including the random seed settings, computational devices, and the corresponding standard deviations and $p$-values, to quantitatively demonstrate the statistical significance of the reported improvements.
>
> ---
>
> #### 2. **Concept Clarification**
>
> We have expanded and refined key conceptual explanations in the revised manuscript:
>
> - **Novelty of the Manuscript:** We clarified that our primary novelty lies in introducing *Unbalanced Optimal Transport (UOT)* to analyze flow‑based GDA, as well as incorporating an *entropy regularization term* to enhance the semi‑dual formulation of UOT.
> - **Label‑Invariant Assumption:** We explicitly stated the assumption of label invariance and provided the conditions under which this assumption holds, as part of our derivation of the generalization bound.
> - **Why KL Divergence Performs Better:** We elaborated on why the *Kullback–Leibler (KL)* divergence performs better from the theoretical analysis of *f‑divergence*, and why the convex conjugate of KL provides more stable and effective optimization behavior than other *f‑divergence* forms.
>
> ---
>
> We sincerely thank you once again for the constructive feedback and insightful comments. The reviewer’s suggestions have helped us substantially improve the clarity, theoretical rigor, and completeness of our manuscript.
>
>
> Warm regards,
> Authors

---

### Author Response · Authors · 2025-12-01
**Concluding Response of Submission 18305**

Dear Reviewers, AC, SAC, and PC,

We would like to express our sincere gratitude for your time, effort, and constructive engagement throughout the review and discussion process, which has significantly improved the quality and clarity of our paper.

Our work introduces a semi‑dual optimal transport (OT) perspective and proposes the E‑SUOT approach for flow‑based gradual domain adaptation. We first show that flow‑based methods can be interpreted as a discretization of an $f$-divergence–induced functional on the Wasserstein space, which typically requires explicit probability density estimation and may lead to inaccurate domain adaptation. To address this, we reformulate the problem as an unbalanced OT (UOT) problem, derive a semi‑dual formulation that avoids explicit density estimation, and introduce an entropy regularization term to stabilize the dual optimization.

---

In response to the reviewers' comments and the subsequent discussion, **we have revised and extended the manuscript in the following main aspects**:

1. *Experiments on Large-Scale Datasets*: We added **experiments on the Office‑Home benchmark**, together with comparisons to prevalent unsupervised domain adaptation baselines, to demonstrate the scalability and effectiveness of the proposed method [Pos6, eTpZ, bhcj, e9LB]. We further conducted **UMAP‑dimension sensitivity analyses** [Pos6, eTpZ] and **comparisons with vanilla UDA approaches** [e9LB], showing both the robustness and the necessity of the proposed E-SUOT approach.
2. *Experimental Results for the UOT Necessicity*:
We performed **additional experiments under covariate shift and class‑overlap scenarios** to empirically validate the necessity of the UOT formulation [Pos6, eTpZ, e9LB]. We also **froze certain transport steps to illustrate the role of the optimal transport map at different stages** of the procedure [Pos6]. Moreover, we reported **standard deviations and $p$-values** to demonstrate statistical significance [Pos6] and to analyze the stability behavior of the method [e9LB].

3. *Theoretical Justification:* **We clarified the assumptions and conditions required for the derivation of the generalization bound** [Pos6, bhcj, e9LB]. Building on this, we proposed the method to **estimate the intractable term appearing in the bound** [bhcj]. In addition, we introduced an **extra proposition and proof regarding the selection of $\eta$—the unbalanced coefficient in UOT**—providing guidance on how to set this parameter [eTpZ].
3. *Limitation Discussions:* **We expanded the discussion of the computational limitations** associated with the parameterizations $w_\phi$ and $\boldsymbol{T}_\theta$, and **outlined potential future directions to mitigate these costs** [eTpZ, e9LB].
4. *Figure Revision:* To avoid any misunderstanding regarding ethical and attribution issues, we **revised Figure 2 and explicitly cited the corresponding reference in its caption** (which was already cited in the technical section of the initial submission) [e9LB]. To better illustrate why E‑SUOT performs better than the conventional Langevin‑based approach for classification task, we also **added the corresponding decision boundary in Figure 1** [e9LB].
6. *Notations*: We **added the Technical Terminology Table** in Appendix A to clarify notation and improve readability [eTpZ].

---

We are deeply grateful for all the feedback and suggestions that have helped us strengthen both the theoretical and empirical aspects of this work. We respectfully ask for your kind consideration of these revisions and clarifications in the final evaluation of our submission.

Sincerely,
Authors of Submission 18305

---

### Meta-Review · Area_Chair_kXxP · 2025-12-24

**Summary:**

This paper studies gradual domain adaptation and proposes an entropy-regularized semi-dual unbalanced optimal transport framework (E-SUOT) to improve flow-based intermediate domain generation. Reviewers generally agree that the theoretical formulation is mathematically sound and that the proposed framework is carefully developed. However, after considering the rebuttal and post-discussion feedback, substantial concerns remain regarding the level of methodological novelty, the realism and coverage of the experimental evaluation, and the strength of empirical evidence supporting the necessity of the proposed formulation in practical GDA settings. In particular, while the semi-dual reformulation and entropy regularization are well motivated, similar ideas have been explored in prior optimal transport and generative modeling literature, and the incremental novelty specific to gradual domain adaptation remains limited. Moreover, despite additional experiments added in the rebuttal, the evaluation still relies heavily on relatively controlled settings and strong preprocessing choices, leaving open questions about robustness under more realistic domain shifts. Overall, while the work is well executed and theoretically grounded, the remaining concerns regarding novelty and empirical breadth leave the submission slightly below the acceptance threshold for ICLR in its current form.

**Reviewer Concerns:**

Reviewer e9LB: Despite the rebuttal, concerns remain about whether the proposed formulation provides a sufficiently distinct conceptual advance over existing OT-based approaches, particularly given similarities to prior semi-dual and entropy-regularized methods.

Reviewer eTpZ: Theoretical analysis is solid, but questions persist regarding how tightly the theory explains the observed empirical gains and whether the assumptions adequately reflect realistic GDA scenarios.

Reviewer bhcj: Empirical results are generally positive, yet the evaluation remains limited in scope, making it difficult to assess robustness and practical impact beyond the studied settings.

Reviewer Pos6: While the rebuttal improves experimental coverage, concerns about methodological novelty and dependence on specific design and preprocessing choices remain only partially resolved.

**Reviewer Scores:**

Reviewer e9LB: Likely no change.

Reviewer eTpZ: Likely no change.

Reviewer bhcj: Likely no change or a slight increase.

Reviewer Pos6: Likely a modest increase.

---

### Decision · Program_Chairs · 2026-01-26

Reject